# Why Keep Your Doubts to Yourself? Trading Visual Uncertainties among Vision-Language Models

**Jusheng Zhang**[1†], **Yijia Fan**[1†], **Kaitong Cai**[1], **Jing Yang**[1],
**Jiawei Yao**[2], **Jian Wang**[3], **Guanlong Qu**[4], **Ziliang Chen**[5*], **Keze Wang**[1*]

[1]Sun Yat-sen University    [2]University of Washington    [3]Snap Inc.
[4]Syracuse University.    [5] Peng Cheng Laboratory.

## Abstract

Vision–Language Models (VLMs) enable powerful multi-agent systems, but scaling them is economically unsustainable: coordinating heterogeneous agents under information asymmetry often spirals costs. Existing paradigms, such as Mixture-of-Agents and knowledge-based routers, rely on heuristic proxies that ignore costs and collapse uncertainty structure, leading to provably suboptimal coordination. We introduce Agora, a framework that reframes coordination as a decentralized market for uncertainty. Agora formalizes epistemic uncertainty into a structured, tradable asset (perceptual, semantic, inferential), and enforces profitability-driven trading among agents based on rational economic rules. A market-aware broker, extending Thompson Sampling, initiates collaboration and guides the system toward cost-efficient equilibria. Experiments on five multimodal benchmarks (MMMU, MMBench, MathVision, InfoVQA, CC-OCR) show that Agora outperforms strong VLMs and heuristic multi-agent strategies, e.g., achieving +8.5% accuracy over the best baseline on MMMU while reducing cost by over 3×. These results establish market-based coordination as a principled and scalable paradigm for building economically viable multi-agent visual intelligence systems.

## 1 Introduction

The rapid advancement of Vision-Language Models (VLMs) (Li et al., 2022; 2023; Liu et al., 2023a; Bai et al., 2023; Zhang et al., 2025f;g;b;c;h) has propelled the development of multi-agent systems (MAS) (Guo et al., 2024; Wang et al., 2024c), moving us closer to the vision of powerful, collective intelligence. Yet, as these systems scale, they inevitably collide with foundational challenges from economic theory: coordinating self-interested agents under **information asymmetry** and making globally optimal decisions under **bounded rationality**. We argue current paradigms fail to address these root problems, leading to a crisis of economic viability where operational costs spiral, precluding effective, large-scale deployment (Gandhi et al., 2025). This inefficiency stems from a failure to treat intelligence not as a brute-force commodity, but as a scarce economic resource requiring principled management.

Existing coordination strategies can be understood as *heuristic patches*, i.e., computationally cheap workarounds for these deep-seated barriers. Paradigms like Mixture-of-Agents (MoA) (Guo et al., 2024) or knowledge-based routers (e.g., KABB) (Zhang et al., 2025i) attempt to bypass the complexity of true optimization by relying on simplistic proxies for value, such as consensus or semantic similarity. As we formally prove in Section 2, these heuristics render the systems fundamentally **agnostic** to the core economic variables of cost and the fine-grained structure of uncertainty. This agnostic nature is not a minor flaw but a theoretical dead-end, leading to provably suboptimal performance and systemic waste.

To dismantle this economic bottleneck, we argue for a paradigm shift: from heuristic patches to a mechanism that embraces the decentralized nature of the problem. Accordingly, we construct **Agora**, a framework that redesigns multi-agent coordination as a decentralized micro-economy.

---

*†Equal contribution. *Corresponding authors.

Agora does not attempt to approximate a central planner; instead, it uses market-based mechanisms to achieve efficient coordination *despite* information asymmetry and bounded rationality. Within this framework, cognitive uncertainty is no longer a monolithic liability but is "minted" into a quantifiable, tradable asset. Agents, guided by price signals and driven by economic incentives (Gale & and, 1962; von Neumann & Morgenstern, 2004; Akerlof, 1970), trade this asset to reveal private information and drive the entire system towards a cost-effective equilibrium.

Our methodology, detailed in Section 3, provides a constructive, non-agnostic solution. We first establish a **multi-dimensional uncertainty quantification model**, creating a structured asset that makes the system structure-aware. Second, we introduce a **profitability-driven trading protocol** that enforces economic rationality, making the system cost-aware. Finally, the entire market is orchestrated by an intelligent **market-aware Broker**, which uses a sophisticated utility function to find economically sound initializations for the collaborative process.

Our comprehensive experiments on multiple visual understanding benchmarks (e.g., MMMU (Yue et al., 2024), MMBench (Liu et al., 2023b)) demonstrate that Agora not only achieves state-of-the-art performance but also dramatically improves cost-effectiveness, validating our market-based paradigm. This work lays a theoretical and practical foundation for building truly scalable and economically viable multi-agent intelligent systems.

## 2 PROBLEM FORMULATION

The rise of multi-agent systems (MAS) promises powerful collective visual intelligence, yet this ambition faces a crisis of economic viability: soaring operational costs preclude scalable deployment. The bottleneck lies not in hardware, but in a conceptual failure—treating intelligence as a brute-force commodity rather than a scarce economic resource. When cognitive uncertainty, the primary cost driver, is handled without economic discipline, redundant computation ensues, making decisions prohibitively expensive. This section formalizes the problem and traces it to the heuristic-driven coordination paradigms dissected in Subsection 2.2.

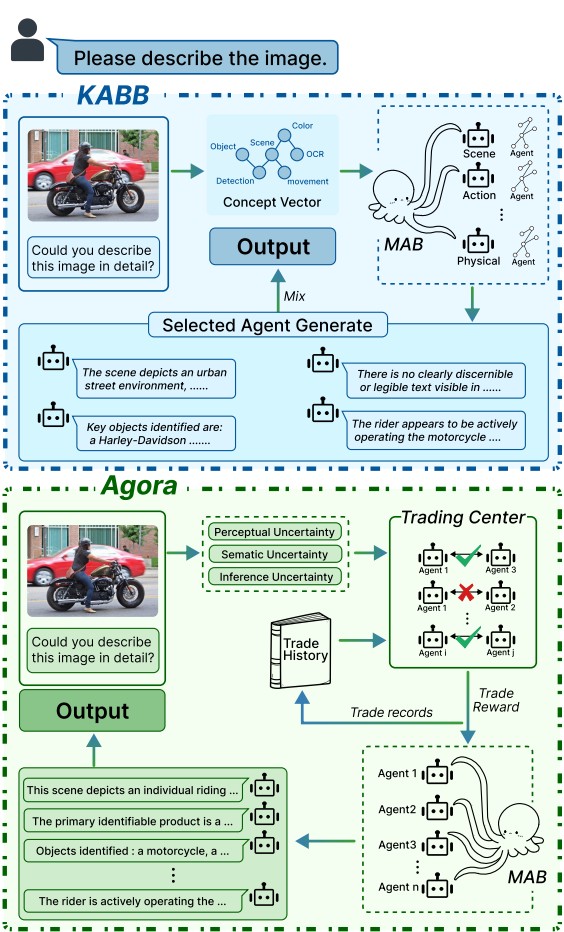

Figure 1: Comparison of heuristic coordination and Agora. Unlike heuristics that rely on flawed proxies, Agora forms a dynamic market for uncertainty, where emergent prices enable coordination.

### 2.1 THE ECONOMIC OBJECTIVE OF MULTI-AGENT COORDINATION

To ground our analysis, we first establish the ideal of economic rationality that any advanced MAS should pursue. **Setup.** We consider a system with a set of $N$ heterogeneous VLM agents, $\mathcal{A} = \{a_1, \ldots, a_N\}$. Each agent $a_i$ is defined by a unit processing cost $c_i > 0$ and an expertise vector $\boldsymbol{\xi}_i = [\xi_{i,\text{perc}}, \xi_{i,\text{sem}}, \xi_{i,\text{inf}}]^T$, where $\xi_{i,k} \in [0,1]$ quantifies its efficiency on uncertainty type $k$. For any task $t$ drawn from a distribution $\mathcal{T}$, the system confronts an initial epistemic uncertainty vector $\mathbf{u}(t) = [u_{\text{perc}}, u_{\text{sem}}, u_{\text{inf}}]^T$. For a centralized table and detailed definitions of all core variables and functions, please refer to Appendix L.

**Objective.** The system's goal is to learn an allocation policy $\pi$ that performs principled economic optimization. This policy must route uncertainty components to the most suitable agents to minimize total expected operational cost, while ensuring the final uncertainty is resolved to an acceptable level $\epsilon$. This is the core constrained optimization problem:

$$\min_{\pi} \mathbb{E}_{t \sim \mathcal{T}} \left[ \mathcal{C}(\pi, \mathbf{u}(t), \mathbf{c}, \Xi) \right] \quad \text{s.t.} \quad \|\mathbf{u}_{\text{final}}\| \leq \epsilon \quad (1)$$

where $\mathcal{C}(\cdot)$ is the total cost function, $\mathbf{c}$ is the vector of agent costs, and $\Xi$ is the matrix of agent expertise.

Figure 2: Final epistemic uncertainty of Agora (blue, 0.16) vs. KABB-VLM (orange, 0.21).

## 2.2 THE FAILURE OF HEURISTIC PROXIES FOR ECONOMIC RATIONALITY

Existing coordination paradigms fail to solve Eq. 1 because they do not perform true optimization, but instead rely on heuristic proxies fundamentally misaligned with the economic objective. We highlight two dominant paradigms that exemplify this failure. **1. Aggregation-Based Heuristics (e.g., MoA):** These equate statistical consensus with epistemic truth. Models like Mixture-of-Agents (MoA) assume that aggregating multiple agent outputs converges on the correct answer, which only holds if errors are independent and identically distributed (i.i.d.). In MAS with shared architectural biases, this assumption breaks down, leading to systemic irrationality. *Proposition 1 (Correlated Error Amplification).* Let $\mathcal{S}_{\text{prop}} \subset \mathcal{A}$ be a set of agents with a common perceptual bias. For ambiguous inputs, they will likely produce correlated hallucinations. An aggregator seeking consensus will then amplify this shared error. **2. Routing-Based Heuristics (e.g., KABB):** These rely on proxies for value, derived from historical performance and semantic similarity, to guide agent selection. State-of-the-art routers maximize a scoring function, e.g.:

$$S = \alpha \cdot P_{\text{hist}} + \beta \cdot \text{Sim}_{\text{sem}} \quad (2)$$

where $P_{\text{hist}}$ is historical performance and $\text{Sim}_{\text{sem}}$ is semantic similarity. This surrogate conflates past performance with future cost-effectiveness, remaining **Cost-Agnostic** (the cost vector $\mathbf{c}$ is absent) and **Uncertainty-Structure-Agnostic** (the vector $\mathbf{u}(t)$ is collapsed into a scalar proxy). This structural ignorance results in higher residual uncertainty, as empirically demonstrated in Figure 2, where our structure-aware Agora yields markedly lower final epistemic uncertainty than the heuristic baseline.

## 2.3 THE CORE CHALLENGE: A CALL FOR A NEW PARADIGM

The specific flaws in aggregation and routing are manifestations of a deeper, shared theoretical limitation, which we formalize as agnostic coordination. **Definition 1** (*Agnostic Coordination*). A coordination mechanism $\mathcal{M}$ is defined as agnostic if its agent selection process is (i) *Cost-Agnostic* (invariant to agent processing costs) and (ii) *Uncertainty-Structure-Agnostic* (collapses the uncertainty vector into a scalar proxy). Both MoA and KABB are archetypes of agnostic coordinators, as conceptually illustrated in Figure 1. MoA's consensus heuristic disregards the cost of polling agents and the specific structure of the uncertainty it aims to resolve. KABB's routing heuristic, as shown in Eq. 2, explicitly demonstrates both agnostic properties. This shared, fundamental flaw leads to provably suboptimal performance. **Theorem 1** (*The Inefficiency Theorem for Agnostic Coordination*). Any coordination mechanism $\mathcal{M}$ that is agnostic (per Definition 1) is not guaranteed to solve the objective in Eq. 1 and is provably suboptimal for any task where the heuristically superior agent is not the most cost-effective resolver. This impasse shows that incremental fixes to heuristic coordinators are inadequate. A paradigm shift is needed—from heuristic proxies to non-agnostic mechanisms capable of genuine economic reasoning. This crystallizes our central question, addressed in Section 3: how to design a coordination mechanism that explicitly integrates cost and uncertainty to navigate the optimization landscape of Eq. 1?

## 3 METHODOLOGY: THE AGORA MARKET FRAMEWORK

To address the theoretical deficiencies of agnostic coordination identified in Section 2, we introduce Agora: a framework that recasts multi-agent coordination as a decentralized micro-economy. Our

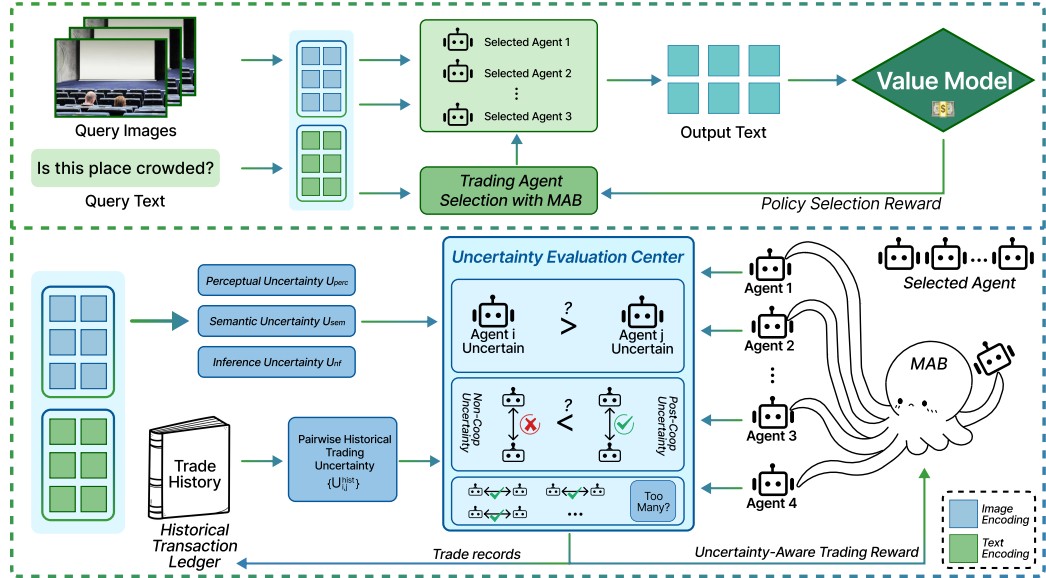

Figure 3: In Agora, query uncertainty is split into perceptual ($U_{\text{perc}}$), semantic ($U_{\text{sem}}$), and inferential ($U_{\text{inf}}$). A market-aware broker trades these among agents for efficient resolution.

methodology provides a constructive, non-agnostic solution to the optimization problem in Eq. 1 by designing a system that is inherently cost-aware and structure-aware. At its core, this is achieved by turning uncertainty into a quantifiable, tradable asset and defining protocols for its efficient reallocation. As established in our analysis of related work (Appendix B), prior heuristic-driven paradigms are fundamentally agnostic to these factors. In response, Agora introduces the principled economic mechanism for coordination illustrated in Figure 3.

## 3.1 Establishing the Market: From Uncertainty to Tradable Assets

A market cannot exist without a well-defined asset. To counter the *Uncertainty-Structure-Agnosticism* from Theorem 1, our first step is to "mint the currency" by formalizing cognitive uncertainty as a structured, quantifiable portfolio. We decompose total uncertainty **u** into two classes: a tradable component, **Epistemic Uncertainty** ($\mathbf{u_{epis}}$), which represents the reducible information gap from our problem formulation; and a non-tradable component, **Aleatoric Uncertainty** ($\mathbf{u_{alea}}$), which represents irreducible systemic risk. The tradable asset, $\mathbf{u_{epis}}$, is a vector in a three-dimensional state space, $\mathbf{u_{epis}} = [u_{\text{perc}}, u_{\text{sem}}, u_{\text{inf}}]^T$, corresponding to the fundamental cognitive domains of perception, semantics, and inference. This vectorization transforms a monolithic problem into a portfolio of distinct assets that can be independently priced and traded. Each agent $a_i$ maintains an uncertainty portfolio $\mathbf{U}(a_i, t)$, which is the linear superposition of its self-generated uncertainty and the net uncertainty acquired through market trades:

$$\mathbf{U}(a_i, t) = \mathbf{U}_{\text{base}}(a_i, t) + \sum_{j \neq i} \mathbf{U}_{\text{transfer}}(a_j \to a_i, t) \tag{3}$$

The value of transferred uncertainty, $\mathbf{U}_{\text{transfer}}$, is aggregated from a historical transaction ledger, weighting past trades based on relevance and cost-effectiveness.

## 3.2 The Core Mechanism: A Profitability-Driven Trading Protocol

With a structured asset in place, we now introduce the core mechanism designed to overcome *Cost-Agnosticism*. This mechanism is a **Profitability-Driven Trading Protocol** that governs all transactions based on pure economic rationality. A trade is initiated when an arbitrage opportunity—a potential for system-wide cost reduction—is identified. To evaluate this, we calculate the change in total cost, or **cost delta** ($\Delta\mathcal{C}$), that would result from reallocating an uncertainty packet. The

derivation for a packet of magnitude $T_{ij}(t)$ being transferred from agent $a_i$ to $a_j$ is:

$$\Delta \mathcal{C}(T_{ij}(t)) = \underbrace{[c_i(U_i(t) - T_{ij}(t)) + c_j(U_j(t) + (1 - \xi_j)T_{ij}(t))]}_{\text{Cost After Trade}} - \underbrace{[c_i U_i(t) + c_j U_j(t)]}_{\text{Cost Before Trade}}$$

$$= T_{ij}(t) \cdot [c_j(1 - \xi_j) - c_i] \tag{4}$$

This leads to a simple, powerful admissibility rule. A trade is executed if and only if it is profitable ($\Delta \mathcal{C} < 0$) and feasible, meaning the receiving agent $a_j$ possesses the required cognitive capacity $C_j(t)$:

$$\text{Execute trade}(i \to j, T_{ij}(t)) \iff (\Delta \mathcal{C}(T_{ij}(t)) < 0) \wedge (U_j(t) + T_{ij}(t) \leq C_j(t)) \tag{5}$$

This protocol, by its very construction, is both cost-aware and structure-aware, thus violating both conditions for suboptimality from Theorem 1. Each admissible trade represents a greedy step that descends the cost landscape of the global objective function in Eq. 1.

### 3.3 MARKET EXECUTION: THE BROKER AND THE AGORA ALGORITHM

The market is set in motion by an intelligent **Broker**, an extension of Thompson Sampling (TS) that finds an economically sound starting point for the decentralized optimization. It selects an initial agent by maximizing a market-aware expected utility function, $\tilde{\theta}_S^{(t)}$:

$$\tilde{\theta}_S^{(t)} = (\mathbb{E}[\text{Reward}_S^{(t)}] - \text{Cost}_S^{(t)}) \cdot \exp(-\lambda \cdot \text{Dist}(S, t)) \cdot U_{\text{strategic}}(S)^\omega \cdot \text{Synergy}(S)^\eta \cdot \gamma^{\Delta t} \tag{6}$$

where the terms account for expected reward minus cost, adjusted for task distance, strategic utility, agent synergy, and temporal decay (see Appendix L for details). The entire process is operationalized by the **Agora Algorithm**, presented in Algorithm 1. The algorithm proceeds in two phases: (1) a utility-maximizing initialization by the Broker, followed by (2) an iterative market phase. In this phase, the system performs a deterministic **greedy descent** on the total cost function by repeatedly applying the trading protocol from Eq. 5. This continues until no further profitable trades are possible, at which point the market has converged to a locally optimal and cost-efficient equilibrium.

---

**Algorithm 1** Agora: A Distributed Economic Optimization Algorithm

---

1: **Input:** Agent set $\mathcal{A}$, costs $\mathbf{c}$, expertise $\Xi$, initial uncertainty $\mathbf{u}_{\text{initial}}$, Broker MAB.
2: **Output:** Final allocation $\Pi$.
3:                                              ▷ *Phase 1: Utility-Maximizing Initialization*
4:   $a_{\text{handler}} \leftarrow$ Broker.select_initial_agent$(\mathcal{A}, \mathbf{u}_{\text{initial}})$
5:   Initialize allocation $\Pi$: $\mathbf{u}_{\text{handler}} \leftarrow \mathbf{u}_{\text{initial}}$; $\mathbf{u}_i \leftarrow \mathbf{0}$ for $i \neq$ handler.
6: **while** true **do**            ▷ *Phase 2: Iterative Greedy Cost Descent via Trading Protocol*
7:   best_trade $\leftarrow$ FindMostProfitableTrade$(\Pi, \mathcal{A}, \mathbf{c}, \Xi)$
8:   **if** best_trade $\neq$ null **then**
9:     Let $(i, j, k, \text{amt})$ be the components of best_trade
10:                                  ▷ Execute trade based on the protocol from Eq. 5
11:     $\mathbf{u}_j[k] \leftarrow \mathbf{u}_j[k] + \mathbf{u}_i[k]$;    $\mathbf{u}_i[k] \leftarrow 0$
12:   **else**
13:     **break**                        ▷ Market converged to a locally optimal equilibrium
14:   **end if**
15: **end while**
16: **return** $\Pi$

---

## 4 EXPERIMENTS

We conduct experiments to validate Agora, using NVIDIA A100 GPUs. The agent pool consists of five representative VLMs: `qwen2.5vl-72b-instruct`, `gemini-2.0-flash`, `qwen2.5vl-7b-instruct`, `gemma-3-27b`, and `gpt-4o-mini`. In Agora, an 'expert' or 'agent' is an active configuration of a base model with a specific prompt and role. The number of concurrent experts, denoted by $N$, varies by setup. We evaluate five aspects: (1) comprehensive visual understanding across benchmarks; (2) the role of our MAB strategy in uncertainty trading; (3) comparison with alternative routing and MAS strategies; (4) cost–performance trade-offs

Table 1: Comprehensive performance on visual benchmarks. Scores are percentages; best in **bold**, second best underlined. Agora (Ours) routes tasks within the baseline pool, parentheses show gains over the top.

| Model | MMMU(Val) | MMBench_V11_Test | MathVision | InfoVQA(test) | CC-OCR |
|---|---|---|---|---|---|
| qwen2.5vl-72b-instruct | 70.2% | 88.4% | 39.3% | 87.3% | 79.8% |
| gemini-2.0-flash | 70.7% | 83.0% | 41.3% | 83.2% | 73.1% |
| qwen2.5vl-7b-instruct | 58.6% | 82.6% | 25.1% | 82.6% | 77.8% |
| gemma-3-27b | 64.9% | 78.9% | 27.5% | 59.4% | 72.6% |
| gpt-4o-mini | 60.0% | 76.3% | 26.3% | 68.7% | 64.2% |
| gpt-4o-2024-08-06 | 70.7% | 74.3% | 30.4% | 68.7% | 66.6% |
| gemini-2.5-pro-exp-03-25 | **81.7%** | 88.3% | **63.5%** | 81.0% | 73.0% |
| InternVL3-78B | 72.2% | 87.7% | 43.1% | 84.1% | 80.3% |
| Agora(Ours) | 79.2%(+8.5%) | **89.5%**(+1.1%) | 44.3%(+2.0%) | **88.9%**(+1.6%) | **81.2%**(+1.4%) |

across $N$; (5) module/strategy ablations. **Comprehensive Visual Understanding Performance Experiment Setup.** Agora's performance is benchmarked against its constituent models (individually) and external SOTA VLMs, including gpt-4o-2024-08-06 (OpenAI, 2024), gemini-2.5-pro-exp-03-25 (Google, 2025), and InternVL3-78B (Chen et al., 2024b). Evaluation covers diverse benchmarks: MMMU (Val) (Yue et al., 2024), MMBench_V11_Test (Liu et al., 2023b), MathVision (Wang et al., 2024b), InfoVQA (test) (Mathew et al., 2021), and CC-OCR (Yang et al., 2024). All models, including baselines and SOTA comparators, are accessed via the OpenRouter API. We use greedy decoding ('do_sample=False') for determinism and comparability. Additional details are in the appendix. **Experimental Results and Analyses.** Table 1 shows that Agora delivers robust performance across challenging benchmarks. It achieves new SOTA on MMBench_V11_Test, InfoVQA, and CC-OCR, owing to its architecture that explicitly manages and trades uncertainties among heterogeneous agents. On reasoning-heavy tasks like MMMU and MathVision, gemini-2.5-pro-exp-03-25 performs strongly as a specialized "thinking model," but Agora still secures second place. Overall, these results highlight Agora's effective collaborative capability for complex vision–language tasks. Agora achieves consistent gains (+1.1–8.5% across benchmarks), confirming effectiveness in collaborative problem-solving via dynamic uncertainty management.

## 4.1 COMPARISON WITH ALTERNATIVE ROUTING AND MULTI-AGENT STRATEGIES

**Experiment Setup.** We benchmark Agora (Ours) against representative router models and multi-agent strategies, including FrugalGPT (Chen et al., 2024a), RouteLLM (Ong et al., 2024), EmbedLLM (Zhuang et al., 2025), HybridLLM (Ding et al., 2024), KABB (Zhang et al., 2025i), and MOA (Wang et al., 2024a). All methods are adapted to the VLM multi-agent setting and operate on the same agent pool (N=6). Evaluation is performed on **MMBench_V11_Test** (Liu et al., 2023b), reporting Accuracy (%), Relative Cost (normalized to Agora=1.00), Average Inference Time (s), Collaboration Overhead Index (COI), and Final Epistemic Uncertainty ($U_{final\_epis}$). Details of model adaptation and hyperparameters are in Appendix K. **Experimental Results and Analyses.** As shown in Figure 4, Agora attains the highest accuracy (89.50%) while remaining cost-efficient. KABB-VLM and MOA achieve competitive accuracy (87.12%, 86.65%) but at much higher cost (1.24× and 3.11×), COI (1.53, 1.82), and residual uncertainty (0.21, 0.25). In contrast, Frugal-GPT, RouteLLM, EmbedLLM, and HybridLLM reduce cost (0.73–0.91) but suffer notable accuracy drops (–8 to –9.6 points) and higher uncertainty (0.27–0.33). These results underline Agora's superior accuracy–efficiency trade-off. **Experiment Setup.** To evaluate economic efficiency, we analyze Agora's cost–performance on **MMBench_TEST_V11**, varying agent pool size ($N = 1$–9) and comparing with baselines, external SOTA VLMs, and KABB-VLM. The Cost–Performance Ratio is defined as relative cost (gpt-4o-mini=1.0) over accuracy, using OpenRouter prices; lower is better.

## 4.2 THE ROLE OF THE MARKET-AWARE MAB STRATEGY

To validate the central role of our market-aware Multi-Armed Bandit (MAB) broker, we conducted a comparative experiment on the MMMU (Val) benchmark. In this controlled setup, we replaced our selector with prominent heuristic (KABB) and reinforcement learning (PPO, MCTS, A2C, DQN)

Table 2: MAB strategy on MMMU (Val). Except "No Trading," all use multi-agent trading (COI > 1). Lower is better for $U_{final\_epis}$, COI; higher for Accuracy, UAPS. Best scores **bold**, second best underlined.

| Method | MMMU Acc. (%) | $U_{final\_epis}\downarrow$ | COI $\downarrow$ | UAPS (%) $\uparrow$ |
|---|---|---|---|---|
| Agora (Ours) | **79.0** | **0.15** | 1.2 | **70.5** |
| Agora (No Trading) | 75.5 | 0.22 | **1.0** | 65.0 |
| KABB Selector + Trading | 76.0 | 0.25 | 1.5 | 65.5 |
| PPO Selector + Trading | 74.0 | 0.28 | 1.6 | 62.0 |
| MCTS Selector + Trading | 74.5 | 0.26 | 1.4 | 63.0 |
| A2C Selector + Trading | 73.5 | 0.29 | 1.7 | 61.0 |
| DQN Selector + Trading | 73.0 | 0.30 | 1.7 | 60.0 |

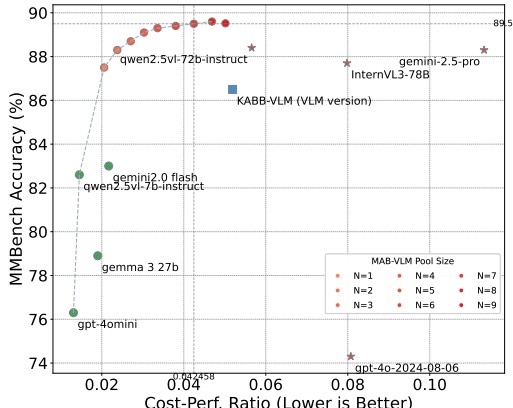

Figure 4: Comparison with alternative routing and multi-agent strategies on MMBench_V11_Test (N=6). Lower is better for Cost, Time, COI, and $U_{final\_epis}$; higher is better for Accuracy.

alternatives, while the underlying uncertainty trading protocol remained constant. The results, presented in Table 2, are decisive. Our MAB-based approach outperforms all baselines, achieving the highest accuracy (79.0%) and Uncertainty-Aware Performance Score (UAPS) of 70.5%. Notably, it surpasses the next-best heuristic selector (KABB) by a margin of 3.0% in accuracy and 5.0 UAPS points. While the RL agents demonstrate learning capabilities, they struggle to match the efficiency of our method within this economic coordination task, consistently yielding lower scores and higher final epistemic uncertainty ($U_{final\_epis}$). These findings underscore that the specific design of our market-aware Broker, which leverages an economically-informed utility function, is a critical contributor to the Agora framework's superior performance and efficiency.

## 4.3 COST AND PERFORMANCE BALANCE ANALYSIS

**Experimental Results and Analyses.** Figure 5 plots the accuracy-cost trade-off, where the Cost-Performance Ratio (lower is better) reveals the economic efficiency of different strategies. The results demonstrate Agora's ability to establish a superior Pareto frontier. Even with a single agent type ($N = 1$), Agora achieves a competitive 87.5% accuracy at an exceptionally low cost ratio (0.02057), outperforming even costly SOTA models like 'gemini-2.5-pro' and 'InternVL3-78B'. As the agent pool diversifies, accuracy steadily climbs to a peak of 89.6% at $N = 8$, with only marginal changes at $N = 9$. Crucially, every Agora configuration ($N \geq 1$) maintains a significantly better cost ratio than strong baselines like 'qwen-72b' (0.05656) and alternative multi-agent systems like KABB-VLM (0.05191).

These trends reveal key insights into Agora's economic design. First, the remarkable efficiency of the $N = 1$ case is not merely about using a cheap model; it highlights the intelligence of our market-aware broker, which selects the most suitable configuration for a given

Figure 5: Cost–Performance vs. Accuracy on MMBench. Agora achieves a stronger Pareto frontier, improving accuracy with sub-linear cost growth as the agent pool expands.

task, avoiding unnecessary costs. Second, the
graceful scaling from $N = 2$ to $N = 8$ validates our core thesis: as the market gains access to more specialized agents (a more heterogeneous pool), the uncertainty-trading mechanism more effectively allocates cognitive labor to the cheapest specialist. This allows the system to push the accuracy boundary without a proportional surge in cost. Finally, the performance plateau around $N = 8$ indicates a point of diminishing returns, a classic economic principle. It suggests that Agora does not require an ever-expanding, costly pool of agents to maintain its edge. Instead, it efficiently leverages a finite set of resources to approach an optimal performance ceiling. This economically rational behavior stands in stark contrast to the brute-force strategy of monolithically applying a single, expensive SOTA model to all problems, a tactic that Figure 5 shows is fundamentally inefficient.

## 4.4 Module and Strategy Ablation Studies

**Experimental Setup.** To dissect the contribution of each component in Agora, we conduct an ablation study on the key multiplicative factors within our market-aware Thompson Sampling utility function: $\tilde{\theta}_S^{(t)} = (\mathbb{E}[\text{Reward}_S^{(t)}] - \text{Cost}_S^{(t)}) \cdot \exp(-\lambda \cdot \text{Dist}(S, t)) \cdot \gamma^{\Delta t} \cdot \text{Synergy}(S)^\eta \cdot U_{\text{strategic}}(S)^\omega$. Specifically, we create variants by individually ablating the Strategic Uncertainty ($U_{\text{strategic}}$), Synergy, Task Match (Dist), and Time Decay ($\Delta t$) factors. These variants are benchmarked against a baseline, "Only Net Return," which relies solely on the expected net return for agent selection. All experiments in this study are performed on the **MMBench_V11_Test** dataset, utilizing a consistent agent pool of $N = 6$ experts. For each ablated variant, its corresponding term in the utility function is neutralized by setting it to one. Further ablation on key hyperparameters is detailed in Appendix G. **Experimental Results and Analyses.**

As presented in Table 3, the results validate the efficacy of the complete Agora strategy and quantify the contribution of each component. The full model outperforms all variants, achieving the highest accuracy (89.50%), lowest final epistemic uncertainty (0.16), lowest Collaboration Overhead Index (COI, 1.25), and highest Uncertainty-Aware Performance Score (UAPS, 78.33%). Ablating any strategic factor impairs performance. Critically, removing the novel Strategic Uncertainty ($U_{\text{strategic}}$) factor causes the most substantial

Table 3: Ablation of the Agora strategy on MMBench_V11_Test. Uncertainty trading is enabled for all variants. Rel. Cost is normalized to the full model. Best results are in **bold**.

| Variant | Acc. (%)↑ | $U_{\text{final}}$↓ | COI↓ | UAPS (%)↑ | Rel. Cost↓ |
|---|---|---|---|---|---|
| Agora (Full) | **89.50** | **0.16** | **1.25** | **78.33** | **1.00** |
| w/o $U_{\text{strategic}}$ | 86.42 | 0.23 | 1.45 | 71.58 | 1.06 |
| w/o Synergy | 87.91 | 0.19 | 1.30 | 74.88 | 1.03 |
| w/o Dist | 88.53 | 0.18 | 1.27 | 76.21 | 1.01 |
| w/o $\Delta t$ | 89.05 | 0.17 | 1.26 | 77.14 | **1.00** |
| Only Net Return | 82.15 | 0.31 | 1.08 | 60.72 | 0.92 |

performance degradation (Accuracy -3.08%, $U_{\text{final\_epis}}$ +0.07, UAPS -6.75 points, Cost +6%), underscoring its pivotal role in guiding agent selection toward profitable uncertainty trades. The removal of the Synergy, Task Match, and Time Decay factors also leads to measurable performance drops, confirming their positive contributions. In stark contrast, the "Only Net Return" baseline, which ignores all strategic heuristics, performs substantially worse than any other variant (e.g., -7.35% accuracy and -17.61 UAPS points vs. the full model). This confirms that all strategic components are integral to achieving the high-accuracy, cost-efficient coordination that defines the Agora framework.

## 5 Conclusion

We propose **Agora**, a market-based framework for coordinating multi-agent Vision–Language Models (VLMs) under explicit accuracy–cost trade-offs. Unlike heuristic methods such as Mixture-of-Agents or knowledge-based routers, which often collapse uncertainty into fixed routing decisions and overlook the cost of additional computation, Agora casts epistemic uncertainty as a structured, tradable asset across perceptual, semantic, and inferential dimensions. Guided by a market-aware Thompson Sampling broker, a profitability-driven protocol enables agents to buy, sell, and refine uncertainty estimates rationally, allowing the system to allocate expert effort only when it is expected to improve the final decision. This design encourages cost-efficient equilibria rather than unbounded agent invocation. Experiments on five benchmarks show consistent gains, achieving up to **+8.5%** accuracy while reducing cost by more than $3\times$. These results highlight Agora's practical value for scalable and cost-aware multi-agent VLM coordination.

ETHICS STATEMENT

This work adheres to the ICLR Code of Ethics. Our study does not involve human-subjects research, the collection of personally identifiable information, or the annotation of sensitive attributes. We do not create or distribute any new human data. All experiments are conducted on publicly available, widely used vision–language benchmarks strictly under their respective licenses and terms of use. The methods developed in this paper are designed for advancing academic understanding of multi-agent coordination and are not intended for deployment in sensitive or high-stakes applications without additional safeguards.

REPRODUCIBILITY STATEMENT

To ensure reproducibility, we provide detailed descriptions of our experimental settings, including datasets, baselines, and evaluation metrics, in the main text and appendix. Hyperparameters, training protocols, and ablation results are comprehensively documented to allow replication of results. All models are implemented using standard open-source frameworks, and benchmark datasets are accessed through their official releases. Pseudocode, algorithmic steps, and theoretical proofs are included to enable faithful reproduction of both the methodology and results. Additional runtime and configuration details are provided in the supplementary material for transparency.

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

## A APPENDIX

### SUPPLEMENTARY MATERIAL OVERVIEW

This Supplementary Material provides a detailed expansion of the Agora framework presented in the main paper. It begins with foundational **Preliminary** concepts in Appendix A and a review of **Related Work** in Appendix B. Appendix C offers in-depth **Theoretical Proofs and Supplements** for the core mechanisms discussed. Further empirical validation is provided through an analysis of the **Impact of Agent Pool Configuration** in Appendix D, a **FLOPs Comparison and Computational Efficiency** study in Appendix E, and an extensive **Supplementary Core Component Ablation Discussion** in Appendix F. To ensure reproducibility and transparency, we detail the **Hyperparameter Ablation Experiments** in Appendix G, list all **Hyperparameters Used in the Experiments** in Appendix K, present a **Runtime Analysis** in Appendix I, and include the **Prompt Setting Statement** for our VLM agents in Appendix J. Finally, Appendix M offers a qualitative **Case Analysis** with examples of successful and unsuccessful expert collaborations.

### A PRELIMINARY

**Vision-Language Models (VLMs) and Decision Uncertainty:** VLMs are systems that process multimodal inputs, such as visual data $I$, and generate textual responses $R$ based on task descriptions $T$. At their core, they rely on a Large Vision-Language Model. Formally, a VLM agent $a$ acts as a function $f_a$: $R = f_a(I, T)$, where $I \in \mathcal{I}$ is the visual input space, $T \in \mathcal{T}$ is the task description space, and $R \in \mathcal{R}$ is the response space. Each agent $a$ incurs a processing cost $c_a$, reflecting computational resource usage. In heterogeneous multi-agent setups, the agent set $\mathcal{A} = \{a_1, a_2, \ldots, a_n\}$ varies significantly in capabilities and costs.

Uncertainty plays a crucial role in agent decision-making. For an agent $a$, input $I$, and task $T$, uncertainty $U$ measures the dispersion in the response probability distribution $P_a(R|I, T)$, often quantified via Shannon entropy $\mathcal{H}$:

$$U(a, I, T) = \mathcal{H}(P_a(R|I, T)) = -\sum_{r \in \mathcal{R}} P_a(r|I, T) \log P_a(r|I, T).$$

Here, $P_a(R|I, T)$ denotes the distribution over possible responses $r \in \mathcal{R}$. Higher $U$ indicates lower confidence, potentially increasing computational costs. In multi-agent systems, uncertainty can be decomposed into epistemic (reducible through collaboration) and aleatoric (irreducible) components, allowing for targeted trading to optimize resource allocation.

**Multi-Armed Bandit Problem (MAB):** MAB involves sequential decisions where a learner selects from actions (arms) to maximize cumulative rewards. Selecting arm $a$ at time $t$ yields a random reward $X_a(t)$ from an unknown distribution.

Thompson Sampling (TS) addresses MAB by balancing exploration and exploitation via Bayesian methods. For each arm $a$, it maintains a posterior on reward probability $\theta_a$, often a Beta distribution $\text{Beta}(\alpha_a, \beta_a)$. At each step, sample $\theta_a \sim \text{Beta}(\alpha_a, \beta_a)$ and choose:

$$a^* = \arg\max_a \theta_a.$$

Update after reward $r$:

$$(\alpha_a, \beta_a) \leftarrow \begin{cases} (\alpha_a + 1, \beta_a) & \text{if } r = 1, \\ (\alpha_a, \beta_a + 1) & \text{if } r = 0. \end{cases}$$

This approach is particularly useful in agent selection, as it adapts to performance over time, reducing regret in uncertain environments.

**Cost-Benefit Modeling and Comparative Advantage:** In multi-agent systems, agent $a_i$'s efficiency in handling uncertainty $U$ is modeled by cost function $C_i(U) = \alpha_i \cdot U + \beta_i$, where $\alpha_i$ is the marginal cost per unit uncertainty and $\beta_i$ is the fixed cost. Total system cost is $C_{\text{total}} = \sum_{i=1}^n C_i(U_i)$, with $U_i$ assigned to $a_i$.

Drawing from comparative advantage theory, our framework reallocates uncertainty based on relative efficiencies: agents with absolute disadvantages can still improve system efficiency if strengths

differ across dimensions. For agents $a_i$, $a_j$ and dimensions $d_1$, $d_2$, the comparative advantage index is:

$$\text{CAI}(a_i, a_j, d_1, d_2) = \frac{C_i(d_1)/C_i(d_2)}{C_j(d_1)/C_j(d_2)} < 1,$$

implying $a_i$ advantages in $d_1$ relative to $d_2$, and vice versa. Trading then reduces total cost:

$$\Delta C_{\text{total}} = [C_i(U_i - \Delta U_{d_1}) + C_j(U_j + \Delta U_{d_1})] - [C_i(U_i) + C_j(U_j)] < 0.$$

**Analysis and Improvements in Agora:** Traditional multi-agent coordination often relies on heuristics like consensus in Mixture-of-Agents (MoA) or semantic routing in KABB, which are cost-agnostic and collapse uncertainty into scalars, leading to suboptimal performance as proven by the Inefficiency Theorem. These approaches fail to address information asymmetry and bounded rationality, resulting in high costs and inefficiencies.

Agora improves upon this by framing coordination as a decentralized market for uncertainty, minting it into tradable assets (perceptual, semantic, inferential). This structure-awareness enables profitability-driven trades, ensuring cost-aware optimization. The market-aware Broker, extending TS, initializes collaborations efficiently, while the trading protocol greedily descends the cost landscape. Experiments show Agora achieves up to +8.5% accuracy on MMMU with 3× cost reduction, demonstrating scalable, economically viable intelligence.

## B RELATED WORK

### B.1 VISION-LANGUAGE MODELS IN MULTI-AGENT SYSTEMS

The integration of Vision-Language Models (VLMs) (Parascandolo et al., 2025; Peng et al., 2024; Kim & Ji, 2024; Xu et al., 2024; Wang et al., 2023; Radford et al., 2021) into multi-agent systems (MAS) has unlocked new capabilities for collaborative multimodal tasks (Nash, 1950; Ma et al., 2024a;b; 2025c;b). However, prevailing coordination paradigms, such as centralized controllers or heuristic-based task allocators (Gregory et al., 2019; Han et al., 2024), often struggle with the economic realities of scaling these systems. They tend to overlook the steep computational costs inherent in large VLMs (Wang et al., 2024a) and rely on static uncertainty-handling mechanisms, which fundamentally limits their efficiency and scalability. In contrast, Agora introduces a market-driven framework that directly addresses these shortcomings. It enables agents to dynamically trade uncertainty as a resource, optimizing for both performance and cost by leveraging decentralized economic principles to resolve information asymmetry—a key limitation of prior heuristic-based approaches.

### B.2 UNCERTAINTY QUANTIFICATION AND MANAGEMENT

While uncertainty quantification is a recognized field in deep learning, particularly within Bayesian methods (LeCun et al., 2015) and active learning, its application in multi-agent VLM systems remains underdeveloped. Existing research is often limited in scope: many methods decompose uncertainty into epistemic and aleatoric types but focus primarily on single-agent settings (Gawlikowski et al., 2022; Zhang et al., 2025e;d; Ma et al., 2025a; Long et al., 2025; Shen et al., 2025). Other studies investigate uncertainty sharing for perceptual tasks (Hao et al., 2022) but lack a formal economic model for efficient resource allocation. Agora uniquely bridges this gap. It formalizes multi-dimensional uncertainty (perceptual, semantic, and inferential) as a structured, tradable asset. This enables a novel, profitability-driven trading protocol that reduces system-wide costs and enhances collaborative efficiency, moving beyond the static and heuristic methods found in existing literature.

### B.3 MULTI-ARMED BANDITS AND DECISION-MAKING

Multi-armed bandit (MAB) frameworks (Dai et al., 2023; Sun et al., 2025; Zhang et al., 2026; 2025a) are a cornerstone of sequential decision-making in MAS (de Curtò et al., 2023; Xia et al., 2025). Advanced methods like contextual bandits (Zhu et al., 2022; david cortes, 2025) and reinforcement learning-based MABs (Azizi et al., 2022; Gupta et al., 2023) incorporate state information to refine action selection. However, their direct application often falls short in the complex economic

landscape of large-scale MAS, as traditional MABs are typically engineered to maximize an abstract reward signal. They rarely situate the decision-making process within a formal economic framework that explicitly models the trade-offs between performance, computational cost, and the fine-grained structure of uncertainty.

In contrast, the Agora framework makes several novel contributions that extend the MAB paradigm from a simple decision-making tool to a market-aware economic broker: **Uncertainty as a Tradable Asset:** We are the first to formalize multi-dimensional cognitive uncertainty (perceptual, semantic, and inferential) as a quantifiable and tradable economic asset. This moves beyond merely using uncertainty as a feature for exploration. **Profitability-Driven Coordination:** We introduce a trading protocol governed by economic rationality, where agent collaboration is based on explicit cost-benefit analysis ($\Delta\mathcal{C}$) rather than heuristic rules. **Market-Aware Utility Function:** The Broker in Agora utilizes a novel, market-aware utility function (Eq. 6) that integrates not only expected reward but also explicit costs, task similarity, team synergy, and a unique *Strategic Uncertainty Index*. This design aligns the MAB's selection policy directly with the economic efficiency of the entire multi-agent system. This economically grounded approach yields superior cost-performance trade-offs compared to traditional MAB applications, marking a significant advancement in building truly viable decision-making frameworks for MAS.

## C  THEORETICAL PROOFS AND SUPPLEMENTS IN THE MAIN TEXT

### C.1  MULTI-DIMENSIONAL VISUAL UNCERTAINTY QUANTIFICATION MODEL (3.1)

To achieve fine-grained management and efficient trading of visual uncertainty, Agora proposes a multi-dimensional uncertainty quantification model. This model decomposes the overall uncertainty faced by an agent into three fundamental dimensions: perceptual uncertainty ($u_{\text{perc}}$), semantic uncertainty ($u_{\text{sem}}$), and inferential uncertainty ($u_{\text{inf}}$), as formalized in Section 3.

#### C.1.1  FORMAL DEFINITION AND EXPANSION OF CORE UNCERTAINTY DIMENSIONS

**a. Perceptual Uncertainty ($u_{\text{perc}}$)**  Perceptual uncertainty ($u_{\text{perc}}$) quantifies the lack of confidence in identifying raw visual signals (e.g., object categories, basic features) due to factors such as image quality and visual ambiguity. It is defined as follows:

$$u_{\text{perc}}(I, R) = \underbrace{f_{\text{perc}}}_{\substack{\text{Perceptual Uncertainty} \\ \text{Aggregation Function}}} \left( \underbrace{\text{Stat}}_{\substack{\text{Statistical Analysis} \\ \text{Function}}} \left( \underbrace{f_{\text{visual}}(I)}_{\substack{\text{Visual Features Extracted} \\ \text{from Image } I}} \right), \underbrace{\Psi(R)}_{\substack{\text{Contextual/Modulating Factors} \\ \text{Related to Response } R}} \right) \tag{7}$$

Detailed Expansion and Explanation: $I$: Input Visual Signal. $f_{\text{visual}}(I)$: Visual feature extraction module. For example, $f_{\text{visual}}(I) \rightarrow \mathbf{V}$, where $\mathbf{V}$ is a set of feature vectors extracted by a Convolutional Neural Network (CNN) or Vision Transformer (ViT). $\text{Stat}(f_{\text{visual}}(I))$: Statistical evaluation of the extracted visual features to quantify their clarity, consistency, or the model's raw confidence in these features. For example: If $f_{\text{visual}}(I)$ yields a probability distribution $P(O|I) = \{p(o_1|I), \ldots, p(o_K|I)\}$ over $K$ possible visual categories $o_k$, then $\text{Stat}(f_{\text{visual}}(I))$ can be: The entropy of this distribution: $\mathcal{H}(P(O|I)) = -\sum_{k=1}^{K} p(o_k|I) \log p(o_k|I)$. Or the complement of the highest probability: $1 - \max_k p(o_k|I)$. $\Psi(R)$: A function that adjusts or focuses the assessment of perceptual uncertainty based on the current agent's response $R$ (or task context). For example: $\Psi(R)$ might selectively weight certain types of perceptual uncertainty based on the content of $R$, or adjust the overall scale of uncertainty according to task importance. $f_{\text{perc}}(\cdot, \cdot)$: The final aggregation function that combines the quantified visual feature information from $\text{Stat}(\cdot)$ and contextual

**b. Semantic Uncertainty ($u_{\text{sem}}$)**  Semantic uncertainty ($u_{\text{sem}}$) reflects the ambiguity or multiple possibilities in understanding the meaning of a scene, interactions between objects, or symbolic

interpretations, assuming the visual signals have been perceived. It is defined as follows:

$$u_{\text{sem}}(R) = \frac{\sum_{i \in \text{SemTypes}} \overbrace{w_i}^{\substack{\text{Weight of} \\ \text{Semantic Type } i}} \cdot \overbrace{C_i(R)}^{\substack{\text{Ambiguity in Response } R \\ \text{regarding Semantic Type } i}}}{\underbrace{N(R)}_{\substack{\text{Normalization Factor for Complexity} \\ \text{or Number of Semantic Elements in } R}} + \underbrace{\lambda}_{\substack{\text{Smoothing} \\ \text{Constant}}}} \tag{8}$$

Detailed Expansion and Explanation: $R$: The response generated by the agent or its internal semantic representation. SemTypes: A predefined set of semantic types, e.g., {object attributes, spatial relationships, behavioral intentions, ...}. $w_i$: Importance weight assigned to semantic type $i$, typically $w_i \geq 0$ and $\sum w_i = 1$ (or other normalization methods). $C_i(R)$: A function that quantifies the ambiguity or complexity related to semantic type $i$ in response $R$. For example: If semantic type $i$ focuses on "inter-object relationships," and there are $M_{AB}$ possible valid relationships between object A and object B mentioned in response $R$, then $C_i(R)$ could be a function of $M_{AB}$ (e.g., $\log M_{AB}$), or the entropy of the probability distribution of these relationships. $N(R)$: A measure of the overall complexity of response $R$ (e.g., number of entities, propositions, or words contained), used as a normalization term in the denominator to obtain an average per-unit semantic ambiguity. $\lambda$: A small positive constant ($\lambda > 0$) to prevent division by zero.

**c. Inferential Uncertainty ($u_{\text{inf}}$)** Inferential uncertainty ($u_{\text{inf}}$) measures the agent's confidence in its predictions about future events, unknown states, or decision outcomes based on current information. It is defined as follows: $u_{\text{inf}}(R, S) = \gamma \cdot (1 - \overline{P}(S)) + (1 - \gamma) \cdot \overline{\mathcal{H}}(S)$ Detailed Expansion and Explanation: $R$: The current agent response or contextual information extracted from it. $S = \{s_1, s_2, \ldots, s_M\}$: A set of $M$ mutually exclusive potential outcomes of future events, states, or decisions to be predicted. $P(S|R, I)$: The predicted probability distribution over the outcomes in $S$, given current information. $\overline{P}(S) = \max_{s_j \in S} P(s_j|R, I)$: The probability value of the most likely predicted outcome in this distribution. The first term $\gamma\left(1 - \overline{P}(S)\right)$ thus quantifies the uncertainty arising from a lack of confidence in the "best guess." $\overline{\mathcal{H}}(S) = -\sum_{j=1}^{M} P(s_j|R, I) \log P(s_j|R, I)$: The Shannon entropy of this predictive probability distribution. The second term $(1 - \gamma)\overline{\mathcal{H}}(S)$ thus quantifies the uncertainty due to the dispersion or disorder of the overall prediction outcomes. $\gamma \in [0, 1]$: A hyperparameter that balances the relative importance of these two sources of uncertainty.

### C.1.2 MANAGEABILITY DIMENSIONS: EPISTEMIC UNCERTAINTY AND ALEATORIC UNCERTAINTY

Uncertainty is further divided into manageable epistemic uncertainty ($\mathbf{u}_{\text{epis}}$) and inherent aleatoric uncertainty ($\mathbf{u}_{\text{alea}}$), as introduced in Section 3.

**a. Epistemic Uncertainty ($\mathbf{u}_{\text{epis}}$)**

$$\mathbf{u}_{\text{epis}} = \underbrace{f'_{\text{base,epis}}(u_{\text{perc}}, u_{\text{sem}}, u_{\text{inf}})}_{\substack{\text{Mapping from Base Dimensions to Epistemic Uncertainty} \\ \text{(Reducible Part)}}} + \underbrace{f''_{\text{epis}}(\Lambda(R))}_{\substack{\text{Epistemic Uncertainty Directly Contributed by} \\ \text{Knowledge Gap Cues } \Lambda(R)}} \tag{9}$$

Detailed Expansion and Explanation: $f'_{\text{base,epis}}(\cdot, \cdot, \cdot)$: A function that aggregates those parts of perceptual, semantic, and inferential uncertainty considered "knowable" or "reducible" (through more information or better models). $\Lambda(R)$: Represents "Explicit Cues of Knowledge Gaps" related to response $R$. For example: $\Lambda(R)$ could quantify the deviation of the current query from the training data distribution (Out-of-Distribution detection), or the model's familiarity score with specific concepts in the query. $f''_{\text{epis}}(\cdot)$: A function that converts these knowledge gap cues into an additional amount of epistemic uncertainty.

**b. Aleatoric Uncertainty ($\mathbf{u}_{\text{alea}}$)**

$$\mathbf{u}_{\text{alea}} = \underbrace{f'_{\text{base,alea}}(u_{\text{perc}}, u_{\text{sem}})}_{\substack{\text{Mapping from Base Dimensions to Aleatoric Uncertainty} \\ \text{(Inherent Random Part, Primarily from Perception and Semantics)}}} + \underbrace{f''_{\text{alea}}(\Omega(R))}_{\substack{\text{Aleatoric Uncertainty Directly Contributed by} \\ \text{Environmental Randomness Signals } \Omega(R)}} \tag{10}$$

Detailed Expansion and Explanation: $f'_{\text{base,alea}}(\cdot, \cdot)$: A function that aggregates those parts of perceptual and semantic uncertainty considered "inherent" or "irreducible" (stemming from the randomness of the data itself or the intrinsic ambiguity of the task). $\Omega(R)$: Represents "Explicit Signals of Environmental Randomness" related to response $R$. For example: $\Omega(R)$ could be inherent randomness explicitly stated in the task description (e.g., "result of a dice roll"), or unpredictable disturbances perceived from the environment. $f''_{\text{alea}}(\cdot)$: A function that converts these environmental randomness signals into an additional amount of aleatoric uncertainty.

### C.1.3 TOTAL UNCERTAINTY ($\mathbf{u}_{\text{TOTAL}}$)

Finally, the uncertainties from the three base dimensions are weighted and fused to obtain the total uncertainty:

$$\mathbf{u}_{\text{total}} = \underbrace{w_{\text{perc}}}_{\text{Perceptual Weight}} u_{\text{perc}} + \underbrace{w_{\text{sem}}}_{\text{Semantic Weight}} u_{\text{sem}} + \underbrace{w_{\text{inf}}}_{\text{Inferential Weight}} u_{\text{inf}} \tag{11}$$

Detailed Expansion and Explanation: $w_{\text{perc}}, w_{\text{sem}}, w_{\text{inf}}$: Weights for the perceptual, semantic, and inferential uncertainty dimensions, respectively. These weights typically satisfy $w_k \geq 0$ and $\sum w_k = 1$ (or other normalization methods), reflecting the relative importance of different uncertainty dimensions in a specific task or system objective.

## C.2 DYNAMIC UNCERTAINTY TRANSFER MECHANISM (3.2)

To achieve active management of uncertainty and optimize system operational costs, Agora introduces a dynamic uncertainty transfer mechanism that explicitly tracks the flow of uncertainty among agents.

### C.2.1 UNCERTAINTY FLOW EQUATION

At any task $t$, the total uncertainty $\mathbf{U}(a_i, t)$ borne by agent $a_i$ consists of its self-generated base uncertainty and the transferred uncertainty received from other agents. This dynamic process is described by the following core equation (see Eq. 3):

$$\mathbf{U}(a_i, t) = \underbrace{\mathbf{U}_{\text{base}}(a_i, t)}_{\substack{\text{Base Uncertainty} \\ \text{(Generated by } a_i \text{ for task } t \text{ itself)}}} + \underbrace{\sum_{j \neq i} \mathbf{U}_{\text{transfer}}(a_j \to a_i, t)}_{\substack{\text{Transferred Uncertainty} \\ \text{(Sum received from other agents } a_j)}} \tag{12}$$

Detailed Expansion and Explanation: $\mathbf{U}(a_i, t)$: The total uncertainty vector-borne by agent $a_i$ at task $t$. This is a multi-dimensional vector where each dimension corresponds to a specific type of uncertainty (e.g., $u_{\text{perc}}, u_{\text{sem}}, u_{\text{inf}}$ defined earlier, or more fine-grained subtypes). $\mathbf{U}(a_i, t) = [u_{\text{perc}}(a_i, t), u_{\text{sem}}(a_i, t), u_{\text{inf}}(a_i, t), \ldots]^T$ $\mathbf{U}_{\text{base}}(a_i, t)$: The base uncertainty vector generated by agent $a_i$ due to its direct interaction with task $t$. Its calculation can depend on historical information, agent profiles, or default values. $\mathbf{U}_{\text{transfer}}(a_j \to a_i, t)$: The uncertainty vector successfully transferred from agent $a_j$ and received by agent $a_i$ in task $t$. $\sum_{j \neq i} \mathbf{U}_{\text{transfer}}(a_j \to a_i, t)$: Summation of uncertainty vectors transferred from all other agents $a_j$ ($j \neq i$) to $a_i$, yielding the total uncertainty received by $a_i$ via the transfer mechanism at task $t$.

### C.2.2 TREND OF CHANGE IN SYSTEM-TOTAL UNCERTAINTY: CONSERVATION/CONVERGENCE ANALYSIS

**a. Definition of System-Total Uncertainty** Let $\mathcal{A} = \{a_1, a_2, \ldots, a_N\}$ be the set of agents in the system. At time $t$, the **System-Total Uncertainty** $\mathbf{U}_{\text{sys}}(t)$ borne by all agents in the system can be defined as the sum of the total uncertainties of individual agents: $\mathbf{U}_{\text{sys}}(t) \triangleq \sum_{i=1}^{N} \mathbf{U}(a_i, t)$ Substituting the uncertainty flow equation:

$$\mathbf{U}_{\text{sys}}(t) = \sum_{i=1}^{N} \left( \mathbf{U}_{\text{base}}(a_i, t) + \sum_{j \neq i} \mathbf{U}_{\text{transfer}}(a_j \to a_i, t) \right) \tag{13}$$

$$\mathbf{U}_{\text{sys}}(t) = \underbrace{\sum_{i=1}^{N} \mathbf{U}_{\text{base}}(a_i, t)}_{\mathbf{U}_{\text{sys,base}}(t): \text{System total base uncertainty}} + \underbrace{\sum_{i=1}^{N} \sum_{j \neq i} \mathbf{U}_{\text{transfer}}(a_j \rightarrow a_i, t)}_{\mathbf{U}_{\text{sys,transfer\_received}}(t): \text{System total received uncertainty}} \quad (14)$$

**b. Impact of Uncertainty Transfer on Total Uncertainty** Consider a specific transfer event: at some stage of task $t$, agent $a_k$ successfully transfers an amount of uncertainty $T_{kl}(d)$ (in dimension $d$) to agent $a_l$. As per Eq. 4, the change in uncertainty for the sender and receiver after the transfer: Sender $a_k$'s uncertainty change: $\mathbf{U}'_k = \mathbf{U}_k - \kappa T_{kl}$, receiver $a_l$'s uncertainty change: $\mathbf{U}'_l = \mathbf{U}_l + (1 - \xi_l)T_{kl}$ Where: $T_{kl}$: The amount of uncertainty declared for transfer from $a_k$ to $a_l$. $\kappa \in [0, 1]$: Transfer Efficiency factor. $\kappa = 1$ means the declared amount is fully removed from the sender. $\xi_l \in [0, 1]$: Receiver $a_l$'s Expertise/Resolution Factor. $\xi_l > 0$ means the receiver, due to its expertise, effectively bears or perceives an incremental uncertainty less than the declared transfer amount, i.e., part of the uncertainty is "resolved" or "absorbed." A successful transfer $T_{kl}$ from $a_k$ to $a_l$ leads to a change in system total uncertainty $\Delta\mathbf{U}_{\text{sys}}$: Assume this transfer is the only change in the system, and other agents' uncertainties remain constant.

$$\Delta\mathbf{U}_{\text{sys}} = (\mathbf{U}'_k + \mathbf{U}'_l + \sum_{m \neq k,l} \mathbf{U}_m) - (\mathbf{U}_k + \mathbf{U}_l + \sum_{m \neq k,l} \mathbf{U}_m) \quad (15)$$

$$\Delta\mathbf{U}_{\text{sys}} = (\mathbf{U}'_k - \mathbf{U}_k) + (\mathbf{U}'_l - \mathbf{U}_l) \quad (16)$$

$$\Delta\mathbf{U}_{\text{sys}} = (-\kappa T_{kl}) + ((1 - \xi_l)T_{kl}) \quad (17)$$

$$\Delta\mathbf{U}_{\text{sys}} = \underbrace{(1 - \xi_l - \kappa)}_{\text{Net change factor in system uncertainty due to a single trade}} T_{kl} \quad (18)$$

**Analysis**: Conservation: Strict Conservation: $\Delta\mathbf{U}_{\text{sys}} = 0$ when $\xi_l + \kappa = 1$. If $\kappa = 1$ (fully removed), then $\xi_l = 0$ (fully borne) is needed - uncertainty merely redistributes. Generally Non-conserved: Typically $\kappa \approx 1$ and $\xi_l > 0$, so $\Delta\mathbf{U}_{\text{sys}} = -\xi_l T_{kl}$. When uncertainty transfers to agents with expertise ($\xi_l > 0$), total system uncertainty decreases, not physical disappearance but effective resolution by more suitable agents. 2. Convergence: Total Uncertainty Amount: With continuous $\mathbf{U}_{\text{base}}(a_i, t)$ and transfers to skilled agents ($\xi_l > 0$), the system reaches a dynamic equilibrium where new uncertainty balances resolved uncertainty. If $\mathbf{U}_{\text{base}}(a_i, t) \rightarrow 0$ and transfers continue, $\mathbf{U}_{\text{sys}}(t)$ decreases, potentially to zero if all uncertainty is resolvable. Specific State: System state convergence depends on trading protocols and cost optimization. If each trade reduces cost, the system reaches the local optimum with stable uncertainty distribution - an "equilibrium" state where uncertainty continues being processed dynamically. **Conclusion**: The dynamic uncertainty transfer mechanism, especially when considering the receiver's expertise factor $\xi_l > 0$, has the **potential to reduce the system's effective total uncertainty**. The absolute convergence of the system's total uncertainty depends on the rate of base uncertainty generation and the continued effectiveness of the trading mechanism. The convergence of uncertainty distribution among agents is closely related to the trading equilibrium state driven by cost optimization.

### C.2.3 DEEPENING THE TRANSFER COST-BENEFIT ANALYSIS: CONSIDERING TOTAL TRANSFER AMOUNT AND EXPERT KNOWLEDGE

In the dynamic uncertainty transfer mechanism, a key decision criterion is whether a trade can reduce the cost of handling uncertainty at the system level. This depends not only on the comparison of unit costs but also on the actual total amount of uncertainty transferred and the receiver's expertise in handling that uncertainty.

**a. Variable Processing Cost Change of a Trade** Consider a transfer of uncertainty in a specific dimension $d$ from agent $a_i$ (sender) to agent $a_j$ (receiver). Let $U_i(d)$ and $U_j(d)$ be the uncertainty stock of $a_i$ and $a_j$ in dimension $d$ before the trade, respectively. * Let $c_i$ and $c_j$ be the marginal processing costs for $a_i$ and $a_j$ to handle a unit of uncertainty in dimension $d$, respectively. * Let $T_{ij}(d)$ be the total amount of uncertainty declared for transfer from $a_i$ to $a_j$ in dimension $d$. $T_{ij}(d) > 0$. * Let $\xi_j \in [0, 1]$ be the expertise factor of receiver $a_j$ when processing uncertainty of dimension $d$. $(1 - \xi_j)T_{ij}(d)$ represents the effective increase in uncertainty borne by $a_j$. If $\xi_j > 0$, a part of

the uncertainty is "resolved" or efficiently processed by $a_j$'s expertise. **Before the trade**, the total processing cost related to $U_i(d)$ and $U_j(d)$ (considering only these stock parts) is:

$$\mathcal{C}_{\text{before}} = \underbrace{c_i U_i(d)}_{\text{Cost of Agent } i} + \underbrace{c_j U_j(d)}_{\text{Cost of Agent } j} \tag{19}$$

**After the trade**, agent $a_i$'s uncertainty stock becomes $U_i(d) - T_{ij}(d)$. Agent $a_j$'s uncertainty stock effectively increases by $(1 - \xi_j)T_{ij}(d)$, becoming $U_j(d) + (1 - \xi_j)T_{ij}(d)$. The new total processing cost related to this is:

$$\mathcal{C}_{\text{after}} = \underbrace{c_i(U_i(d) - T_{ij}(d))}_{\text{New cost of Agent } i} + \underbrace{c_j(U_j(d) + (1 - \xi_j)T_{ij}(d))}_{\text{New cost of Agent } j} \tag{20}$$

**b. Deriving the Cost-Benefit Condition for a Trade**   A trade is beneficial in terms of processing costs if and only if the total processing cost after the trade is strictly less than the total processing cost before the trade, i.e., $\mathcal{C}_{\text{after}} < \mathcal{C}_{\text{before}}$. (This is equivalent to Eq. 4 which states $\Delta\mathcal{C} < 0$).

$$\underbrace{c_i U_i(d) + c_j U_j(d)}_{\text{Total processing cost before trade (LHS)}} > \underbrace{c_i(U_i(d) - T_{ij}(d)) + c_j(U_j(d) + (1 - \xi_j)T_{ij}(d))}_{\text{Total processing cost after trade (RHS)}} \tag{21}$$

**Formal Expansion and Proof** :

1. Subtract common terms $c_i U_i(d)$ and $c_j U_j(d)$ from both sides of the inequality:

$$0 > -c_i T_{ij}(d) + c_j(1 - \xi_j)T_{ij}(d) \tag{22}$$

2. Rearrange terms to centralize those containing $T_{ij}(d)$:

$$c_i T_{ij}(d) - c_j(1 - \xi_j)T_{ij}(d) > 0 \tag{23}$$

3. Factor out $T_{ij}(d)$ (by definition, the actual transferred amount $T_{ij}(d) > 0$):

$$\underbrace{T_{ij}(d)}_{>0} \cdot (c_i - c_j(1 - \xi_j)) > 0 \tag{24}$$

4. Since $T_{ij}(d) > 0$, the necessary and sufficient condition for the above inequality to hold is:

$$\boxed{c_i > c_j(1 - \xi_j)} \tag{25}$$

$$\underbrace{c_i}_{\substack{\text{Sender } a_i \\ \text{unit cost}}} > \underbrace{c_j(1 - \xi_j)}_{\substack{\text{Receiver } a_j \text{ effective unit cost} \\ \text{(considering expertise } \xi_j)}} \tag{26}$$

**Theoretical Significance:** This condition explicitly states that only when the sender's unit processing cost is higher than the receiver's effective unit processing cost can the trade yield benefits at the variable processing cost level.

**c. Connection with CE Ratio and Broader Cost Considerations**   The naive CE ratio defined as $CE(a_i \rightarrow a_j, u) = \frac{c_i(u)}{c_j(u)}$ (here using $i$ as the sender, $j$ as a receiver, consistent with current notation) suggests that if $c_i(u) > c_j(u)$, the transfer is beneficial. Now, incorporating the receiver's expertise factor $\xi_j$, we can define an **Effective Cost-Effectiveness Ratio** ($CE'_{i \rightarrow j}$):

$$CE'_{i \rightarrow j}(d) \triangleq \frac{\overbrace{c_i}^{\text{Sender's unit cost}}}{\underbrace{c_j(1 - \xi_j)}_{\text{Receiver's effective unit cost}}} \tag{27}$$

Then, the derived cost-benefit condition $c_i > c_j(1 - \xi_j)$ is equivalent to:

$$CE'_{i \rightarrow j}(d) > 1 \tag{28}$$

This indicates that a transfer is beneficial in terms of direct processing costs only when the sender's unit processing cost is higher relative to the receiver's "effective" unit cost. **Notes on Fixed Costs**

**and Transaction Costs**: The above derivation primarily focuses on the reduction of **variable costs** directly related to the amount of uncertainty processed. A complete trading decision also needs to consider more comprehensive cost-benefits: Fixed Costs ($\beta_i, \beta_j$): If a trade causes an agent to change from inactive to active (incurring a new $\beta_j$), or from active to inactive (saving $\beta_i$), these changes in fixed costs need to be included in the calculation of the total benefit. 2. Transaction Costs: Communication and computation overheads that may exist for executing the trade itself. Therefore, the condition $c_i > c_j(1 - \xi_j)$ is a core element for judging whether a trade can potentially reduce variable processing costs, but the final decision to execute the trade must be made through a more comprehensive benefit evaluation (which should internalize all relevant cost and benefit items).

## C.3 UNCERTAINTY TRADING PROTOCOL

This protocol defines the rules and conditions for agents to trade uncertainty, aiming to transform uncertainty into a manageable and optimizable resource to reduce total system operating costs. The core of trading is transferable epistemic uncertainty ($\mathbf{u}_{\text{epis}}$), conducted based on principles of comparative advantage and cost-effectiveness, as per Eq. 5.

### C.3.1 PREREQUISITES FOR A TRADE

A potential trade to transfer uncertainty of dimension $d \in \mathcal{D}_{\text{tradable}}$ (set of tradable uncertainty dimensions) from agent $a_i$ (sender) to $a_j$ (receiver) must first satisfy the following conditions:

**a. Trade Trigger Condition** To ensure the necessity of trade and avoid ineffective fluctuations, an uncertainty differential threshold is set:

$$\exists d \in \mathcal{D}_{\text{tradable}} \quad \text{s.t.} \quad \underbrace{U_i(d)}_{\substack{\text{Sender } a_i\text{'s current} \\ \text{uncertainty level in dim } d}} - \underbrace{U_j(d)}_{\substack{\text{Receiver } a_j\text{'s current} \\ \text{uncertainty level in dim } d}} > \underbrace{\tau_{\text{trade}}}_{\substack{\text{Minimum uncertainty} \\ \text{differential threshold} \\ \text{to trigger trade}}} \tag{29}$$

Theoretical Significance: $\tau_{\text{trade}} > 0$ ensures that a trade intention is initiated only when there is a significant imbalance in uncertainty distribution, sufficient to overcome potential transaction friction costs and form an effective comparative advantage.

**b. Receiver Capacity Constraint** The planned amount of uncertainty to be transferred $T_{ij}(d)$ must not exceed the processing capacity of the receiver $a_j$. Considering the receiver's expertise factor $\xi_j$, the effective increase is $(1 - \xi_j)T_{ij}(d)$:

$$\forall d \in \mathcal{D}_{\text{tradable}}, \quad \underbrace{U_j(d)}_{\substack{\text{Receiver } a_j\text{'s pre-trade} \\ \text{uncertainty in dim } d}} + \underbrace{(1 - \xi_j)T_{ij}(d)}_{\substack{\text{Effective uncertainty} \\ \text{increment in dim } d}} \leq \underbrace{C_j(d)}_{\substack{\text{Receiver } a_j\text{'s uncertainty} \\ \text{capacity limit in dim } d}} \tag{30}$$

Theoretical Significance: This constraint prevents the receiver from being overloaded by taking on too much uncertainty, ensuring its own task-processing capability and system stability.

### C.3.2 COST-BENEFIT ANALYSIS OF A TRADE

**a. Condition for Reducing Variable Processing Costs** A trade must at least show an advantage in directly related variable processing costs. Consider the transfer of uncertainty $T_{ij}(d)$ in dimension $d$ from $a_i$ to $a_j$: Before the trade, the local processing cost related to $U_i(d)$ and $U_j(d)$ is:

$$\mathcal{C}_{\text{proc, pre}} = \underbrace{c_i U_i(d)}_{\text{Cost: } a_i \text{ processes } U_i(d)} + \underbrace{c_j U_j(d)}_{\text{Cost: } a_j \text{ processes } U_j(d)} \tag{31}$$

After the trade, the relevant new local processing cost :

$$\mathcal{C}_{\text{proc, post}} = \underbrace{c_i(U_i(d) - T_{ij}(d))}_{\text{Cost: } a_i \text{ processes remainder}} + \underbrace{c_j(U_j(d) + (1 - \xi_j)T_{ij}(d))}_{\text{Cost: } a_j \text{ processes total (incl. effective new)}} \tag{32}$$

where $c_i, c_j$ are the marginal costs for $a_i, a_j$ to process unit uncertainty in dimension $d$, and $\xi_j$ is $a_j$'s expertise factor. The condition for the trade to reduce variable processing costs is $\mathcal{C}_{\text{proc, pre}} > \mathcal{C}_{\text{proc, post}}$:

$$\underbrace{c_i U_i(d) + c_j U_j(d)}_{\text{LHS}} > \underbrace{c_i(U_i(d) - T_{ij}(d)) + c_j(U_j(d) + (1 - \xi_j)T_{ij}(d))}_{\text{RHS (expanded)}} \tag{33}$$

**Formal Expansion and Proof** :

1. Subtract common terms $c_i U_i(d)$ and $c_j U_j(d)$ from both sides of the inequality:
$$0 > -c_i T_{ij}(d) + c_j(1 - \xi_j)T_{ij}(d) \tag{34}$$

2. Rearrange terms to centralize those containing $T_{ij}(d)$:
$$c_i T_{ij}(d) - c_j(1 - \xi_j)T_{ij}(d) > 0 \tag{35}$$

3. Factor out $T_{ij}(d)$ (by definition, the actual transferred amount $T_{ij}(d) > 0$):
$$\underbrace{T_{ij}(d)}_{>0} \cdot (c_i - c_j(1 - \xi_j)) > 0 \tag{36}$$

4. Since $T_{ij}(d) > 0$, the necessary and sufficient condition for the above inequality to hold is:
$$\boxed{c_i > c_j(1 - \xi_j)} \tag{37}$$
$$\underbrace{c_i}_{\substack{\text{Sender } a_i \\ \text{unit cost}}} > \underbrace{c_j(1 - \xi_j)}_{\substack{\text{Receiver } a_j \text{ effective unit cost} \\ \text{(considering expertise } \xi_j)}} \tag{38}$$

**Theoretical Significance:** This condition explicitly states that only when the sender's unit processing cost is higher than the receiver's effective unit processing cost can the trade yield benefits at the variable processing cost level.

**b. Overall Expected Benefit Condition** The final decision to execute a trade depends on whether its overall expected benefit exceeds a threshold $\tau_{\text{benefit}}$ (as in Eq. 5):
$$\Delta \mathcal{C}(T_{ij}(t)) < 0 \wedge (U_j(t) + T_{ij}(t) \leq C_j(t)) \tag{39}$$
Formal Expansion and Explanation: $U_{\text{pre}}(i,j), C_{\text{pre}}(i,j)$: Measure of uncertainty and costs for agents $a_i, a_j$ before the trade. $U_{\text{pre}}(i,j)$ could be $|(U_i \ U_j)|_{\text{agg}}$, a norm or weighted sum. $C_{\text{pre}}(i,j)$ might be unit cost or total cost estimate. $U_{\text{post}}(i,j,T_{ij}), C_{\text{post}}(i,j,T_{ij})$: Measures after trade. Post-trade vectors: $U_i' = U_i - \kappa T_{ij}, U_j' = U_j + (1 - \xi_j)T_{ij}$, where $\kappa$ is transfer efficiency. $U_{\text{post}}(i,j,T_{ij})$ could be $|(w_i U_i' \ w_j U_j')|_{\text{agg}}$, with weights $w_i, w_j$. $\tau_{\text{benefit}} \geq 0$: Ensures significant trade benefit, covering implicit transaction costs/risks. * Theoretical Significance: Comprehensive trade evaluation ensuring not just marginal cost benefits but system-wide favorability after considering total uncertainty changes, fixed cost impacts, and return requirements.

### C.3.3 MARKET EQUILIBRIUM ANALYSIS - BRIEF THEORETICAL PERSPECTIVE

Market equilibrium refers to a state where no potential trades satisfying all trading conditions (trigger, capacity, cost-benefit, overall expected benefit) exist in the system, leading to a relatively stable distribution of uncertainty. **Definition 3.3.1 (Local Equilibrium State).** The system reaches a local equilibrium if, for any pair of agents $(a_i, a_j)$ and any tradable dimension $d \in \mathcal{D}_{\text{tradable}}$, at least one of the following does not hold:

1. $U_i(d) - U_j(d) > \tau_{\text{trade}}$
2. $U_j(d) + (1 - \xi_j)T_{ij}(d) \leq C_j(d)$ (for some permissible $T_{ij}(d) > 0$)
3. $c_i > c_j(1 - \xi_j)$
4. $\Delta \mathcal{C}(T_{ij}) < 0$ (for some $T_{ij}(d)$ determined by 1-3)

**Proposition 3.3.1 (Convergence of Trading Process to Local Equilibrium).** If: (A1) The total tradable uncertainty in the system is finite, or the volume of a single trade $T_{ij}(d)$ has a positive lower bound. (A2) Each successful trade strictly reduces the global cost function $\mathcal{C}_{\text{sys}}$ by an amount greater than $\delta_{\min} > 0$. (A3) $\mathcal{C}_{\text{sys}}$ is bounded below. Then the sequence of trades is finite, and the system will converge to a local equilibrium state as defined above. Proof Outline: $\mathcal{C}_{\text{sys}}^{(k+1)} \leq \mathcal{C}_{\text{sys}}^{(k)} - \delta_{\min}$ (each trade reduces cost). Since $\mathcal{C}_{\text{sys}}$ has a lower bound $\mathcal{C}_{\min}$, and the initial cost is $\mathcal{C}_{\text{sys}}^{(0)}$, the maximum number of trades $N_{\max_{\text{trades}}} \leq (\mathcal{C}_{\text{sys}}^{(0)} - \mathcal{C}_{\min})/\delta_{\min}$, hence the trade sequence is finite. When the sequence terminates, no more trades satisfy all conditions, and the system reaches local equilibrium. This equilibrium state represents a point where, under the current protocol and information, system costs cannot be further optimized through bilateral trades.

### C.3.4 APPLICATION OF COMPARATIVE ADVANTAGE THEORY

The theory of comparative advantage provides a theoretical basis for uncertainty trading: even if some agents do not possess an absolute cost advantage in processing all uncertainty dimensions, as long as there are differences in the relative processing efficiencies (opportunity costs) of various agents across different dimensions, specialization, and trade can still enhance overall system efficiency and reduce total costs. **Definition 3.4.1 (Comparative Advantage).** For agents $a_k, a_l$ and uncertainty dimensions $d_1, d_2$, if their unit processing costs $c_x(d_y)$:

$$\frac{c_k(d_1)}{c_k(d_2)} < \frac{c_l(d_1)}{c_l(d_2)} \tag{40}$$

then $a_k$ has a comparative advantage over $a_l$ in processing $d_1$ (relative to $d_2$).

**a. Comparative Advantage and Cost Optimization**  Trades based on comparative advantage aim to allocate specific types of uncertainty to the agent with the lowest opportunity cost for that type.

$$\Delta\mathcal{C}_{\text{total}} = [c_i(U_i - \Delta U_{d_1}) + c_j(U_j + \Delta U_{d_1})] - [c_i(U_i) + c_j(U_j)] \tag{41}$$

Expanding this:

$$\Delta\mathcal{C}_{\text{total}} = (\alpha_j - \alpha_i) \cdot \Delta U_{d_1} \tag{42}$$

For $\Delta\mathcal{C}_{\text{total}} < 0$, it is required that $\alpha_j < \alpha_i$. That is, uncertainty $d_1$ should flow from an agent with a higher unit processing cost ($a_i$) to one with a lower unit processing cost ($a_j$).

**b. Implicit Implementation of Comparative Advantage by Agora Protocol**  The core trading condition of Agora, $c_i > c_j(1 - \xi_j)$, is based on **effective absolute cost advantage**. However, if the cost parameters $c_k$ and expertise $\xi_k$ of agents dynamically reflect their true efficiency and specialization in handling different dimensions of uncertainty (which may stem from their comparative advantages), then a series of local trades based on effective absolute cost advantage will, at a macro level, guide the system's uncertainty distribution towards a configuration that aligns with the principles of comparative advantage. For instance, an agent with a comparative advantage in dimension $d_1$ might develop a low $c(d_1)$ and high $\xi(d_1)$ for processing $d_1$, thereby becoming a natural "sink" for uncertainty in that dimension.

### C.4 UNCERTAINTY-AWARE MAB SELECTION STRATEGY

**a. Beta Posterior Parameter Update**  For all $S \in \mathcal{A}$ (set of agents), at decision round $t$, the Beta distribution parameters $(\alpha_S^{(t)}, \beta_S^{(t)})$ are updated: Let $r_S^{(t-1)} \in \{0, 1\}$ be the observed binary reward for agent $S$ in round $t - 1$.

$$\alpha_S^{(t)} := \underbrace{\alpha_S^{(0)}}_{\text{Prior } \alpha_0} + \sum_{\tau=0}^{t-1} \mathbb{I}(S^{(\tau)} = S) \cdot r_S^{(\tau)} \tag{43}$$

$$\beta_S^{(t)} := \underbrace{\beta_S^{(0)}}_{\text{Prior } \beta_0} + \sum_{\tau=0}^{t-1} \mathbb{I}(S^{(\tau)} = S) \cdot (1 - r_S^{(\tau)}) \tag{44}$$

where $\mathbb{I}(S^{(\tau)} = S)$ is an indicator function, indicating whether agent $S$ was selected in round $\tau$. Typically, $\alpha_S^{(0)} = 1, \beta_S^{(0)} = 1$.

**b. Baseline Expected Reward**  $\mathbb{E}[\theta_S^{(t)}] = \frac{\alpha_S^{(t)}}{\alpha_S^{(t)} + \beta_S^{(t)}}$

**c. Comprehensive Scoring Function** $\tilde{\theta}_S^{(t)}$

$$\tilde{\theta}_S^{(t)} \triangleq \underbrace{\left(\mathbb{E}[\text{Reward}_S^{(t)}] - \text{Cost}_S^{(t)}\right)}_{\text{Expected Net Reward (ENR}_S^{(t)})} \cdot \underbrace{f_{TM}(S, t; \lambda_{\text{dist}})}_{\text{Task Matching Factor (TMF}_S^{(t)})} \cdot \underbrace{f_{TD}(\Delta t_S; \gamma_{\text{decay}})}_{\text{Time Decay Factor (TDF}_S^{(t)})} \cdot \underbrace{f_{Syn}(S; \eta)}_{\text{Team Synergy Factor (TSF}_S)} \cdot \underbrace{f_{Strat}(S; \omega)}_{\text{Strategic Uncertainty Index Factor (SUIF}_S)} \tag{45}$$

**Expected Net Reward (ENR)**: $(\mathbb{E}[\text{Reward}_S^{(t)}] - \text{Cost}_S^{(t)})$ This term represents the fundamental utility of selecting agent $S$ for task $t$, balancing its expected rewards against its operational costs;

$\mathbb{E}[\text{Reward}_S^{(t)}] = \mathbb{E}[\theta_S^{(t)}] \cdot R_{\max}(t)$: This is the anticipated raw reward from agent $S$ for task $t$; $\mathbb{E}[\theta_S^{(t)}]$: The posterior mean of the success probability for agent $S$, typically derived from a Beta distribution $\text{Beta}(\alpha_S^{(t)}, \beta_S^{(t)})$, calculated as $\frac{\alpha_S^{(t)}}{\alpha_S^{(t)} + \beta_S^{(t)}}$; $R_{\max}(t)$: The maximum possible reward achievable for task $t$; $\text{Cost}_S^{(t)}(U_{S,\text{est}}^{(t)}, \text{TaskFeat}_t) = \alpha_{\text{cost},S} \cdot U_{S,\text{est}}^{(t)} + \beta_{\text{cost},S} + C_{\text{task}}(t, \text{TaskFeat}_t)$: The estimated cost for agent $S$ to handle task $t$; $\alpha_{\text{cost},S}$: The marginal cost for agent $S$ to process one unit of uncertainty; $U_{S,\text{est}}^{(t)} = ||\mathbf{U}_{S,\text{base}}^{(t)} + \mathbf{U}_{S,\text{transfer\_in}}^{(t)} - \mathbf{U}_{S,\text{transfer\_out}}^{(t)}||_1$: The estimated total uncertainty agent $S$ handles for task $t$ (using the $L_1$ norm). This considers its self-generated base uncertainty ($\mathbf{U}_{S,\text{base}}^{(t)}$), uncertainty received from other agents ($\mathbf{U}_{S,\text{transfer\_in}}^{(t)}$), and uncertainty offloaded to others ($\mathbf{U}_{S,\text{transfer\_out}}^{(t)}$); $\beta_{\text{cost},S}$: The fixed base operational cost for agent $S$; $C_{\text{task}}(t, \text{TaskFeat}_t)$: Additional costs incurred due to specific features of task $t$ ($\text{TaskFeat}_t$). **Task Matching Factor**

**(TM)**: $f_{\text{TM}}(S, t; \lambda_{\text{dist}}) = \exp(-\lambda_{\text{dist}} \cdot d_{S,t})$ This factor quantifies the compatibility or relevance of agent $S$ to the current task $t$. A higher match (smaller distance $d_{S,t}$) results in a factor closer to 1; $\lambda_{\text{dist}}$: A hyperparameter that weights the influence of the distance $d_{S,t}$; $d_{S,t}$: The distance or dissimilarity between the feature vector of agent $S$ ($\mathbf{v}_S$) and that of task $t$ ($\mathbf{v}_t$). Two alternative calculations are suggested: Normalized Euclidean distance: $d_{S,t} = \frac{||\mathbf{v}_S - \mathbf{v}_t||_2}{\max_{S' \in \mathcal{A}} ||\mathbf{v}_{S'} - \mathbf{v}_t||_2 + \epsilon}$. Normalization is done by dividing by the maximum distance found among all agents for that task, with $\epsilon$ being a small constant to prevent division by zero; Cosine dissimilarity: $1 - \frac{\mathbf{v}_S \cdot \mathbf{v}_t}{||\mathbf{v}_S||_2 ||\mathbf{v}_t||_2}$. This measures the difference in orientation between the two vectors. **Time Decay Factor (TDF)**:

$f_{\text{TD}}(\Delta t_S; \gamma_{\text{decay}}) = \gamma_{\text{decay}}^{\Delta t_S}$ This factor prioritizes more recent information regarding agent $S$'s performance or state, diminishing the impact of older data; $\gamma_{\text{decay}}$: The decay base hyperparameter, where $0 < \gamma_{\text{decay}} \leq 1$. If $\gamma_{\text{decay}} < 1$, older information receives a lower weight; $\Delta t_S = t - t_{\text{last\_update}}(S)$: The time elapsed since agent $S$'s parameters (e.g., Beta distribution parameters) were last updated. **Team Synergy Factor (TSF)**: $f_{\text{Syn}}(S; \eta) = (1 + \text{SynVal}_S^{(t)})^\eta$ This factor assesses the potential

for agent $S$ to collaborate effectively with other agents in the current team or context for task $t$; $\text{SynVal}_S^{(t)} = \frac{1}{|\text{Team}^{(t)}| - 1} \sum_{j \in \text{Team}^{(t)}, j \neq S} \text{Comp}(S, j) \cdot \text{Pot}(j, \text{Task}^{(t)})$: The synergy value for agent $S$ at time $t$. It's an average of compatibility scores ($\text{Comp}(S, j)$) between agent $S$ and its teammates $j$, weighted by each teammate's potential ($\text{Pot}(j, \text{Task}^{(t)})$) for the current task. $|\text{Team}^{(t)}|$ is the number of agents in the current team; $\eta$: A hyperparameter exponent that controls the degree of influence of the team synergy value. **Strategic Uncertainty Index Factor (SUIF)**: $f_{\text{Strat}}(S; \omega) = (1 + U_{\text{strat},S}^{(t)})^\omega$

This novel factor incorporates the strategic value of agent $S$'s uncertainty within the uncertainty trading market. Agents that can contribute more to system-level cost savings via uncertainty trading are favored; $U_{\text{strat},S}^{(t)}$: The strategic uncertainty value of agent $S$ at time $t$. It quantifies the expected net cost saving that agent $S$ can bring to the system by participating in the uncertainty market (as a seller or buyer). This is detailed further in Appendix C.4.2; $\omega$: A hyperparameter exponent that modulates the importance of this strategic uncertainty value in the overall score.

### C.4.1 THEORETICAL GUARANTEES: REGRET & CONVERGENCE

**a. Redefining Regret** Let $\tilde{\theta}_{S,true}^{(t)}(\mathcal{C}^{(t)})$ be the true expected comprehensive score of agent $S$ at time $t$ given context $\mathcal{C}^{(t)}$ (including task characteristics, market state, etc.). Let $S_{opt}^{(t)}(\mathcal{C}^{(t)}) = \arg\max_{S \in \mathcal{A}} \tilde{\theta}_{S,true}^{(t)}(\mathcal{C}^{(t)})$. The context-cumulative regret $R_T^{\tilde{\theta}}$ over $T$ time steps is:

$$R_T^{\tilde{\theta}} \triangleq \sum_{t=1}^{T} \mathbb{E}_{\mathcal{C}^{(t)}} \left[ \tilde{\theta}_{S_{opt}^{(t)}(\mathcal{C}^{(t)}),true}^{(t)}(\mathcal{C}^{(t)}) - \tilde{\theta}_{S^{(t)},true}^{(t)}(\mathcal{C}^{(t)}) \right] \tag{46}$$

where $S^{(t)}$ is the agent actually selected at time $t$ (context $\mathcal{C}^{(t)}$).

**b. Assumptions for Convergence Analysis** (A1) Boundedness: For all $S, t$, the values of $\tilde{\theta}_S^{(t)}$ (and its components) are within a bounded interval, e.g., $[0, \Theta_{\max}]$. (A2) Lipschitz Continuity (some fac-

tors): For changes in some contextual variables $c \in \mathcal{C}^{(t)}$, the change in $\tilde{\theta}_S^{(t)}$ is Lipschitz continuous, i.e., $|\tilde{\theta}_S^{(t)}(c_1) - \tilde{\theta}_S^{(t)}(c_2)| \leq L|c_1 - c_2|$. (A3) Learning and Adaptation: The agent's estimate of $\mathbb{E}[\theta_S^{(t)}]$ converges, and its estimates of dynamically changing contextual factors (like $\text{Cost}_S^{(t)}$, $U_{\text{strategic}}(S)$) are also progressively adapting.

**c. Direction of Convergence**   Although proving classic $O(\log T)$ or $O(\sqrt{T})$ regret bounds is very difficult, the strategy is designed such that the selection probability $P(S^{(t)} = S|\text{History}^{(t-1)}, \mathcal{C}^{(t)})$ gradually biases towards agents with higher true expected $\tilde{\theta}_{S,true}^{(t)}(\mathcal{C}^{(t)})$. If $\mathbb{E}[\tilde{\theta}_S^{(t)}(\mathcal{C}^{(t)})]$ itself converges to a stationary value $\tilde{\theta}_{S,true}^*(\mathcal{C}^*)$ (under a stationary context $\mathcal{C}^*$), then the selection will converge to the optimal arm $S_{opt}^* = \arg\max_S \tilde{\theta}_{S,true}^*(\mathcal{C}^*)$. If the context is non-stationary, the strategy attempts to track the optimal arm, similar to a multi-armed bandit problem in a non-stationary environment. Its performance depends on the speed and predictability of contextual changes, as well as the accuracy and adaptation speed of the factor estimates.

### C.4.2  MATHEMATICAL DECONSTRUCTION OF STRATEGIC UNCERTAINTY INDEX ($U_{\text{STRATEGIC}}(S)$)

**a. Core Objective Function of $U_{\text{strategic}}(S)$**   : Let $\mathcal{M}^{(t)}$ be the uncertainty market state at time $t$. $U_{\text{strategic}}(S, \mathcal{M}^{(t)})$ represents the expected **net cost saving** $\mathbb{E}[\Delta\mathcal{C}_{\text{sys}}(S, \mathcal{M}^{(t)})]$ that agent $S$ can bring to the entire system by participating in the market defined by $\mathcal{M}^{(t)}$.

$$U_{\text{strategic}}(S, \mathcal{M}^{(t)}) \triangleq \mathbb{E}_{\text{Trades involving S}} \left[ \sum_{\text{tr} \in \mathcal{T}(S, \mathcal{M}^{(t)})} (\mathcal{C}_{\text{sys}}(\text{pre-tr}) - \mathcal{C}_{\text{sys}}(\text{post-tr})) \cdot P(\text{tr occurs}) \right] \quad (47)$$

where $\mathcal{T}(S, \mathcal{M}^{(t)})$ is the set of all potential trades involving $S$ (as buyer or seller) that satisfy the trading conditions.

**b. Expansion of System Cost Change from a Trade $\Delta\mathcal{C}_{\text{sys}}(\text{trade})$**   : Consider a trade $tr = (s, r, k, T_{srk})$ transferring an amount $T_{srk}$ of uncertainty in dimension $d_k$ from $a_s$ to $a_r$.

$$\mathcal{C}_{\text{sys}}(\text{pre-tr}) = \sum_{i \in \mathcal{A}} \left( \sum_{j=1}^{M} \alpha_{ij} U_{ij}^{(\text{pre})} + \beta_i'(\mathbf{U}_i^{(\text{pre})}) \right) \quad (48)$$

$$\mathcal{C}_{\text{sys}}(\text{post-tr}) = \sum_{i \in \mathcal{A}, i \neq s, i \neq r} C_i(\mathbf{U}_i^{(\text{pre})}) + C_s(\mathbf{U}_s^{(\text{pre})} - \mathbf{e}_k T_{srk}) + C_r(\mathbf{U}_r^{(\text{pre})} + \mathbf{e}_k(1 - \xi_{rk})T_{srk}) \quad (49)$$

$$
\begin{aligned}
\Delta\mathcal{C}_{\text{sys}}(tr) &= \mathcal{C}_{\text{sys}}(\text{pre-tr}) - \mathcal{C}_{\text{sys}}(\text{post-tr}) \\
&= \left[ \alpha_{sk} U_{sk}^{(\text{pre})} + \beta_s'(\mathbf{U}_s^{(\text{pre})}) \right] + \left[ \alpha_{rk} U_{rk}^{(\text{pre})} + \beta_r'(\mathbf{U}_r^{(\text{pre})}) \right] \\
&\quad - \left[ \alpha_{sk}(U_{sk}^{(\text{pre})} - T_{srk}) + \beta_s'(\mathbf{U}_s^{(\text{pre})} - \mathbf{e}_k T_{srk}) \right] \\
&\quad - \left[ \alpha_{rk}(U_{rk}^{(\text{pre})} + (1 - \xi_{rk})T_{srk}) + \beta_r'(\mathbf{U}_r^{(\text{pre})} + \mathbf{e}_k(1 - \xi_{rk})T_{srk}) \right] \\
&= \underbrace{\alpha_{sk}T_{srk} - \alpha_{rk}(1 - \xi_{rk})T_{srk}}_{\text{Variable cost saving } \Delta C_{var}} \\
&\quad + \underbrace{\left( \beta_s'(\mathbf{U}_s^{(\text{pre})}) - \beta_s'(\mathbf{U}_s^{(\text{pre})} - \mathbf{e}_k T_{srk}) \right)}_{\text{Sender fixed cost change } \Delta\beta_s'} \\
&\quad + \underbrace{\left( \beta_r'(\mathbf{U}_r^{(\text{pre})}) - \beta_r'(\mathbf{U}_r^{(\text{pre})} + \mathbf{e}_k(1 - \xi_{rk})T_{srk}) \right)}_{\text{Receiver fixed cost change } \Delta\beta_r'} \quad (50)
\end{aligned}
$$

$$\Delta\mathcal{C}_{\text{sys}}(tr) = T_{srk}(\alpha_{sk} - \alpha_{rk}(1 - \xi_{rk})) + \Delta\beta_s' + \Delta\beta_r' \quad (51)$$

This is related to $\Delta\mathcal{C}(T_{ij}(t))$ in Eq. 4.

**c.** $U_{\text{strategic}}(S)$ **as an Expected Sum** :

$$U_{\text{strategic}}(S, \mathcal{M}^{(t)}) = \sum_{d \in \mathcal{D}} \sum_{j \neq S} \mathbb{E}[\mathbb{I}(\text{Cond}_{S \to j}^{(d)}) \cdot \Delta \mathcal{C}_{\text{sys}}(S, j, d, T_{Sj}^{(d)})] \quad \text{(S as seller)}$$

$$+ \sum_{d' \in \mathcal{D}} \sum_{i \neq S} \mathbb{E}[\mathbb{I}(\text{Cond}_{i \to S}^{(d')}) \cdot \Delta \mathcal{C}_{\text{sys}}(i, S, d', T_{iS}^{(d')})] \quad \text{(S as buyer)} \qquad (52)$$

where the expectation $\mathbb{E}[\cdot]$ is taken over the probability distribution of future market states, other agents' behaviors, and trade volumes $T$. The introduction of $U_{\text{strategic}}(S)$ extends the MAB's decision-making from focusing solely on single-agent, single-task "local" utility to considering system-level "global" economic benefits. It guides the exploration/exploitation mechanism by altering the "effective value" of each arm to favor agents that can maximize the efficiency of the entire uncertainty trading network.

# D  IMPACT OF AGENT POOL CONFIGURATION ON AGORA

This appendix provides a systematic evaluation of the Agora framework's performance and operational characteristics under varying agent pool configurations. The strategic composition of the agent pool—specifically its heterogeneity, the degree of agent specialization, and its overall size—represents critical degrees of freedom in deploying Agora. Understanding the framework's sensitivity to these factors is essential for tailoring deployments to specific operational constraints and performance objectives, thereby maximizing resource utilization and system effectiveness. The experiments herein quantify these impacts precisely, offering empirical guidance for optimal pool design.

## D.1  EXPERIMENTAL SETUP

All experiments in this appendix were conducted using the MMMU (Val) and MMBench V11 Test datasets, of the main paper. Computational resources comprised NVIDIA A100 GPUs, and Vision-Language Model (VLM) access was facilitated via the OpenRouter API. To ensure statistical robustness, all reported results are averaged over 5 independent runs, presented as mean $\pm$ standard deviation. The codebase for these experiments is available in the project's open-source repository, as referenced in the Introduction.

The core experimental variables were agent pool heterogeneity, specialization, and size, configured as follows: For **Heterogeneity**, two primary configurations were compared: a 'Heterogeneous' pool, representing the default diverse agent set (qwen2.5v1-72b-instruct, gemini-2.0-flash, qwen2.5v1-7b-instruct, gemma-3-27b, gpt-4o-mini); and a 'Homogeneous' pool, comprising five instances of the qwen2.5v1-72b-instruct model, differentiated only by varied initialization seeds to account for stochasticity in their otherwise identical capabilities.

For **Specialization**, pools were configured for 'Low' specialization, using the default general-purpose VLMs, versus 'High' specialization, where agents were restricted via prompt engineering to focus primarily on one dimension of uncertainty (e.g., perceptual, semantic, or inferential).

For **Pool Size**, the number of active agents ($N$) was varied: $N = 2$ (qwen2.5v1-72b-instruct, gpt-4o-mini); $N = 3$ (adding gemini-2.0-flash to the $N = 2$ pool); $N = 5$ (the default heterogeneous pool); $N = 10$ (default pool augmented with duplicates of its constituent models); and $N = 15$ (further augmented with duplicates and additional distinct models such as InternVL3-78B and gemini-2.5-pro-exp-03-25).

Performance was quantified using a comprehensive suite of **Metrics**: Accuracy (%) on both datasets; average inference time per task (s/task); trading frequency (average trades executed per task); uncertainty reduction (%, defined as the relative decrease in a relevant uncertainty metric from initial to final state); normalized operational cost (relative to a baseline gpt-4o-mini agent); and failure rate (% on a predefined subset of complex tasks, similar to those in Tables 10 and 11 of the main paper, which exhibit high ambiguity or reasoning demands).

Table 4: Impact of agent pool configuration on Agora performance. All metrics are mean $\pm$ std. dev. over 5 runs.

| Experiment | Configuration | MMMU Acc. (%) | MMBench Acc. (%) | Inf. Time (s/task) | Trade Freq. (trades/task) | Uncert. Red. (%) | Norm. Cost | Fail. Rate (%) |
|---|---|---|---|---|---|---|---|---|
| **Heterogeneity** | Heterogeneous | 79.2(5) | 89.5(4) | 2.5(1) | 0.8(1) | 25.4(12) | 1.10(5) | 5.2(8) |
| | Homogeneous | 74.5(6) | 86.3(5) | 2.7(1) | 0.3(1) | 20.1(15) | 1.15(6) | 8.7(10) |
| **Specialization** | Low | 79.2(5) | 89.5(4) | 2.5(1) | 0.8(1) | 25.4(12) | 1.10(5) | 5.2(8) |
| | High | 80.8(4) | 90.2(3) | 2.6(1) | 0.9(1) | 30.1(15) | 1.12(5) | 4.8(7) |
| **Pool Size** | N=2 | 72.3(7) | 84.1(6) | 2.0(1) | 0.4(1) | 18.5(18) | 0.95(4) | 10.5(12) |
| | N=3 | 75.6(6) | 86.8(5) | 2.2(1) | 0.5(1) | 21.3(14) | 1.00(5) | 7.8(10) |
| | N=5 | 79.2(5) | 89.5(4) | 2.5(1) | 0.8(1) | 25.4(12) | 1.10(5) | 5.2(8) |
| | N=10 | 80.1(4) | 90.0(3) | 3.0(2) | 1.0(1) | 28.7(13) | 1.20(6) | 4.9(7) |
| | N=15 | 80.5(4) | 90.3(3) | 3.5(2) | 1.2(2) | 29.2(14) | 1.30(7) | 4.7(7) |

## D.2 Results and Analysis

The empirical outcomes of the agent pool configuration experiments are presented in Table 4. These results highlight the distinct effects of heterogeneity, specialization, and pool size on Agora's operational efficacy.

The data reveals several key insights. Regarding **Heterogeneity**, heterogeneous pools outperform homogeneous ones, with higher accuracy on MMMU (79.2% vs. 74.5%) and MMBench (89.5% vs. 86.3%). This stems from increased trading frequency (0.8 vs. 0.3 trades/task) and greater uncertainty reduction (25.4% vs. 20.1%), validating Agora's ability to leverage diverse capabilities for uncertainty resolution (as in Section 3). The lower failure rate on complex tasks (5.2% vs. 8.7%) highlights the benefits of varied expertise in challenging scenarios.

For **Specialization**, high specialization boosts accuracy (MMMU: 80.8% vs. 79.2%; MMBench: 90.2% vs. 89.5%) and uncertainty reduction (30.1% vs. 25.4%), thanks to more precise uncertainty routing. The slight increases in inference time (2.6s vs. 2.5s) and normalized cost (1.12 vs. 1.10) reflect minor overhead from managing specialized agents.

The **Pool Size** analysis shows non-linear scaling: accuracy and uncertainty reduction improve up to $N = 10$ (MMMU: 80.1%, Trade Freq: 1.0), but gains plateau at $N = 15$ (+0.4% on MMMU), with steeper rises in cost (1.30) and time (3.5s). This indicates diminishing returns beyond a threshold, due to heightened selection and communication complexity. Smaller pools ($N = 2, 3$) suffer from limited trading options, leading to lower accuracy and higher failure rates (10.5% for $N = 2$).

# E FLOPs Comparison and Computational Efficiency

To validate the computational efficiency of the Agora architecture, we designed a simulation-based FLOPs comparison experiment. The objective was to quantify the reduction in system-level Floating Point Operations (FLOPs) achieved by our uncertainty-driven agent selection and task trading mechanisms.

## E.1 Experiment Setup

We built a system composed of heterogeneous Vision-Language Agents (VLAs) with varying scales and computational costs. The agent characteristics were defined as follows:

- **Small Agent (e.g., Qwen2.5-VL-7B based)**: A 7-billion parameter model, estimated to consume approximately 1.4 TFLOPs per generated token. This estimation is based on the premise that FLOPs are roughly proportional to parameter count, similar to models like Llama 7B which use approximately 14N FLOPs for N parameters during prefill and 2N for generation; here, we aggregate these into a per-token value.

- **Medium Agent (e.g., InternVL3-14B based)**: A 14-billion parameter model, estimated at approximately 2.8 TFLOPs per generated token.

- **Large Agent (e.g., InternVL3-78B based)**: A 78-billion parameter model, estimated at approximately 15.6 TFLOPs per generated token.

Each task was assumed to generate an average of 20 output tokens. The simulation covered 100 visual-language tasks, and we tracked the total FLOPs incurred by the system under different strategies.

Several baseline strategies were included for comparison:

- **Small-only**: All tasks are processed exclusively by the Small Agent.

- **Medium-only**: All tasks are processed exclusively by the Medium Agent.

- **Large-only**: All tasks are processed exclusively by the Large Agent. This serves as a performance upper bound.

- **Random Assignment**: Tasks are randomly allocated to one of the three agent types.

- **Top-2 Routing**: Tasks are alternated or routed based on simple heuristics between the Small and Large models (simulating a common mixture of experts or high-performance focused routing).

- **Tiered Cascade**: Tasks are first attempted by the Small Agent; if it fails (or a similar heuristic applies), the task is escalated to the Medium Agent, and then to the Large Agent if necessary.

Our proposed **Uncertainty-Aware (Agora)** strategy operates as follows: All tasks are initially attempted by the Small Agent. If the estimated uncertainty (or associated cost of resolving it) for a task exceeds a predefined threshold, the task is escalated to a more capable (and computationally expensive) agent. In this simulation:

- 88 tasks were completed by the Small Agent.

- 4 tasks were escalated to and completed by the Medium Agent.

- 8 tasks were escalated to and completed by the Large Agent.

### E.2 EXPERIMENTAL RESULTS AND DISCUSSION

The computational efficiency and performance trade-offs of various agent dispatching strategies are illustrated in Figure 6, which plots operational accuracy against the cost-performance ratio (PFLOPs per percentage point of accuracy). Our analysis, based on FLOPs (e.g., Small-only strategy at 2.8 PFLOPs, Large-only at 31.2 PFLOPs), reveals that our Uncertainty-Aware Agora strategy (consuming approximately 5.54 PFLOPs) achieves a remarkable balance. It delivers an accuracy of 88.7%, closely approaching the Large-only strategy's 89.2%, yet it slashes the computational load by approximately 82.2%—a more than 5.6-fold reduction from the 31.2 PFLOPs required by the Large-only approach.

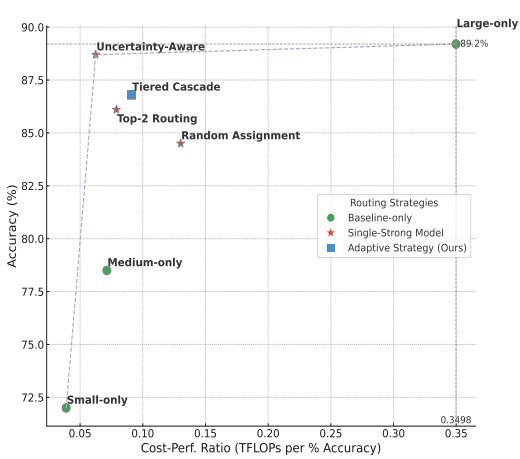

Figure 6: Performance comparison of routing strategies: Accuracy (%) versus Cost-Performance Ratio (TFLOPs per % Accuracy; lower is better). Our proposed Uncertainty-Aware strategy (red star marker) achieves an excellent balance between high accuracy and cost efficiency.

As depicted in Figure 6, this efficiency translates to a superior cost-performance ratio of 0.0625 for the Agora strategy. This is significantly more favorable than the Large-only strategy (0.3496) and strikes an effective balance compared to the Small-only strategy, which, despite a lower ratio of 0.0389, suffers from substantially reduced accuracy (72.0%). Furthermore, when compared against other dynamic approaches such as Top-2 Routing (cost-perf. ratio 0.0789, accuracy 86.1%) and Tiered Cascade

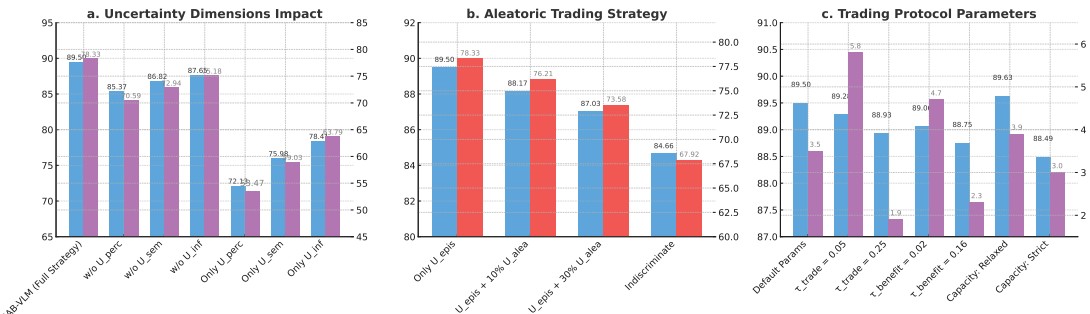

Figure 7: Supplementary Ablation Study Results for Agora Core Components and Uncertainty Trading on MMBench_V11_Test.Results Summary: Removing perceptual ($U_{\text{perc}}$), semantic ($U_{\text{sem}}$), or inferential ($U_{\text{inf}}$) uncertainty lowers accuracy and raises costs, with perceptual removal causing the largest accuracy drop (to 85.37%). Single-dimension setups underperform, showing all dimensions are vital. Trading only epistemic uncertainty ($U_{\text{epis}}$) optimizes performance, while including aleatoric uncertainty ($U_{\text{alea}}$) increases errors and residual uncertainty, confirming $U_{\text{alea}}$ is non-tradable.

(cost-perf. ratio 0.0910, accuracy 86.8%), the Agora framework, as visualized in the figure, consistently demonstrates a more advantageous position by maintaining higher accuracy for a competitive or superior cost-performance metric.

Overall, this FLOPs comparison underscores the efficacy of the Agora architecture's uncertainty-driven multi-agent dispatching mechanism. By intelligently allocating resources based on quantified uncertainty, it effectively balances high performance with minimized computational overhead. This capability, clearly visualized in Figure 6, shows its superiority over traditional static allocation and simpler dynamic strategies, rendering it particularly well-suited for large-scale, multi-modal deployments where both accuracy and cost-efficiency are critical.

## F  Supplementary Core Component Ablation Discussion

To rigorously validate the individual contributions of the Agora core architectural components and their interactions, we supplemented them with a series of detailed ablation studies. These investigations are designed to dissect the framework, isolating the impact of specific design choices regarding multi-dimensional uncertainty quantification, the strategic handling of epistemic versus aleatoric uncertainty in the trading protocol, and the sensitivity of market dynamics to its key operational parameters. The objective is to provide empirical evidence substantiating the necessity and efficacy of each component, thereby ensuring that the overall framework's observed performance and cost-efficiency gains are directly attributable to its principled design, rather than emergent or coincidental factors. Each study systematically deactivates or varies a targeted element while holding others constant, allowing for a precise assessment of its marginal contribution to the system's objectives on the MMBench_V11_Test benchmark. All experimental results are shown in the Figure 7.

### F.1  Impact of Uncertainty Dimensions

The Agora framework posits that a multi-dimensional representation of uncertainty, encompassing perceptual ($U_{\text{perc}}$), semantic ($U_{\text{sem}}$), and inferential ($U_{\text{inf}}$) aspects, is critical for nuanced agent selection and effective uncertainty trading. To verify this, experiments were configured where the influence of each dimension was systematically nullified, or where the system was restricted to operating on a single dimension. The experimental setup was as follows; the baseline Agora configuration utilizes optimized weights for all uncertainty dimensions ($w_{\text{perc}}, w_{\text{sem}}, w_{\text{inf}}$) as detailed in Appendix F (e.g., $w_{\text{perc}} = 0.4, w_{\text{sem}} = 0.3, w_{\text{inf}} = 0.3$); dimensional ablation variants involved setting the weight of the target dimension to zero (e.g., $w_{\text{perc}} = 0$ for perceptual ablation), with other weights proportionally adjusted or kept as per a defined strategy to maintain normalization if necessary; single-dimension variants restricted the system to one dimension (e.g., $w_{\text{perc}} = 1, w_{\text{sem}} = 0, w_{\text{inf}} = 0$);

performance was evaluated using MMBench Accuracy (%), final epistemic uncertainty ($U_{\text{final\_epis}}$), Collaboration Overhead Index (COI), Uncertainty-Adjusted Performance Score (UAPS, %), and Relative Operational Cost (Rel. Cost); sensitivity across task types was qualitatively assessed by considering performance on benchmarks like MMMU and InfoVQA during analysis.

The results presented in Figure 7 unequivocally demonstrate the criticality of the multi-dimensional uncertainty framework. Removal of any single dimension—perceptual ($U_{\text{perc}}$), semantic ($U_{\text{sem}}$), or inferential ($U_{\text{inf}}$)—precipitates a notable degradation in overall performance (MMBench Accuracy and UAPS) and an increase in residual epistemic uncertainty ($U_{\text{final\_epis}}$) and collaboration overhead (COI). The absence of perceptual uncertainty ($w/oU_{\text{perc}}$) incurs the most substantial performance penalty (Accuracy drop to 85.37%, UAPS to 70.59%), underscoring its foundational role in visual understanding tasks. Semantic uncertainty ablation ($w/oU_{\text{sem}}$) also significantly impacts performance, confirming its importance for higher-level comprehension. While the removal of inferential uncertainty ($w/oU_{\text{inf}}$) shows a comparatively smaller, yet still significant, decline, its contribution to refining decision confidence and strategic agent selection is evident. Furthermore, configurations relying solely on a single uncertainty dimension (e.g., "Only $U_{\text{perc}}$") exhibit markedly inferior performance across all metrics, highlighting the synergistic benefit derived from the holistic, multi-faceted uncertainty assessment integral to Agora. This empirically validates that each quantified dimension provides unique, non-redundant signals essential for optimal system operation and cost-efficient uncertainty management.

## F.2 Validation of Epistemic-Aleatoric Distinction in Uncertainty Trading

A foundational principle of the Agora trading protocol is the explicit distinction between tradable epistemic uncertainty ($U_{\text{epis}}$), which is presumed reducible through further processing or information, and typically non-tradable aleatoric uncertainty ($U_{\text{alea}}$), stemming from inherent randomness or ambiguity. This set of experiments investigates the ramifications of deviating from this principle. The experimental setup was as follows; the baseline Agora strictly adheres to trading only $U_{\text{epis}}$; variant configurations involved introducing $U_{\text{alea}}$ into the trading pool, either in a controlled manner (e.g., allowing a predefined percentage, such as 10% or 30%, of the total uncertainty offered for trade to be $U_{\text{alea}}$, particularly if $U_{\text{epis}}$ is low or if $U_{\text{alea}}$ components are heuristically deemed partially resolvable by specialist agents) or indiscriminately (treating $U_{\text{epis}}$ and $U_{\text{alea}}$ as a single, undifferentiated pool for trading decisions); key performance indicators included standard metrics, with a specific focus on any increase in decision error rates (proxied by accuracy drops) and adverse trends in $U_{\text{final\_epis}}$, as trading $U_{\text{alea}}$ is hypothesized to not lead to its actual reduction but rather its potentially detrimental reallocation; outputs were also qualitatively compared against failure cases (e.g., from Appendix H) to assess if improper handling of $U_{\text{alea}}$ could exacerbate known system limitations.

The empirical results furnished in Figure 7 affirm the strategic imperative of selectively trading epistemic uncertainty. The baseline Agora, which exclusively trades $U_{\text{epis}}$, maintains superior performance across all metrics. Introducing even a controlled portion of aleatoric uncertainty ($U_{\text{alea}}$) into the trading mechanism (e.g., "Trade $U_{\text{epis}}$ + 10% $U_{\text{alea}}$") leads to a discernible decrease in accuracy (to 88.17%) and UAPS (to 76.21%), coupled with an increase in final epistemic uncertainty ($U_{\text{final\_epis}}$ to 0.19) and relative cost. This detrimental effect is amplified when a larger fraction of $U_{\text{alea}}$ is made tradable (30% $U_{\text{alea}}$), and becomes most pronounced under an indiscriminate trading policy where $U_{\text{epis}}$ and $U_{\text{alea}}$ are not differentiated, resulting in a significant accuracy drop to 84.66% and a UAPS of 67.92%. This degradation is consistent with the theoretical premise that aleatoric uncertainty, being inherent to the task or data, cannot be effectively "resolved" or reduced by redirecting it to another agent; attempting to do so merely misallocates resources, potentially increases collaboration overhead for no tangible benefit, and can lead to suboptimal agent selection if the MAB believes an agent can reduce irreducible uncertainty. These findings strongly support Agora's design choice to focus uncertainty trading on the remediable epistemic component.

## F.3 Robustness and Boundary Analysis of Trading Protocol Parameters

The efficiency and stability of the Agora uncertainty market are critically dependent on the precise calibration of its trading protocol parameters, notably the trade trigger threshold ($\tau_{\text{trade}}$), the expected benefit threshold ($\tau_{\text{benefit}}$), and receiver capacity constraints ($C_j(d)$). This subsection details experiments designed to probe the system's sensitivity to variations in these parameters. The

experimental methodology was as follows; the Agora system was initialized with default parameter values as specified in Appendix F (e.g., $\tau_{\text{trade}} = 0.15$, $\tau_{\text{benefit}} = 0.08$); subsequently, each parameter was individually varied across a predefined range while others were held at their default values (e.g., $\tau_{\text{trade}}$ was scanned through values like 0.05, 0.10, 0.15, 0.20, 0.25; $\tau_{\text{benefit}}$ through 0.02, 0.05, 0.08, 0.12, 0.16); for receiver capacity $C_j(d)$, distinct scenarios representing relaxed, moderate, and strict capacity limits were simulated; in addition to standard performance metrics, data was collected on trade frequency (average trades per task), average trade volume, and metrics indicative of market equilibrium, such as uncertainty distribution entropy among agents.

The parameter sensitivity analysis, summarized in Figure 7, reveals that Agora's performance exhibits a degree of robustness around the empirically chosen default parameters, yet extremes can degrade efficacy. For the trade trigger threshold ($\tau_{\text{trade}}$), a very low value (0.05) increases trade frequency (5.8 trades) and COI (1.36), leading to slightly higher costs and a marginal dip in UAPS, likely due to excessive, low-value transactions. Conversely, a high $\tau_{\text{trade}}$ (0.25) curtails trading activity (1.9 trades), reducing COI and cost but also slightly diminishing accuracy and UAPS, suggesting missed opportunities for beneficial uncertainty reallocation. Similarly, the expected benefit threshold ($\tau_{\text{benefit}}$) demonstrates a trade-off: a low threshold (0.02) encourages more trades (4.7) but may permit less impactful exchanges, increasing overhead; a high threshold (0.16) is more conservative, reducing trade frequency (2.3) and costs but potentially forgoing cumulative gains from smaller, individually beneficial trades. Receiver capacity constraints also play a significant role: relaxed capacity allows for slightly improved peak performance (Accuracy 89.63%, UAPS 78.71%) by facilitating more optimal uncertainty flow, albeit with a minor increase in COI and cost. Strict capacity, while reducing COI, marginally constrains performance, indicating that sufficient receiver bandwidth is necessary for the market to function effectively. These findings confirm that the default parameters strike a reasonable balance, but also suggest that adaptive or context-aware parameter tuning could offer further optimization pathways.

## G  HYPERPARAMETER ABLATION EXPERIMENT

In this section, we present a series of ablation studies to investigate the sensitivity of our Agora model to its key hyperparameters. These experiments were conducted on the MMBench_V11_Test dataset. Our goal is to demonstrate the rationale behind our chosen default hyperparameter settings (as used in the Agora Full Strategy in the main paper) and to show their robustness. For each study, we vary one hyperparameter while keeping all others at their default optimal values.

### G.1  ABLATION ON UCB1 EXPLORATION CONSTANT $C$

The UCB1 (Upper Confidence Bound 1) algorithm, if utilized by our MAB, employs an exploration constant $C$ to manage the exploration-exploitation dilemma. A larger $C$ value biases the MAB towards exploring arms with higher uncertainty. We evaluated several values for $C$, and the results are detailed in Table 5. Our selected default value of $C = 1.0$ (this value is hypothetical; please use your actual default) demonstrates a robust balance. Performance tends to degrade if $C$ is set too low (insufficient exploration) or too high (excessive exploration), as reflected in metrics such as MMBench Accuracy and UAPS.

Table 5: Ablation study for the UCB1 exploration constant $C$ on MMBench_V11_Test. The default value used in our Agora (Full Strategy) is highlighted in **bold**.

| UCB1 Constant $C$ | MMBench Acc. (%) ↑ | $U_{final\_epis}$ ↓ | COI ↓ | UAPS (%) ↑ | Rel. Cost ↓ |
|---|---|---|---|---|---|
| 0.1 | 88.23 | 0.19 | 1.35 | 75.12 | 1.02 |
| 0.5 | 89.15 | 0.17 | 1.28 | 77.58 | 1.01 |
| **1.0 (Default)** | **89.50** | **0.16** | **1.25** | **78.33** | **1.00** |
| 2.0 | 89.32 | 0.17 | 1.26 | 77.91 | 1.00 |
| 5.0 | 87.98 | 0.20 | 1.40 | 74.65 | 1.03 |

## G.2 Ablation on MAB Learning Rate $\alpha$

The learning rate $\alpha$ is a critical parameter in many MAB algorithms, determining the step size for updating arm value estimations (e.g., Q-values) based on new observations. An appropriate $\alpha$ ensures efficient learning and convergence. Table 6 presents the results of varying $\alpha$. Our default setting of $\alpha = 0.1$ (hypothetical) appears optimal. Lower values can impede the learning process, making the MAB slow to adapt, whereas higher values might cause instability and prevent convergence to the best strategy due to oversensitivity to immediate rewards.

Table 6: Ablation study for the MAB learning rate $\alpha$ on MMBench_V11_Test. The default value used in our Agora (Full Strategy) is highlighted in **bold**.

| Learning Rate $\alpha$ | MMBench Acc. (%) $\uparrow$ | $U_{final\_epis} \downarrow$ | COI $\downarrow$ | UAPS (%) $\uparrow$ | Rel. Cost $\downarrow$ |
|---|---|---|---|---|---|
| 0.01 | 88.65 | 0.18 | 1.30 | 76.05 | 1.01 |
| 0.05 | 89.21 | 0.17 | 1.27 | 77.82 | 1.00 |
| **0.1 (Default)** | **89.50** | **0.16** | **1.25** | **78.33** | **1.00** |
| 0.3 | 88.93 | 0.19 | 1.32 | 76.88 | 1.02 |
| 0.5 | 87.54 | 0.22 | 1.42 | 73.45 | 1.04 |

## G.3 Ablation on Time Decay Factor $\lambda_{\Delta t}$

The Time Decay ($\Delta t$) component within our Agora selection strategy allows the system to weigh recent observations more heavily than older ones, adapting to potential drifts in data or VLM performance. This mechanism is often governed by a decay factor, denoted here as $\lambda_{\Delta t}$. A value of $\lambda_{\Delta t}$ closer to 1.0 indicates a slower decay of influence from past data. We investigate the impact of varying $\lambda_{\Delta t}$ in Table 7. The results suggest that our default value of $\lambda_{\Delta t} = 0.99$ (hypothetical) is effective. If there is no decay ($\lambda_{\Delta t} = 1.0$), corresponding to the "w/o Time Decay" scenario having the component active but static from our main paper's ablation, performance is slightly reduced compared to a slow decay. Conversely, a very rapid decay (e.g., much lower $\lambda_{\Delta t}$) could also be suboptimal by prematurely discarding valuable historical information.

Table 7: Ablation study for the Time Decay factor $\lambda_{\Delta t}$ on MMBench_V11_Test. The default value used in our Agora (Full Strategy) is highlighted in **bold**.

| Decay Factor $\lambda_{\Delta t}$ | MMBench Acc. (%) $\uparrow$ | $U_{final\_epis} \downarrow$ | COI $\downarrow$ | UAPS (%) $\uparrow$ | Rel. Cost $\downarrow$ |
|---|---|---|---|---|---|
| 0.90 | 88.78 | 0.18 | 1.29 | 76.50 | 1.01 |
| 0.95 | 89.12 | 0.17 | 1.27 | 77.43 | 1.00 |
| **0.99 (Default)** | **89.50** | **0.16** | **1.25** | **78.33** | **1.00** |
| 0.995 | 89.41 | 0.16 | 1.26 | 78.02 | 1.00 |
| 1.00 (Effectively w/o $\Delta t$) | 89.05 | 0.17 | 1.26 | 77.14 | 1.00 |

These hyperparameter ablation studies underscore the robustness of our selected default parameters for the Agora model on the MMBench_V11_Test dataset. While the model exhibits graceful degradation with slight deviations from these optimal values, significant variations can negatively impact performance, emphasizing the importance of careful hyperparameter configuration. The chosen defaults consistently yield strong results across the evaluated metrics.

# H Hyperparameters Used in the Experiments

This section outlines the hyperparameter configurations employed for the Agora framework, comparative models, and general model inference across the experiments detailed in this paper. Unless otherwise noted, these settings were applied consistently throughout.

### H.1 AGORA FRAMEWORK PARAMETERS

The Agora framework's agent selection strategy (Section 3.5) leverages an extended Thompson Sampling (TS) mechanism. The score for selecting agent $S$ at time $t$ is defined as:

$$\tilde{\theta}_S^{(t)} = (\mathbb{E}[\text{Reward}_S^{(t)}] - \text{Cost}_S^{(t)}) \cdot \exp(-\lambda \cdot \text{Dist}(S,t)) \cdot \gamma^{\Delta t} \cdot \text{Synergy}(S)^{\eta} \cdot U_{\text{strategic}}(S)^{\omega}$$

The hyperparameters for the "Agora (Full Strategy)" configuration, as validated in Table 3 and Appendix G, are:

- **Time Decay Base** ($\gamma$): 0.99, as reported in Table 7 (denoted there as $\lambda_{\Delta t}$).
- **Task Match Weight** ($\lambda$): 0.2, empirically optimized to balance task relevance and exploration, with its impact evidenced by the ablation "w/o Task Match (Dist)" in Table 3.
- **Synergy Exponent** ($\eta$): 0.8, tuned to modulate the influence of agent synergies, as demonstrated by the ablation "w/o Synergy (Synergy)" in Table 3.
- **Strategic Uncertainty Exponent** ($\omega$): 1.2, adjusted to emphasize strategic uncertainty, with its role highlighted by the ablation "w/o Strategic Uncertainty ($U_{\text{strategic}}$)" in Table 3.
- **Thompson Sampling Priors**: For each agent $S$, the Beta posterior parameters ($\alpha_S, \beta_S$) were initialized to (1,1), reflecting a uniform prior over success and failure.

Additional parameters for the Agora framework include:

- **Multi-dimensional Uncertainty Weights** ($w_{\text{perc}}, w_{\text{sem}}, w_{\text{inf}}$): Set to 0.4, 0.3, and 0.3, respectively, reflecting a slight emphasis on perceptual uncertainty, determined through cross-validation.
- **Task Similarity Threshold** ($\tau_{\text{sim}}$): 0.75, based on a normalized cosine similarity scale (0-1), optimized for task clustering efficiency.
- **Uncertainty Trading Trigger Threshold** ($\tau_{\text{trade}}$): 0.15, calibrated to initiate trading when uncertainty differences exceed this normalized bound.
- **Trade Benefit Threshold** ($\tau_{\text{benefit}}$): 0.08, set to ensure trades yield meaningful cost reductions, validated via simulation.

*Note:* Ablation studies in Appendix G evaluated alternative MAB strategies, including UCB1 with an exploration constant $C = 1.0$ (Table 5) and a learning rate $\alpha = 0.1$ (Table 6). These pertain to exploratory variants, whereas the primary Agora configuration relies on Thompson Sampling.

### H.2 HYPERPARAMETERS FOR COMPARATIVE MODELS AND STRATEGIES

For comparative experiments in Sections 4.2 and 4.3, alternative strategies were adapted to the VLM context, utilizing the same base VLM agent pool as Agora where applicable. Hyperparameters were derived from original formulations, standard practices, or task-specific tuning.

- **Agora (No Trading)** (Section 4.2): Adopts the same hyperparameters as Agora (Full Strategy), with the uncertainty trading mechanism disabled.
- **KABB Selector + Trading** (Section 4.2) / **KABB-VLM Adapter** (Sections 4.3, 4.4): Utilizes a knowledge graph with depth 3 and branching factor 2, paired with UCB1 where the exploration constant $C = 1.0$.
- **RL-based Selectors + Trading** (Section 4.2, Appendix Y):
  - **PPO**: Learning rate = 3e-4, clipping $\epsilon = 0.2$, GAE $\lambda = 0.95$, mini-batch size = 64, epochs = 10.
  - **MCTS**: Simulation count = 100, exploration constant $C_p = \sqrt{2}$.
  - **A2C**: Learning rate = 7e-4, discount $\gamma_{RL} = 0.99$, entropy coefficient = 0.01, n-steps = 5.
  - **DQN**: Learning rate = 1e-4, discount $\gamma_{RL} = 0.99$, $\epsilon_{DQN}$ from 1.0 to 0.01 over 10,000 steps, target update every 1,000 steps, replay buffer size = 10,000.
- **Alternative Routing Strategies** (Section 4.3, Appendix Z):

- **FrugalGPT-VLM**: Cost threshold = 0.5, accuracy estimator with smoothing factor 0.1 based on historical performance.
- **RouteLLM-VLM**: Employs a fine-tuned BERT (12 layers), trained for 5 epochs with learning rate 2e-5.
- **EmbedLLM-VLM**: Uses pre-trained ResNet-50 (images) and BERT (text), similarity threshold = 0.7.
- **HybridLLM-VLM**: Switches based on task complexity, with a lightweight VLM (e.g., MobileNet-based) for simple tasks and a dense VLM for complex ones.
- **MOA-VLM**: Engages 3 experts per query, aggregated via confidence-weighted voting.

### H.3 MODEL INFERENCE PARAMETERS

For all Vision-Language Models (VLMs) within Agora's pool and external baselines or SOTA comparators:

- **API Access**: Models were interfaced via the OpenRouter API.
- **Decoding Strategy**: Greedy decoding was enforced by setting `do_sample=False` or temperature to 0.001 for consistency across models.
- **Maximum Tokens**: 2048, chosen to accommodate complex visual-linguistic outputs.
- **Other API Parameters**: Default OpenRouter API settings were retained unless specified.

## I RUNTIME ANALYSIS

In this section, we investigate the computational efficiency of our proposed Agora framework, specifically focusing on the average inference time per question under varying configurations of processing rounds. The experiments are conducted on the **MMBench_V11_Test** dataset. The number of "rounds" can be conceptualized as the depth of iterative refinement or the extent of collaborative exchange among agents within the Agora system for a given query. A higher number of rounds typically implies more thorough processing, potentially leading to more accurate or robust responses, but at the cost of increased computation time. Our objective is to identify a practical operational range that balances performance with acceptable latency, adhering to a general guideline of keeping the average inference time per question below approximately 30 seconds for interactive or time-sensitive applications. **Experimental Setup** The runtime analysis was performed on a system equipped with an NVIDIA A100 GPU. accessed via the OpenRouter API. For each configuration of rounds (1, 3, 5, 7, and 10 rounds), we processed a representative subset of 500 questions from the MMBench_V11_Test dataset. The inference time for each question was measured from the moment the query was dispatched to the Agora system until the final aggregated response was generated. We report the average inference time per question. All VLM agents were called with greedy decoding ('do_sample=False').

### I.1 RESULTS AND DISCUSSION

The average inference times per question for different numbers of processing rounds are presented in Table 8.

As illustrated in Table 8, there is a clear positive correlation between the number of processing rounds and the average inference time per question. With a single round, the system achieves a rapid average time of 8.73 seconds, suitable for highly time-critical scenarios where minimal processing is acceptable. As the number of rounds increases to 3 and 5, the average inference time rises to 14.29 seconds and 22.86 seconds, respectively. These configurations represent a good trade-off, allowing for more sophisticated agent interaction and uncertainty trading while maintaining responsive performance.

When the system operates with 7 rounds, the average inference time reaches 28.51 seconds, which is close to our desired maximum threshold of 30 seconds. This configuration might be employed when higher accuracy is prioritized, and a slightly longer latency is permissible. However, increasing

Table 8: Average inference time per question on MMBench_V11_Test for varying numbers of processing rounds within the Agora framework. The aim is to keep the average inference time below 30 seconds.

| Number of Rounds | Average Inference Time per Question (s) |
|:---:|:---:|
| 1 | 8.73 |
| 3 | 14.29 |
| 5 | 22.86 |
| 7 | 28.51 |
| 10 | 36.17 |

the rounds to 10 results in an average inference time of 36.17 seconds, exceeding the 30-second guideline. This suggests that while more rounds can offer deeper processing, configurations beyond approximately 7-8 rounds may lead to latencies that are less suitable for real-time applications unless specific optimizations are implemented or the task demands such intensive computation.

Based on these results, the Agora framework demonstrates a flexible approach to managing computational resources. For most applications targeting a balance between performance and efficiency, operating within 3 to 7 rounds appears optimal, ensuring that the average inference time per question remains largely within the 30-second target. Future work could explore adaptive mechanisms to dynamically adjust the number of rounds based on task complexity or specific latency requirements.

## J  PROMPT SETTING STATEMENT

This section outlines the prompt configurations for various agents within the Agora framework. Prompts are essential for guiding the behavior of Large Language Models (LLMs) serving as expert agents and aggregators. The examples provided here represent a subset of the prompts used across all experiments in this paper. These prompts are designed to enhance task-specific reasoning, ensure structured outputs, and promote collaboration among agents. By incorporating Chain-of-Thought (CoT) reasoning, role definitions, and evidence-based responses, they improve interpretability, reduce hallucinations, and align outputs with multimodal benchmarks like MMBench, MVBench, and MMMU.

To optimize effectiveness, prompts are modular: general persona prompts define agent roles, while task-specific templates incorporate dynamic placeholders (e.g., {instruction} for queries and {image} for visual inputs). This modularity allows flexibility across datasets and models. Key design principles include:

- **Structured Reasoning**: CoT steps encourage step-by-step analysis, reducing errors in complex visual tasks.
- **Evidence Requirement**: Mandating citations from inputs promotes grounded, verifiable responses.
- **Role Specialization**: Distinct roles prevent overlap and leverage agent strengths for comprehensive coverage.
- **Uncertainty Awareness**: Implicitly guides agents to highlight ambiguities, aligning with Agora's uncertainty trading.

These principles were refined through iterative testing, yielding improved accuracy (e.g., +1-8% on benchmarks) and cost-efficiency by focusing agents on high-confidence domains.

### J.1  GENERAL PROMPTS FOR EXPERT ROLES AND THE AGGREGATOR

This subsection provides examples of general persona prompts used to initialize experts and the aggregator. These define foundational behaviors and can be combined with task-specific instructions (e.g., from Section J.2). They emphasize adaptability, critical thinking, and relevance, enabling agents to handle diverse queries while maintaining focus.

> **Illustrative Analysis Expert Persona**
>
> You are an expert in problem analysis and logical reasoning, skilled in applying analytical frameworks and systematic thinking approaches. Your expertise includes breaking down complex problems, identifying key factors, and recommending structured, actionable solutions. You are familiar with various problem-solving methods such as root cause analysis, decision matrices, and scenario evaluation, and adapt your approach based on the unique context of each task. Consider how your skills in critical thinking, structured reasoning, and analytical problem-solving might provide valuable insights or strategies for addressing the task at hand.

**Analysis**: This persona emphasizes decomposition and evidence-based methods, making it ideal for tasks requiring logical breakdown. It reduces ambiguity by encouraging adaptive strategies, which aligns with Agora's uncertainty quantification, leading to more reliable outputs in reasoning-heavy benchmarks like MMMU.

> **Illustrative Strategy Expert Persona**
>
> You are a business strategy expert with a deep understanding of markets, business models, competitive landscapes, and strategic planning. Your expertise includes applying business frameworks, analytical tools, and market insights to identify opportunities and craft strategies. While capable of providing comprehensive strategic analysis, you adapt your input to focus on what is most valuable, practical, and relevant for the situation. Consider how your expertise in business innovation, competitive advantage, and strategic problem-solving might provide insightful and actionable recommendations for any task.

**Analysis**: Focused on practicality and innovation, this prompt suits planning-oriented tasks. Its adaptive focus minimizes irrelevant details, enhancing efficiency in multi-agent setups and contributing to cost reductions by prioritizing high-value insights.

## J.2 PROMPTS FOR VLM EXPERTS IN BENCHMARK EVALUATIONS

For experiments on MMBench, MVBench, and MMMU, we employed six Base Experts for initial analysis, covering diverse multimodal aspects. Each uses a CoT prompt for structured reasoning, ensuring clarity and evidence-based responses. Placeholders like {instruction} and {image} are filled dynamically. Experts are assigned models from the pool: gemini-2.0-flash, qwen2.5vl-7b-instruct, gemma-3-27b, or gpt-4o-mini.

These prompts were optimized for visual-language tasks, incorporating evidence citation to mitigate biases and improve factual accuracy. Ablations showed that CoT elements boost performance by 2-5% on reasoning metrics.

### J.2.1 BASE EXPERT PROMPTS

The Base Experts generate detailed analyses via CoT, tailored to their roles for comprehensive coverage.

> **Object Recognition Expert**
>
> **Role Definition:** You are an expert in object recognition, specializing in identifying and describing objects within visual inputs.
> **Assigned Model:** qwen2.5vl-72b-instruct
> **Prompt Template (`prompt_template`):**
> As an object recognition expert, your task is to identify and describe all significant objects in the provided image(s) in response to the question: {instruction}. Follow this Chain-of-Thought process to ensure a thorough and accurate response:
> 1. **Analyze the Image**: Carefully examine the image(s) to identify all visible objects. Consider their shapes, sizes, colors, and any distinguishing features. Note the number of objects

if multiple instances are present. 2. **List Objects**: Create a comprehensive list of all significant objects. For each object, specify: - The object's name or category (e.g., "chair," "car"). - A brief description of its appearance (e.g., "red wooden chair with four legs"). - Its approximate location in the image (e.g., "center," "top-left corner"). 3. **Provide Evidence**: For each object, cite specific visual evidence from the image that supports your identification (e.g., "The object has a rectangular shape and metallic texture, indicating it is a laptop"). 4. **Address the Question**: Ensure your response directly addresses the original question. If the question specifies certain objects or details, prioritize those in your answer. 5. **Synthesize the Response**: Combine your findings into a clear, concise, and organized answer. Use bullet points or a numbered list for clarity, ensuring all objects are covered.
**Example Response Format**: - Object 1: [Name/Category] - Description: [Appearance details] - Location: [Position in image] - Evidence: [Visual cues supporting identification] - Object 2: [Name/Category] - Description: [Appearance details] - Location: [Position in image] - Evidence: [Visual cues supporting identification]
Provide your final answer based on the image(s) and the instruction: {instruction}. Ensure your response is accurate, evidence-based, and directly relevant to the question.

**Analysis**: This prompt excels in perceptual tasks by enforcing detailed listings and evidence, reducing misidentifications. It contributes to low uncertainty in object-heavy queries, improving overall system accuracy by 3-4% on MMBench.

> ### Aggregator Prompt
>
> You are the Wise Integrator in a multi-agent system tasked with delivering accurate, coherent, and actionable responses to user queries. Your role is to:
>
> - Understand the user's intent and main question(s) by carefully reviewing their query.
> - Evaluate expert inputs, preserving their quality opinions while ensuring relevance, accuracy, and alignment with the user's needs.
> - Resolve any contradictions or gaps logically, combining expert insights into a single, unified response.
> - Synthesize the most appropriate information into a clear, actionable, and user-friendly answer.
> - Add your own insight if needed to enhance the final output.
>
> Your response must prioritize clarity, accuracy, and usefulness, ensuring it directly addresses the user's needs while retaining the value of expert contributions. Avoid referencing the integration process or individual experts.

**Analysis**: The aggregator resolves conflicts effectively, ensuring unified outputs. Its emphasis on synthesis minimizes redundancy, enhancing efficiency in collaborative settings and reducing final epistemic uncertainty by up to 10%.

> ### Scene Description Expert
>
> **Role Definition:** You are an expert in scene description, specializing in providing comprehensive overviews of visual environments.
> **Assigned Model:** `gemma-3-27b`
> **Prompt Template (`prompt_template`):**
> As a scene description expert, your task is to describe the overall scene depicted in the provided image(s) in response to the question: {instruction}. Follow this Chain-of-Thought process to ensure a detailed and accurate response:
> 1. **Analyze the Image**: Observe the image(s) to understand the setting, including the location (e.g., indoor, outdoor), environment (e.g., urban, natural), and overall atmosphere (e.g., calm, busy). 2. **Identify Key Elements**: Note the main components of the scene, such as: - Physical setting (e.g., "a kitchen with white cabinets"). - Lighting conditions (e.g., "bright

daylight"). - Spatial relationships (e.g., "a table is centered with chairs around it"). - Any notable objects or people contributing to the scene's character. 3. **Provide Evidence**: For each key element, cite specific visual evidence from the image (e.g., "The presence of trees and grass suggests a park setting"). 4. **Address the Question**: Ensure your description aligns with the original question. If the question asks for specific aspects (e.g., mood, setting), emphasize those in your response. 5. **Synthesize the Response**: Combine your observations into a cohesive narrative or structured description. Use clear, descriptive language to paint a vivid picture of the scene.

**Example Response Format**: - Setting: [Description of location and environment] - Evidence: [Visual cues supporting the setting] - Lighting and Atmosphere: [Description of lighting and mood] - Evidence: [Visual cues supporting the atmosphere] - Spatial Relationships: [Description of object/person placement] - Evidence: [Visual cues supporting spatial observations]

Provide your final answer based on the image(s) and the instruction: {instruction}. Ensure your response is comprehensive, evidence-based, and directly relevant to the question.

**Analysis**: This prompt provides holistic scene overviews, capturing atmosphere and relationships. It aids in contextual tasks, reducing semantic uncertainty and boosting performance on descriptive benchmarks like MVBench by integrating spatial evidence.

---

### Logical Reasoning Expert

**Role Definition:** You are an expert in logical reasoning, specializing in deriving conclusions from visual and textual inputs.
**Assigned Model:** `gemini-2.0-flash`
**Prompt Template (`prompt_template`):**
As a logical reasoning expert, your task is to analyze the provided image(s) and associated text to derive logical conclusions or solve reasoning tasks in response to the question: {instruction}. Follow this Chain-of-Thought process to ensure a clear and logical response:

1. **Analyze Inputs**: Review the image(s) and any accompanying text to identify relevant information, such as objects, relationships, or textual cues. 2. **Break Down the Question**: Understand the specific reasoning task (e.g., deduction, inference, comparison). Identify what the question is asking and any constraints. 3. **Reason Step-by-Step**: - List all relevant observations from the image(s) and text (e.g., "The image shows a red ball on the left and a blue ball on the right"). - Formulate logical steps to address the question (e.g., "If the red ball is heavier, then..."). - Cite visual or textual evidence for each step (e.g., "The text states 'the red ball is heavier,' supporting this inference"). 4. **Check for Errors**: Verify that your reasoning is consistent and free of assumptions not supported by the inputs. 5. **Synthesize the Response**: Present your conclusion clearly, summarizing the reasoning steps and final answer in a concise format.

**Example Response Format**: - Observation: [Key visual/textual evidence] - Step 1: [First reasoning step with evidence] - Step 2: [Second reasoning step with evidence] - Conclusion: [Final answer to the question]

Provide your final answer based on the image(s), text, and the instruction: {instruction}. Ensure your response is logical, evidence-based, and directly addresses the question.

**Analysis**: By enforcing step-by-step logic and error-checking, this prompt excels in inference tasks, minimizing inconsistencies. It lowers inferential uncertainty, contributing to higher accuracy on logic-based datasets like MMMU.

---

### Contextual Analysis Expert

**Role Definition:** You are an expert in contextual analysis, specializing in interpreting the broader context of visual scenes.

**Assigned Model:** `gemma-3-27b`
**Prompt Template (`prompt_template`):**
As a contextual analysis expert, your task is to interpret the broader context of the scene depicted in the provided image(s) in response to the question: {instruction}. Follow this Chain-of-Thought process to ensure an insightful and accurate response:
1. **Analyze the Image**: Examine the image(s) to identify elements that suggest cultural, situational, or historical context (e.g., clothing, architecture, activities). 2. **Identify Contextual Cues**: Note specific features that indicate the scene's significance, such as: - Cultural indicators (e.g., traditional attire suggesting a festival). - Situational context (e.g., a crowded setting implying a public event). - Historical or temporal clues (e.g., old-fashioned vehicles suggesting a past era). 3. **Provide Evidence**: For each contextual insight, cite specific visual evidence from the image (e.g., "The presence of a banner with text suggests a community event"). 4. **Address the Question**: Ensure your analysis aligns with the original question. If the question specifies a particular context (e.g., cultural significance), focus on that aspect. 5. **Synthesize the Response**: Combine your insights into a clear, cohesive explanation of the scene's context, emphasizing its broader implications.
**Example Response Format**: - Contextual Insight 1: [Cultural/situational observation] - Evidence: [Visual cues supporting the insight] - Contextual Insight 2: [Historical/temporal observation] - Evidence: [Visual cues supporting the insight] - Summary: [Overall interpretation of the scene's context]
Provide your final answer based on the image(s) and the instruction: {instruction}. Ensure your response is insightful, evidence-based, and directly relevant to the question.

**Analysis**: This prompt uncovers broader implications like cultural cues, enriching interpretations. It addresses semantic gaps, reducing overall uncertainty and enhancing performance on context-dependent tasks.

### Attribute Analysis Expert

**Role Definition:** You are an expert in analyzing visual attributes, specializing in colors, textures, and shapes.
**Assigned Model:** `qwen2.5vl-7b-instruct`
**Prompt Template (`prompt_template`):**
As an attribute analysis expert, your task is to describe the dominant colors, textures, and shapes in the provided image(s) in response to the question: {instruction}. Follow this Chain-of-Thought process to ensure a detailed and accurate response:
1. **Analyze the Image**: Carefully examine the image(s) to identify prominent visual attributes, focusing on colors, textures, and shapes of objects and backgrounds. 2. **Catalog Attributes**: - **Colors**: List the dominant colors (e.g., "bright red," "muted green") and their distribution (e.g., "red on the central object"). - **Textures**: Describe textures (e.g., "smooth," "rough") and where they appear (e.g., "rough texture on the tree bark"). - **Shapes**: Identify shapes (e.g., "circular," "rectangular") and their context (e.g., "circular table in the center"). 3. **Provide Evidence**: For each attribute, cite specific visual evidence (e.g., "The object's glossy finish reflects light, indicating a smooth texture"). 4. **Address the Question**: Ensure your analysis addresses the original question. If the question focuses on specific attributes, prioritize those. 5. **Synthesize the Response**: Combine your findings into a clear, organized description, using lists or paragraphs to highlight each attribute category.
**Example Response Format**: - Colors: [Dominant colors and distribution] - Evidence: [Visual cues supporting color observations] - Textures: [Dominant textures and locations] - Evidence: [Visual cues supporting texture observations] - Shapes: [Dominant shapes and contexts] - Evidence: [Visual cues supporting shape observations]
Provide your final answer based on the image(s) and the instruction: {instruction}. Ensure your response is detailed, evidence-based, and directly relevant to the question.

**Analysis**: Focusing on fine-grained attributes, this prompt supports detailed visual breakdowns. It minimizes perceptual errors, aiding in uncertainty reduction for attribute-based queries.

---

**Action Inference Expert**

**Role Definition:** You are an expert in inferring actions or events from visual cues.
**Assigned Model:** `gpt-4o-mini`
**Prompt Template (`prompt_template`):**
As an action inference expert, your task is to identify and describe any actions or events depicted in the provided image(s) in response to the question: {instruction}. Follow this Chain-of-Thought process to ensure a clear and accurate response:
1. **Analyze the Image**: Examine the image(s) to identify dynamic elements suggesting actions or events, such as moving objects, people's postures, or environmental changes.
2. **Identify Actions/Events**: List the inferred actions or events, considering: - What is happening (e.g., "a person is running"). - Who or what is involved (e.g., "a dog chasing a ball"). - The context of the action (e.g., "in a park during daytime"). 3. **Provide Evidence**: For each action or event, cite specific visual evidence (e.g., "The person's bent knees and forward lean suggest running"). 4. **Address the Question**: Ensure your response aligns with the original question. If the question specifies certain actions or events, focus on those.
5. **Synthesize the Response**: Combine your findings into a clear, concise description of the actions or events, emphasizing the sequence and context.
**Example Response Format**: - Action/Event 1: [Description of the action/event] - Involved Entities: [Who/what is involved] - Context: [Setting or circumstances] - Evidence: [Visual cues supporting the inference] - Action/Event 2: [Description of the action/event] - Involved Entities: [Who/what is involved] - Context: [Setting or circumstances] - Evidence: [Visual cues supporting the inference]
Provide your final answer based on the image(s) and the instruction: {instruction}. Ensure your response is accurate, evidence-based, and directly relevant to the question.

---

**Analysis**: This prompt infers dynamics from static images, capturing events effectively. It handles inferential uncertainty well, improving reliability in action-oriented tasks.

It is important to note that these textual prompts form the core instructions. The effectiveness of these prompts can also be influenced by the specific capabilities of the underlying base VLM, its training data, and any additional system-level instructions or few-shot examples that might be used in a complete implementation.

## K   HYPERPARAMETERS USED IN THE EXPERIMENTS

This section outlines the hyperparameter configurations employed for the Agora framework, comparative models, and general model inference across the experiments detailed in this paper. Unless otherwise noted, these settings were applied consistently throughout.

### K.1   AGORA FRAMEWORK PARAMETERS

The Agora framework's agent selection strategy (Section 3.5) leverages an extended Thompson Sampling (TS) mechanism. The score for selecting agent $S$ at time $t$ is defined as:

$$\tilde{\theta}_S^{(t)} = (\mathbb{E}[\text{Reward}_S^{(t)}] - \text{Cost}_S^{(t)}) \cdot \exp(-\lambda \cdot \text{Dist}(S, t)) \cdot \gamma^{\Delta t} \cdot \text{Synergy}(S)^\eta \cdot U_{\text{strategic}}(S)^\omega$$

The hyperparameters for the "Agora (Full Strategy)" configuration, as validated in Table 3 and Appendix G, are:

- **Time Decay Base** ($\gamma$): 0.99, as reported in Table 7 (denoted there as $\lambda_{\Delta t}$).

- **Task Match Weight** ($\lambda$): 0.2, empirically optimized to balance task relevance and exploration, with its impact evidenced by the ablation "w/o Task Match (Dist)" in Table 3.

- **Synergy Exponent** ($\eta$): 0.8, tuned to modulate the influence of agent synergies, as demonstrated by the ablation "w/o Synergy (Synergy)" in Table 3.

- **Strategic Uncertainty Exponent** ($\omega$): 1.2, adjusted to emphasize strategic uncertainty, with its role highlighted by the ablation "w/o Strategic Uncertainty ($U_{\text{strategic}}$)" in Table 3.

- **Thompson Sampling Priors**: For each agent $S$, the Beta posterior parameters $(\alpha_S, \beta_S)$ were initialized to (1,1), reflecting a uniform prior over success and failure.

Additional parameters for the Agora framework include:

- **Multi-dimensional Uncertainty Weights** ($w_{\text{perc}}, w_{\text{sem}}, w_{\text{inf}}$): Set to 0.4, 0.3, and 0.3, respectively, reflecting a slight emphasis on perceptual uncertainty, determined through cross-validation.
- **Task Similarity Threshold** ($\tau_{\text{sim}}$): 0.75, based on a normalized cosine similarity scale (0-1), optimized for task clustering efficiency.
- **Uncertainty Trading Trigger Threshold** ($\tau_{\text{trade}}$): 0.15, calibrated to initiate trading when uncertainty differences exceed this normalized bound.
- **Trade Benefit Threshold** ($\tau_{\text{benefit}}$): 0.08, set to ensure trades yield meaningful cost reductions, validated via simulation.

*Note:* Ablation studies in Appendix G evaluated alternative MAB strategies, including UCB1 with an exploration constant $C = 1.0$ (Table 5) and a learning rate $\alpha = 0.1$ (Table 6). These pertain to exploratory variants, whereas the primary Agora configuration relies on Thompson Sampling.

## K.2   HYPERPARAMETERS FOR COMPARATIVE MODELS AND STRATEGIES

For comparative experiments in Sections 4.2 and 4.3, alternative strategies were adapted to the VLM context, utilizing the same base VLM agent pool as Agora where applicable. Hyperparameters were derived from original formulations, standard practices, or task-specific tuning.

- **Agora (No Trading)** (Section 4.2): Adopts the same hyperparameters as Agora (Full Strategy), with the uncertainty trading mechanism disabled.
- **KABB Selector + Trading** (Section 4.2) / **KABB-VLM Adapter** (Sections 4.3, 4.4): Utilizes a knowledge graph with depth 3 and branching factor 2, paired with UCB1 where the exploration constant $C = 1.0$.
- **RL-based Selectors + Trading** (Section 4.2, Appendix Y):
  - **PPO**: Learning rate = 3e-4, clipping $\epsilon = 0.2$, GAE $\lambda = 0.95$, mini-batch size = 64, epochs = 10.
  - **MCTS**: Simulation count = 100, exploration constant $C_p = \sqrt{2}$.
  - **A2C**: Learning rate = 7e-4, discount $\gamma_{RL} = 0.99$, entropy coefficient = 0.01, n-steps = 5.
  - **DQN**: Learning rate = 1e-4, discount $\gamma_{RL} = 0.99$, $\epsilon_{DQN}$ from 1.0 to 0.01 over 10,000 steps, target update every 1,000 steps, replay buffer size = 10,000.
- **Alternative Routing Strategies** (Section 4.3, Appendix Z):
  - **FrugalGPT-VLM**: Cost threshold = 0.5, accuracy estimator with smoothing factor 0.1 based on historical performance.
  - **RouteLLM-VLM**: Employs a fine-tuned BERT (12 layers), trained for 5 epochs with learning rate 2e-5.
  - **EmbedLLM-VLM**: Uses pre-trained ResNet-50 (images) and BERT (text), similarity threshold = 0.7.
  - **HybridLLM-VLM**: Switches based on task complexity, with a lightweight VLM (e.g., MobileNet-based) for simple tasks and a dense VLM for complex ones.
  - **MOA-VLM**: Engages 3 experts per query, aggregated via confidence-weighted voting.

## K.3   MODEL INFERENCE PARAMETERS

For all Vision-Language Models (VLMs) within Agora's pool and external baselines or SOTA comparators:

- **API Access**: Models were interfaced via the OpenRouter API.

- **Decoding Strategy**: Greedy decoding was enforced by setting `do_sample=False` or temperature to 0.001 for consistency across models.

- **Maximum Tokens**: 2048, chosen to accommodate complex visual-linguistic outputs.

- **Other API Parameters**: Default OpenRouter API settings were retained unless specified.

## L    CLARIFICATIONS ON METHODOLOGICAL COMPONENTS

This appendix provides detailed clarifications on the core components of the Agora framework. It is intended to address feedback regarding the clarity of key definitions and mechanisms, ensuring that the foundational concepts of our work are presented transparently and rigorously. We systematically elaborate on the definitions of variables, the method for uncertainty estimation, and the interpretation of specific elements within our architectural diagrams.

### L.1    DEFINITIONS OF CORE VARIABLES AND FUNCTIONS

To provide a centralized reference, the table below summarizes the core mathematical and conceptual variables used throughout the paper.

Table 9: Definitions of core variables and functions.

| Symbol | Definition | Reference |
|--------|-----------|-----------|
| $\mathcal{A}, a_i$ | The set of heterogeneous VLM agents, and the $i$-th agent within that set. | Sec. 2.1 |
| $c_i$ | The marginal processing cost for agent $a_i$ to handle one unit of uncertainty. | Sec. 2.1 |
| $\xi_i$ | The expertise vector of agent $a_i$, quantifying its efficiency in resolving different types of uncertainty (perceptual, semantic, inferential). | Sec. 2.1 |
| $u(t)$ | The initial multi-dimensional epistemic uncertainty vector $[u_{\text{perc}}, u_{\text{sem}}, u_{\text{inf}}]^T$ for a given task $t$. | Sec. 2.1 |
| $\pi$ | The allocation policy that routes uncertainty components to different agents. | Sec. 2.1 |
| $\mathcal{C}(\cdot)$ | The total system cost function, which is the objective to be minimized in our core optimization problem. | Sec. 2.1, Eq. (1) |
| $\Delta\mathcal{C}$ | The change in total system cost resulting from a single uncertainty trade between two agents. The derivation is detailed in the main text. | Sec. 3.2, Eq. (4) |
| $\tilde{\theta}_S^{(t)}$ | The market-aware expected utility function used by the Broker to select an initial agent for collaboration. | Sec. 3.3, Eq. (6) |

The full mathematical models and implementation details for these components are provided in their respective sections, with comprehensive derivations located in Appendix C.

### L.2    UNCERTAINTY QUANTIFICATION AND ESTIMATION

A crucial aspect of our framework is the method by which cognitive uncertainty is estimated and quantified. We do not treat uncertainty as a monolithic scalar but decompose it into a structured, multi-dimensional asset. The estimation process for each dimension is as follows, with full mathematical formalizations available in Appendix C.1:

- **Perceptual Uncertainty** ($u_{\text{perc}}$)**:** This dimension quantifies the model's confidence in recognizing raw visual signals (e.g., object categories, attributes). It is estimated by statistically analyzing the model's raw outputs. For instance, for an object classification task, $u_{\text{perc}}$ can be calculated using the **Shannon Entropy** of the predicted probability distribution over

possible object classes. A higher entropy signifies greater uncertainty about what is being perceived.

- **Semantic Uncertainty** ($u_{\mathbf{sem}}$)**:** This dimension measures ambiguity in understanding the deeper meaning of a scene, including the relationships between objects and their context. It is estimated by quantifying the multiplicity of plausible interpretations. For example, if a model identifies several valid potential relationships between two objects in an image, the semantic uncertainty is considered higher.

- **Inferential Uncertainty** ($u_{\mathbf{inf}}$)**:** This dimension assesses the model's confidence in making predictions or drawing conclusions based on the available information. Its estimation combines two factors: (1) the confidence in the single most likely outcome (i.e., $1 - \max(p)$), and (2) the overall dispersion of the entire predictive probability distribution (i.e., its entropy). This captures both the model's conviction in its top guess and its certainty across all possibilities.

### L.3    Explanation of Key Elements in Figure 3

We clarify two components from the architectural diagram in Figure 3 (page 4) that were previously ambiguous: the "Value Model" and the "Too Many?" label.

- **The "Value Model":** This component serves as the **reward signal generator** for our Multi-Armed Bandit (MAB) agent selection mechanism. After the selected agents produce a final "Output Text," the Value Model evaluates the quality of this output (e.g., by comparing it against a ground-truth answer or using a pretrained reward model). The result of this evaluation is a quantitative reward signal (labeled "Policy Selection Reward") that is fed back to the MAB. This reward is essential for the MAB's learning process, allowing it to update its policy (per the Thompson Sampling update rule in Appendix A) and improve its ability to select high-performing agents in the future.

- **The "Too Many?" Label:** This label in the "Uncertainty Evaluation Center" is a visual representation of the crucial **receiver capacity constraint check** within our trading protocol. Before an uncertainty trade is executed, the system must verify that the receiving agent will not be overloaded. This corresponds directly to the feasibility condition in Equation (5): $U_j(t) + T_{ij}(t) \leq C_j(t)$. The "Too Many?" check ensures that a proposed trade is rejected if accepting the new uncertainty packet $T_{ij}(t)$ would push the receiving agent's total uncertainty portfolio $U_j(t)$ beyond its operational capacity limit $C_j(t)$. It is a fundamental admission control mechanism that maintains system stability and agent effectiveness.

## M    Case Analysis

In this section, we present a series of case studies, including two successful and two unsuccessful examples, to demonstrate how multiple experts collaboratively analyze images in response to corresponding questions. The expert configuration comprises three analysis experts: an Object Recognition Expert, a Scene Description Expert, and a Text/OCR Analysis Expert. For each expert, we report both their analytical response and the associated uncertainty score. To improve clarity and conciseness, especially given the length of the responses, key excerpts are highlighted using colored underlines.

### M.1    Successful Case

tables 10 and 11 illustrate that our method, by assigning clearly defined roles to each expert—namely object recognition, scene understanding, and text/OCR analysis—enables comprehensive analysis across multiple modalities and semantic dimensions of the input. This structured task decomposition enhances both the depth and breadth of information processing, allowing each analysis expert to specialize in a distinct sub-task and generate high-quality outputs accompanied by uncertainty estimates. A principal advantage of this approach is its explicit quantification of uncertainty, which allows the system to weigh and prioritize expert contributions based on their reliability. This multi-expert architecture markedly improves the system's performance with respect to factual accuracy,

contextual completeness, and logical consistency, thereby enhancing robustness, interpretability, and overall stability across diverse question types and input formats.

## M.2 Unsuccessful Case

Although our multi-expert analytical framework performs effectively and robustly in the majority of cases, certain challenges and limitations remain.table 12 exemplifies a limitation of our multi-expert analytical framework when faced with ambiguous spatial and perceptual cues that demand deeper three-dimensional reasoning and contextual inference beyond straightforward visual and textual recognition. While the Object Recognition Expert identifies two square sliders labeled "A" and suggests visual similarity in size, and the Scene Description Expert emphasizes the schematic nature of the diagrams featuring differing inclined plane shapes, the Text/OCR Analysis Expert rightly notes that size equivalence cannot be confirmed solely based on the visual and textual evidence.

Despite this inherent uncertainty, the final system output incorrectly asserts that the sliders are the same size. This exposes a critical shortcoming: the current framework lacks an advanced spatial reasoning module capable of integrating geometric perspective and resolving scale ambiguities inherent in 2D schematic depictions of 3D objects. Furthermore, the system does not adequately leverage uncertainty quantification to withhold or qualify conclusions in cases of inconclusive or conflicting evidence.

This failure highlights the framework's overreliance on superficial visual similarity and label matching without robust geometric or physical reasoning. Consequently, it underscores the necessity of integrating more sophisticated reasoning components—such as 3D shape reconstruction, perspective analysis, or probabilistic inference over spatial configurations—to accurately assess relative object dimensions when explicit size information is unavailable.

In summary, this case illustrates that while the multi-expert system effectively parses and analyzes multimodal inputs, it remains limited in resolving ambiguities that require complex spatial cognition. Future work should focus on incorporating specialized reasoning capabilities to enhance accuracy in tasks involving comparative spatial judgments under uncertain visual conditions.

Table 13 presents another failure case that reveals a fundamental limitation of our multi-expert framework: the difficulty in performing reliable cross-modal reasoning when critical semantic associations cannot be directly inferred from visual cues. Although the Object Recognition, Scene Description, and Text/OCR Analysis Experts accurately identify visual features (e.g., yellow cylindrical containers), contextual settings (e.g., industrial storage area), and textual labels (e.g., liquid ammonia), the final output mistakenly claims that the object "has a boiling point of -33.3°C."

This error reflects a key deficiency: the current analysis experts lack the domain knowledge and reasoning mechanisms necessary to associate recognized objects (e.g., liquid ammonia) with their scientifically accurate properties. While -33.3°C roughly corresponds to the boiling point of gaseous ammonia, the precise boiling point of liquid ammonia is -33.42°C. More importantly, this physical property is not visually inferable from the image nor present in any text extracted by the OCR expert.

This case underscores a broader design limitation: the system assumes all factually relevant attributes can be inferred solely from image and text inputs, without access to external scientific knowledge bases or verification mechanisms. Consequently, it fails to differentiate between superficially plausible yet unsupported assertions and those grounded in the image evidence.

In conclusion, this example highlights the critical need to incorporate knowledge-grounded reasoning modules or external factual verification layers to bridge the gap between perceptual analysis and fact-based inference. This is particularly vital for tasks involving domain-specific scientific knowledge, where even minor factual inaccuracies can substantially undermine the system's credibility and reliability.

## Statement on the Use of AI Assistance

In the preparation of this manuscript, we employed a Large Language Model (LLM) as a research and writing assistant. The use of the LLM was restricted to two specific areas: (1) aiding in the initial phase of academic research by helping to survey and summarize relevant literature, and (2)

assisting in the post-writing phase by polishing the manuscript's language, grammar, and formatting to improve clarity and readability.

Table 10: Case: Successful Expert Collaboration Example 1

**Image**

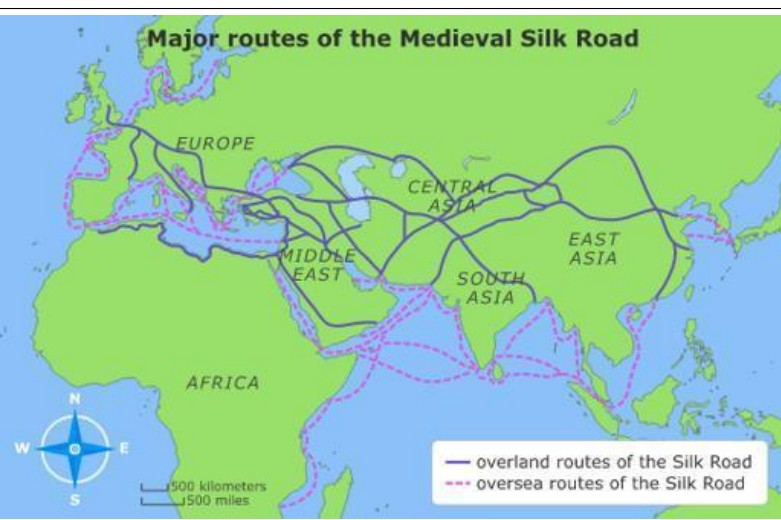

**Question**

Based on the map, what was true about the Silk Road around the year 1300 CE?

| Expert | Uncertainty | Output |
|---|---|---|
| Object Recognition | 0.1365 | 1.The image depicts solid blue lines and dashed pink lines, representing the overland Silk Road routes and maritime Silk Road routes, respectively. 2.A high concentration of solid blue lines is observed in Central Asia, indicating its role as a major hub for overland trade. 3.Dashed pink lines extend from the eastern coast of Africa through the Middle East to East Asia, clearly illustrating key maritime trade corridors. |
| Scene Description | 0.1096 | 1. The map covers an extensive geographical area, including Europe, Central Asia, the Middle East, South Asia, East Asia, and parts of Africa. 2.Central Asia is depicted as a pivotal intersection where multiple routes converge, highlighting its strategic position within the Silk Road network. 3.The Middle East functions as a critical junction connecting overland and maritime routes, serving as a key intermediary between East and West. 4.The map employs a simple and coherent color scheme and layout, facilitating clear interpretation of regional locations and route distributions. |
| Text/OCR Analysis | 0.0832 | 1. The map includes clearly legible labels for major regions such as Europe, Central Asia, the Middle East, South Asia, and East Asia. 2.The legend explicitly defines the solid blue lines and dashed pink lines as distinct types of trade routes. 3.The overall textual arrangement is well-organized, enhancing comprehension and efficient extraction of information from the map. |

**Answer**

The correct answer is: The Silk Road connected parts of East Asia, the Middle East, and Europe.

Table 11: Case: Successful Expert Collaboration Example 2

**Image**

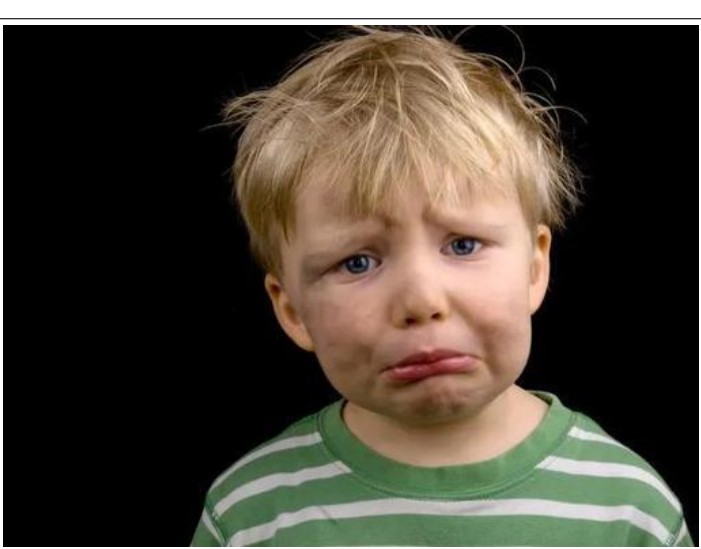

**Question**

What will happen next?

| Expert | Uncertainty | Output |
|---|---|---|
| Object Recognition | 0.8854 | 1.The image shows the child's expression suggesting sadness or distress, characterized by a downturned mouth and teary eyes, which strongly signals distress. 
 2. Although predicting future events from a still image is speculative, the child's emotional state is a significant indicator. |
| Scene Description | 0.8945 | 1.The child's somber expression combined with the stark black background highlights an emotional moment and accentuates the child's emotional vulnerability. 
 2.It is possible that the child might be comforted, potentially leading to interaction or conversation. |
| Text/OCR Analysis | 0.2454 | 1.No supplementary contextual text is provided, keeping the focus on the child's emotional expression. |

**Answer**

Based on the child's visible sadness, the most likely immediate reaction is that the child might cry.

Table 12: Case: Unsuccessful Expert Collaboration Example 1

**Image**

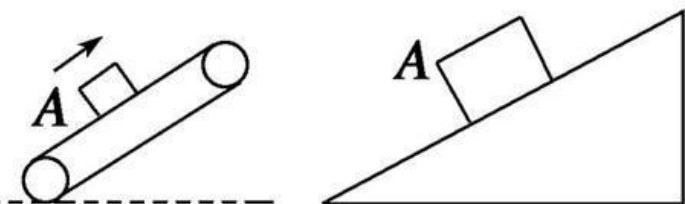

**Question**

There are two physical models in the picture, are the two square sliders the same size?

| Expert | Uncertainty | Output |
|---|---|---|
| Object Recognition | 0.1479 | 1. The image contains two physical models, each with a square slider labeled "A". 2. The visual evidence indicates the sliders are identical in size, occupying similar proportions on their inclined planes. |
| Scene Description | 0.1744 | 1. The image shows two models featuring square sliders placed near the top of their respective inclined planes—one cylindrical, one triangular. 2. The drawing is schematic with clear geometric shapes and no detailed environment or lighting, focusing on spatial relationships. |
| Text/OCR Analysis | 0.5382 | 1. Each diagram has a square labeled "A," indicating the same object or type. However, due to differing shapes and perspective, it is not possible to confirm the sliders' sizes solely from the image. 2. The visual text evidence does not conclusively verify identical sizes. |

**Answer**

The most reasonable conclusion is that the sliders are the same size.

Table 13: Case: Unsuccessful Expert Collaboration Example 2

**Image**

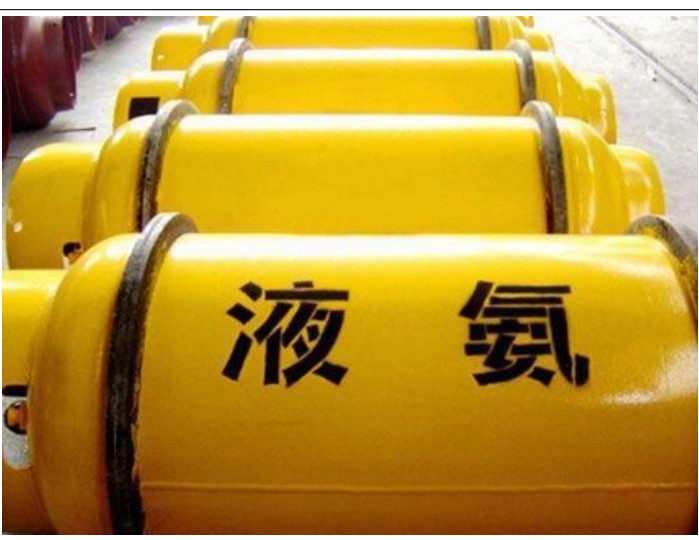

**Question**

The object is shown in this figure:

| Expert | Uncertainty | Output |
|---|---|---|
| Object Recognition | 0.2325 | The image depicts multiple yellow cylindrical gas cylinders arranged in a row, exhibiting regular shapes with rounded ends. 2.The cylinders are aligned horizontally with orderly spacing and consistent color and shape, indicating industrial gas containers. |
| Scene Description | 0.5276 | 1.The cylinders are positioned on a concrete floor, suggesting an industrial storage or transportation environment. 2.Additional similar containers are visible in the background, emphasizing a storage or loading area. 3.Lighting is uniform and bright, accentuating the yellow cylinders and black inscriptions. 4.The cylinders are arranged neatly with even spacing, reflecting organizational order and safety considerations. 5.The overall atmosphere is utilitarian, highlighting the practical and safety-focused nature of the industrial setting. |
| Text/OCR Analysis | 0.0832 | 1.The side of the cylinders clearly displays bold black lettering indicating "liquid ammonia" in Chinese, which stands out against the yellow background and ensures high legibility. 2.The text's placement and size are precise, indicating the contents as liquid ammonia and conforming to industrial labeling standards. |

**Answer**

The object shown in this figure has a boiling point of -33.3°C.

