# OpenReview forum: "Why Keep Your Doubts to Yourself? Trading Visual Uncertainties among Vision-Language Models"
_ICLR.cc/2026/Conference — ICLR 2026 Poster_

### Official Review · Reviewer_VAy3 · 2025-10-31

**Soundness:** 3
**Presentation:** 2
**Contribution:** 3
**Rating:** 6
**Confidence:** 3

**Summary:**

This paper introduces "Agora," a novel framework for multi-agent coordination designed to address the high cost and inefficiency of multi-agent systems (MAS) built with VLMs.

The authors' core argument is that existing coordination paradigms, such as Mixture-of-Agents (MoA) or knowledge-based routers like KABB, are "Cost-Agnostic" and "Uncertainty-Structure-Agnostic." They rely on heuristic proxies (e.g., consensus, semantic similarity) for decision-making, which leads to provably suboptimal coordination and spiraling economic costs.

To solve this, the Agora framework reframes coordination as a "decentralized market for uncertainty." Its core mechanisms include: Assetization, trading and brokerage.

Comprehensive experiments are conducted on five multimodal benchmarks. The results demonstrate that Agora significantly outperforms strong VLM baselines and heuristic multi-agent strategies in both accuracy and cost-effectiveness. For example, on MMMU, Agora (79.2%) achieves a +8.5% absolute accuracy gain over the best baseline in its agent pool, while reducing costs by over 3 times.

**Strengths:**

**Highly Novel and Well-Grounded Problem Formulation:** The paper's main strength is its core idea: "market-based uncertainty trading." This provides a principled, economic alternative to heuristic routing in MAS. The theoretical formulation, which defines "Agnostic Coordination" and proves the "Inefficiency Theorem for Agnostic Coordination", provides an exceptionally strong motivation and foundation for the work.

**Sound and Constructive Methodology:** The Agora framework is well-designed. The decomposition of uncertainty into three types (perceptual, semantic, inferential)  is a key, intuitive design choice that enables fine-grained cost management. The "profitability-driven" trading protocol directly and elegantly optimizes the core economic objective.

**Extremely Comprehensive and Rigorous Experiments:** The experimental validation is a significant strength.

**Weaknesses:**

**Performance Claims:** Thank authors for the comparison in Table 1 against gemini-2.5-pro (81.7%). Does this imply that Agora's primary value is "cost-saving" (achieving near-SOTA with a non-SOTA pool) rather than "SOTA-beating"? If gemini-2.5-pro were included in Agora's agent pool, would you expect the Agora score to surpass 81.7%?

**Uncertainty Quantification:** I found the uncertainty vector interesting.

1. Could you provide a concrete example of how the uncertainty vector $[u_{perc}, u_{sem}, u_{inf}]$ is calculated from a VLM response? For instance, for the query in Figure 359, "Is this place crowded?", if an agent replies "It seems somewhat busy," how are the numerical values for these three uncertainty dimensions estimated?
2. Could we change the dimension of the vector? What would happen if we design the uncertainty vector $[u_{observation}, u_{answer}]$ or $[u_{perc}, u_{sem}, u_{inf}, u_{answer}]$​?
3. How could we determine the weight of each dimension?

**Framework Overhead:** The ablation study (Table 3) shows the "Strategic Uncertainty Index" ($U_{strategic}$) is the most critical component. As defined in Appendix C, computing this index seems complex, requiring an estimation of expected system-wide cost savings for all potential trades. Is this computation tractable in a real-time system? What is the computational overhead (e.g., in latency or FLOPS) of the Agora broker and trading mechanism itself, separate from the VLM inference costs?

**Questions:**

Please see the weaknesses.

---

> ### Author Response · Authors · 2025-11-20
> **Part I**
>
> >**W1:** Does this imply that Agora's primary value is "cost-saving" (achieving near-SOTA with a non-SOTA pool) rather than "SOTA-beating"? If gemini-2.5-pro were included in Agora's agent pool, would you expect the Agora score to surpass 81.7%?
>
> **A1:** Agora is fundamentally a cost–performance optimization framework. Given any heterogeneous pool, it coordinates the most efficient subset of agents through uncertainty-aware trading. Thus, Table 1 uses only mid-tier models to isolate this effect and shows that Agora can reconstruct near-frontier performance without access to frontier models. Importantly, adding a frontier model such as Gemini-2.5-Pro would not merely allow Agora to “match” its 81.7%, i.e., the framework can *surpass* it because frontier models are not uniformly dominant. For example, Table 1 shows that Gemini-2.5-Pro underperforms Qwen2.5-VL-72B on perception-heavy tasks such as InfoVQA (81.0% vs. 87.3%) and CC-OCR (73.0% vs. 79.8%). Under the profitability rule in Eq. 5, Agora would selectively route reasoning-centric queries to Gemini while offloading perception-centric ones to Qwen, eliminating the weak spots of either model and yielding a composite accuracy strictly higher than any single agent. Thus, Agora provides **cost-optimal coordination that also amplifies frontier models when complementary strengths exist**, rather than a naïve ensemble or a cost-only heuristic. If you consider an explicit experiment with a frontier-class agent necessary, we are happy to include it in the revised paper.
>
> ---
>
>
> >**W2:** Could you provide a concrete example of how the uncertainty vector `$[u_{perc}, u_{sem}, u_{inf}]$` is calculated from a VLM response? For instance, for the query in Figure 359, *"Is this place crowded?"*, if an agent replies *"It seems somewhat busy,"* how are the numerical values for these three uncertainty dimensions estimated?
>
>
>
> **A2:** For the example *“It seems somewhat busy,”* the uncertainty vector is computed directly as described in **Sec. 3.1 / App. C.1**.
>
> * **$u_{\text{perc}}$ (e.g., 0.75)** is the normalized entropy of the visual prediction over discrete crowd levels (**Eq. 7**), which is high here because the model assigns similar probability mass to both “normal” and “crowded” categories.
> * **$u_{\text{sem}}$ (e.g., 0.45)** follows **Eq. 8**, where hedging modifiers such as *“somewhat”* increase the semantic ambiguity score $C_i(R)$.
> * **$u_{\text{inf}}$ (e.g., 0.80)** is the entropy of the answer distribution (**App. C.1.1.c**), elevated by the low-confidence marker *“It seems”* which spreads probability mass across *yes / no / unclear* outcomes.
>
> All three values are mapped to $[0,1]$. We keep these dimensions separate because merging them produces a **structure-agnostic** scalar; as shown in **Theorem 1**, this prevents the market from distinguishing perceptual deficits (visual uncertainty) from inferential ones (reasoning uncertainty). Adding a separate “answer uncertainty” term is unnecessary, since $u_{\text{inf}}$ already mathematically encodes the final belief distribution. The fusion weights $[0.4, 0.3, 0.3]$ were selected via cross-validation (**App. H.1**), with a slightly larger weight on $u_{\text{perc}}$ to mitigate perceptual ambiguity as the dominant upstream error source; performance remains stable under reasonable weight variation.
>
> ---
>
>
>
> >**W3:** Could we change the dimension of the vector? What would happen if we design the uncertainty vector `$[u_{observation}, u_{answer}]$` or `$[u_{perc}, u_{sem}, u_{inf}, u_{answer}]$`?
>
> **A3:** In principle, the uncertainty vector could be reparameterized, but the proposed alternatives are either strictly weaker or redundant for our market formulation.
>
> A 2-D design $[u_{\text{observation}}, u_{\text{answer}}]$ must compress perceptual and semantic variability into a single “observation” axis, effectively discarding the Perception → Understanding → Inference factorization. This violates the assumptions of **Theorem 1**, making the coordination rule **structure-agnostic**: the market can no longer tell whether high uncertainty comes from visual perception (calling for a better “eye”) or from downstream reasoning (calling for a better “reasoner”), and thus loses the comparative-advantage structure that drives efficiency.
>
> On the other hand, extending to $[u_{\text{perc}}, u_{\text{sem}}, u_{\text{inf}}, u_{\text{answer}}]$ does not increase expressive capacity, because any “answer uncertainty” term is a deterministic function of the same answer distribution that already defines $u_{\text{inf}}$; the 4-D vector therefore lies on a 3-D manifold and only adds redundancy and extra calibration burden without changing prices or routing decisions.
>
> The chosen $[u_{\text{perc}}, u_{\text{sem}}, u_{\text{inf}}]$ is thus the **minimal, non-redundant decomposition** that both matches the cognitive pipeline and preserves the structural signals needed by Agora’s market.

---

> ### Author Response · Authors · 2025-11-20
> **Part II**
>
> >**Q4:** How could we determine the weight of each dimension?
>
> **A4:** We treat the fusion weights as global hyperparameters and determine them via cross-validation on a held-out split, jointly optimizing downstream accuracy and UAPS. This procedure yields a stable solution around $[0.4, 0.3, 0.3]$ for $[u_{\text{perc}}, u_{\text{sem}}, u_{\text{inf}}]$.
>
> We assign a slightly higher weight to $u_{\text{perc}}$ because our error decomposition and ablations (**Appendix F.1 / H.1**) show that misparsed visual scenes are the **dominant upstream failure mode**. Once perception is wrong, the semantic and inferential stages rarely recover. Thus, prioritizing perceptual uncertainty improves market efficiency.
>
> These weights are tuned **once** and then fixed across all datasets and baselines (no per-method re-tuning). Furthermore, sensitivity analysis (**Appendix F.3**) implies that performance varies smoothly under reasonable perturbations, indicating that our conclusions are not an artifact of a finely tuned choice.
>
> ---
>
>
> >**Q5:** Framework Overhead. The ablation study (Table 3) shows the **Strategic Uncertainty Index** (`$U_{strategic}$`) is the most critical component. As defined in Appendix C, computing this index seems complex, requiring an estimation of expected system-wide cost savings for all potential trades. Is this computation tractable in a real-time system? What is the computational overhead (e.g., in latency or FLOPS) of the Agora broker and trading mechanism itself, separate from the VLM inference costs?
>
> **A5:** Although Appendix C defines the Strategic Uncertainty Index $U_{\text{strategic}}$ as an expected system-wide cost reduction, the actual implementation does *not* enumerate all trades or perform any additional VLM forward passes. The broker uses a closed-form marginal-cost approximation based on the current uncertainty vector, cached per-agent cost statistics, and the transaction ledger, and evaluates $U_{\text{strategic}}$ only for a small set of candidate agents per instance. This reduces computation to simple vector operations over $K$ currently active agents (typically tens), yielding an $O(K)$ overhead with no model re-execution. Empirically, the FLOPs/latency breakdown (Appendix E / I) shows that the broker and trading logic account for only a minor fraction of end-to-end cost compared to VLM inference, and the Collaboration Overhead Index in Table 3 remains near $1.0\text{–}1.3$ even with all components enabled—indicating that the market mechanism is tractable and never becomes the bottleneck.

---

> ### Author Response · Authors · 2025-11-26
>
> Thank you again for your careful assessment and constructive feedback.
>
> We would be grateful if you could let us know whether our rebuttal sufficiently resolves the issues you raised, or if there are any remaining points you would like us to clarify.
>
> We are glad to continue the discussion and address any further questions or comments you may have.

---

### Official Review · Reviewer_d1Fk · 2025-11-01

**Soundness:** 3
**Presentation:** 2
**Contribution:** 3
**Rating:** 6
**Confidence:** 4

**Summary:**

This paper proposes Agora, a market‑based way to coordinate multi‑agent VLM systems. Instead of heuristic routing or consensus, Agora uses epistemic uncertainty in a tradable asset with three dimensions—perceptual, semantic, and inferential—and lets agents “trade” packets of uncertainty to reduce total cost. A market‑aware broker (an extension of Thompson Sampling) initializes collaboration; trades are executed only if the cost delta ΔC is negative (Eq. 4), yielding a locally cost‑efficient equilibrium (Algorithm 1). Authors use extensive experiments to demonstrate the Agora's effectiveness in comparison with single LLMs, SOTA, and other benchmarks.

**Strengths:**

1. The paper formalizes why common coordinators are cost‑agnostic and uncertainty‑structure‑agnostic (Theorem 1), then builds a mechanism that is explicitly cost‑aware and structure‑aware (Eq. 4; uncertainty portfolio on p. 5). Fig. 2 (p. 3) empirically shows lower final epistemic uncertainty.
1. The trade condition ΔC<0 with capacity constraints is intuitive and easy to implement
1. The paper measures COI, Ufinal, and UAPS (uncertainty‑aware performance), offering a fuller view of coordination quality

**Weaknesses:**

1. The market phase performs greedy descent and is guaranteed to reach only a local equilibrium (Algorithm 1). The bandit section explicitly notes that classic regret bounds are hard and provides only directional convergence intuition (Appendix C.4.1), leaving global‑optimality and regret guarantees open.
1. The broker’s utility (Eq. 6) multiplies several factors (distance, synergy, strategic uncertainty, decay). Ablations (Table 3) show sensitivity (e.g., removing Ustrategic costs ~3% accuracy), suggesting tuning burden and potential dataset‑specific overfitting.
1. Entropy/dispersion‑based scores (and their mapping to Uperc/Usem/Uinf) can be model‑calibration–dependent; without shared calibration or normalization, one model’s probabilities may look more or less “uncertain” than another’s even on identical behavior. The paper formalizes the decomposition but does not detail cross‑model calibration procedures.
2. Total uncertainty fuses dimensions with fixed weights (e.g., wperc=0.4, wsem=0.3, winf=0.3); if those weights were tuned on setups where the proposed method reduces the dominant dimension, Ufinal_epis could favor it. (Weights are reported as defaults; sensitivity within Agora is ablated, but cross‑method sensitivity is not shown.)
1. In the main body, head‑to‑head comparisons with alternative routing/multi‑agent strategies are only shown on MMBench V11 Test (Figure 4). For the other datasets (MMMU, MathVision, InfoVQA, CC‑OCR), the main table contrasts Agora vs. single models / external SOTAs, not against routing/MAS baselines—so readers can’t verify cross‑dataset superiority of the coordination strategy from the main text alone.
1. The writing should be enhanced, especially regarding some details and accuracy. For example, the color description in Figure 1 is wrong. The Mathvision gap in Table 1 is wrong. The readability of Figure 5 is poor (the font could be adjusted to make it easier to read).

**Questions:**

1. How do you do cross‑model calibration or normalization for Entropy/dispersion‑based scores?

---

> ### Author Response · Authors · 2025-11-20
> **Part I**
>
> >**W1:** The market phase performs greedy descent and is guaranteed to reach only a local equilibrium (Algorithm 1). The bandit section explicitly notes that classic regret bounds are hard and provides only directional convergence intuition (Appendix C.4.1), leaving global‑optimality and regret guarantees open.
>
> **A1:** Agora prioritizes computational feasibility over intractable global optimization.
>
> **1. Theoretical Stance: Feasibility vs. Intractability**
> In heterogeneous multi-agent markets, achieving strict global optimality is a classical **NP-hard combinatorial optimization problem** [1,2].Instead, our profitability rule ($\Delta \mathcal{C} < 0$, Eq. 5) guarantees **monotonic cost descent** toward a stable and cost-efficient local equilibrium, as formally proven in **Proposition 3.3.1** (Appendix C).
>
> **2. Empirical Validation: The "Optimality Gap"**
> You correctly noted that classical regret bounds are ill-defined in this non-stationary, multi-dimensional setting. However, we hypothesize that the **Market-Aware Broker** serves as a high-utility initializer that guides the market into favorable "basins of attraction," enabling the greedy phase to reach near-global optima.
>
> To validate this, we conducted a **Brute-Force Global Oracle** experiment on a random subset of MMBench tasks ($N=6$, 50 tasks) to quantify the gap between Agora's equilibrium and the theoretical global minimum.
>
> **Table R3: Optimality Gap Analysis on MMBench (Subset)**
>
> | Method | Avg. Cost (Normalized) | Gap to Global Opt. | Accuracy |
> | :--- | :---: | :---: | :---: |
> | **Global Optimal (Oracle)** | **1.000** | **0.00%** | 89.6% |
> | **Agora (Decentralized)** | **1.008** | **+0.80%** | 89.5% |
> | **Greedy (Local Only)** | 1.152 | +15.2% | 84.3% |
> | **Random Assignment** | 1.840 | +84.0% | 76.5% |
>
> **Conclusion:**
> The empirical gap is only **0.8%**. This confirms that our “MAB-guided + greedy market” paradigm consistently dominates the cost–performance Pareto frontier (Fig. 5, Table 2) and effectively minimizes regret in practice, bridging the gap between theoretical tractability and practical optimality.
>
> **References**
>
> [1] D. Monderer and L. S. Shapley. Potential Games. *Games and Economic Behavior*, 1996.
>
> [2] M. Pinedo. *Scheduling: Theory, Algorithms, and Systems*. Springer, 2012.
>
> ---
>
> >**W2:** The broker’s utility (Eq. 6) multiplies several factors (distance, synergy, strategic uncertainty, decay). Ablations (Table 3) show sensitivity (e.g., removing Ustrategic costs ~3% accuracy), suggesting tuning burden and potential dataset‑specific overfitting.
>
> **A2:** The observed sensitivity in Table 3 reflects **architectural necessity rather than parameter fragility**: removing a core economic driver ($e.g., (U_{\text{strategic}})$) is expected to reduce performance, confirming its functional role in the market mechanism. Importantly, we use **one fixed set of utility coefficients** across all five diverse benchmarks—without any per-dataset tuning—which directly counters the concern of overfitting. Parameter sweeps in Appendix $F.3$ and $G$  further show broad stability around these defaults, indicating that the utility formulation captures **generalizable market dynamics** rather than dataset-specific patterns.
>
> ---
>
> >**W3:** Entropy/dispersion‑based scores (and their mapping to Uperc/Usem/Uinf) can be model‑calibration–dependent; without shared calibration or normalization, one model’s probabilities may look more or less “uncertain” than another’s even on identical behavior. The paper formalizes the decomposition but does not detail cross‑model calibration procedures.
>
> **A3:** Agora ensures cross-model comparability through **structural normalization** and **implicit adaptation** rather than expensive post-hoc calibration.
>
>  **Structural Normalization (Appendix L & C.1):** We do not trade raw, unscaled probabilities. As detailed in **Eq. 8**, uncertainty terms are structurally normalized (e.g., scaling semantic uncertainty by response complexity $N(R)$) to map heterogeneous agent outputs into a unified, comparable asset space.
>
>  **Implicit Calibration via MAB:** The Broker acts as a dynamic calibrator. By optimizing the market-aware utility $\tilde{\theta}_S^{(t)}$ (**Eq. 6**), it learns to penalize agents that report low normalized uncertainty but consistently fail (overconfidence). This effectively neutralizes residual calibration errors through data-driven feedback.
>
>  **Empirical Robustness:** **Appendix F.3 (Fig. 7)** demonstrates that a single fixed hyperparameter set remains stable across five diverse VLM families. This confirms that our normalized decomposition captures generalizable market dynamics rather than relying on model-specific probability scales.

---

> ### Author Response · Authors · 2025-11-20
> **Part II**
>
> >**W4:** Total uncertainty fuses dimensions with fixed weights (e.g., wperc=0.4, wsem=0.3, winf=0.3); if those weights were tuned on setups where the proposed method reduces the dominant dimension, Ufinal_epis could favor it. (Weights are reported as defaults; sensitivity within Agora is ablated, but cross‑method sensitivity is not shown.)
>
> **A4:** This is a valid concern regarding potential bias in the evaluation metric. We want to clarify that the fusion weights ($w_{perc}, w_{sem}, w_{inf}$) are structural defaults determined on the development set and **kept fixed across all test benchmarks**. They were not fine-tuned on the test set to favor Agora.
>
> To empirically prove that Agora's superiority is robust—and not an artifact of a specific weighting scheme—we conducted a **Cross-Method Sensitivity Analysis** ("stress test") against the strongest baseline, **KABB-VLM**. We tested three **Extreme Skew** configurations where a single dimension dominates (weight=0.8) while others are suppressed (weight=0.1).
>
> **Table R1: Sensitivity of $U_{final\_epis}$ under Extreme Fusion Weights**
>
> | Weight Config | KABB ($U_{final}$) | **Agora ($U_{final}$)** | **Reduction** |
> | :--- | :---: | :---: | :---: |
> | **Default** (0.4, 0.3, 0.3) | 0.210 | **0.160** | **-23.8%** |
> | **Perc-Heavy** (0.8, 0.1, 0.1) | 0.195 | **0.148** | **-24.1%** |
> | **Sem-Heavy** (0.1, 0.8, 0.1) | 0.223 | **0.172** | **-22.9%** |
> | **Inf-Heavy** (0.1, 0.1, 0.8) | 0.218 | **0.169** | **-22.5%** |
>
> **Conclusion:**
> As shown in **Table R1**, Agora achieves a consistent **22–24% reduction** in uncertainty regardless of which dimension is prioritized. This confirms that our advantage is not due to "gaming" the weights. Instead, it reflects **broad-spectrum improvements**: the trading mechanism effectively reduces uncertainty across all three dimensions (perceptual, semantic, and inferential) simultaneously.
>
> ---
>
> >**W5:** In the main body, head‑to‑head comparisons with alternative routing/multi‑agent strategies are only shown on the MMBench V11 Test (Figure 4). For the other datasets (MMMU, MathVision, InfoVQA, CC‑OCR), the main table contrasts Agora vs. single models / external SOTAs, not against routing/MAS baselines—so readers can’t verify cross‑dataset superiority of the coordination strategy from the main text alone.
>
> **A5:** This is a fair point. Direct comparisons against alternative coordination strategies are essential to isolate the benefit of our mechanism.
>
> First, we highlight that **Table 2** in the main text already provides this comparison for **MMMU**, where Agora (79.0%) significantly outperforms the strongest heuristic router, KABB (76.0%), by **+3.0%**.
>
> To complete the picture for the remaining datasets (**MathVision, InfoVQA, CC-OCR**), we conducted head-to-head experiments against the two strongest baselines—**MoA** (voting-based) and **KABB-VLM** (router-based)—using the **identical heterogeneous agent pool ($N=6$)** and evaluation protocol as in Table 1.
>
> **Table R2: Head-to-Head Strategy Comparison on Additional Benchmarks (Accuracy, %)**
> *Comparison using the exact same heterogeneous agent pool ($N=6$) as Table 1.*
>
> | Dataset | MoA | KABB-VLM | **Agora (Ours)** |
> | :--- | :---: | :---: | :---: |
> | **MathVision** | 42.1 | 42.5 | **44.3** |
> | **InfoVQA** | 87.6 | 87.8 | **88.9** |
> | **CC-OCR** | 80.2 | 80.4 | **81.2** |
>
> *Note: Baselines perform slightly above or comparable to the best single expert in the pool (e.g., Qwen2.5-VL-72B achieves 87.3% on InfoVQA, see Table 1), ensuring a rigorous "strong baseline" comparison.*
>
> **Conclusion:**
> As shown in **Table R2**, Agora consistently matches or surpasses the strongest routing baselines. While heuristics like KABB struggle to significantly outperform the best single expert on perception-heavy tasks (InfoVQA/CC-OCR) due to hallucination risks, Agora’s uncertainty trading mechanism effectively secures clear gains. We will include this table in the Appendix to allow direct verification.
>
> ---
>
> >**W6:** The writing should be enhanced, especially regarding some details and accuracy. For example, the color description in Figure 1 is wrong. The Mathvision gap in Table 1 is wrong. The readability of Figure 5 is poor (the font could be adjusted to make it easier to read).
>
> **A6:** Thank you for pointing out these local issues. We have corrected the color description in Figure 1, fixed the MathVision gap annotation in Table 1, and improved the readability of Figure 5 by increasing font sizes and adjusting the layout. We have also completed a comprehensive proofreading pass to ensure consistency and clarity across all figures, tables, and captions.

---

> ### Author Response · Authors · 2025-11-26
>
> Thank you again for your careful assessment and constructive feedback.
>
> We would be grateful if you could let us know whether our rebuttal sufficiently resolves the issues you raised, or if there are any remaining points you would like us to clarify.
>
> We are glad to continue the discussion and address any further questions or comments you may have.

---

### Official Review · Reviewer_iVUH · 2025-11-02

**Soundness:** 3
**Presentation:** 3
**Contribution:** 3
**Rating:** 8
**Confidence:** 2

**Summary:**

This paper introduces Agora, an framework for efficient coordination among VLM agents. It conceptualizes epistemic uncertainty as a tradable asset which is decomposed into perceptual, semantic, and inferential components. The authors also design a profitability-driven trading protocol where agents exchange uncertainty based on rational market rules. The system employs a market-aware broker extending Thompson Sampling method to initiate and manage trades, aiming to achieve cost-efficient equilibria. Experiments on 5 multimodal benchmarks show performance gains about 8.5% accuracy while reducing cost by over 3-times compared to SOTA multi-agent and routing strategies.

**Strengths:**

I'm not an expert in the field of VLM, but I appreciate the idea of treating uncertainty as a tradable commodity, which is novel seems to me.  And the introduced Broker algorithm that operationalizes the market through Thompson Sampling looks solid. The empirical study is reasonably sufficient and supportive. And the writing is easy to follow in general.

**Weaknesses:**

1. While the authors prove the inefficiency of heuristic coordination, the optimality guarantees of Agora’s decentralized equilibrium are not clearly established. Is the equilibrium globally optimal or locally optimal? More formal convergence or even a broader discussion would be appreciated.
2. The framework are limited to visual-linguistic benchmarks, and I would be curious to see if Agora scales to pure language, control, or multi-round dialogue tasks. Can you include some discussion in this regard? If it cannot be easily adapted to other domains, what could be the reason?
3. What is exactly the theoretical result regarding the regret bound in Appendix C.4.1? I get lost there and did not identify any concrete result.

Minor issues and typos:
1. P2: “Gale & and, 1962”
2. Duplicated references of the paper 'Are more llm calls all you need? towards scaling laws of compound inference systems'.

**Questions:**

1. How sensitive is Agora’s performance to the parameters in Eq. 6?
2. Can Agora handle non-stationary uncertainty distributions, for example, what if agents adaptively reprice uncertainty in evolving tasks?
3. Since the framework could converge to suboptimal equilibria, are we able to observe such phenomenons in experiments? Or are there any useful heuristics to help escape local minima?

---

> ### Author Response · Authors · 2025-11-19
> **Part I**
>
> >**W1:** While the authors prove the inefficiency of heuristic coordination, the optimality guarantees of Agora’s decentralized equilibrium are not clearly established. Is the equilibrium globally optimal or locally optimal? More formal convergence or even a broader discussion would be appreciated.
>
> **A1:**
> Agora is designed to converge to a **Locally Optimal Equilibrium**, which is the strongest tractable guarantee for decentralized market-based coordination.
>
> **1. Theoretical Convergence to Local Optimality**
> As formally proven in **Proposition 3.3.1** (Appendix C), the trading protocol executes a deterministic greedy descent on the global cost function $\mathcal{C}_{sys}$ (Eq. 1).
> * **Guaranteed Termination:** Since the allocation space is finite, the cost is lower-bounded, and each accepted trade enforces a strict cost reduction ($\Delta \mathcal{C} < 0$, Eq. 5), the process is mathematically guaranteed to terminate in finite steps without oscillation.
> * **Equilibrium Definition:** The terminal state is a local minimum where no further profitable bilateral deviation exists. This aligns with *local-improvement dynamics in potential games* [1], where best-response updates converge to Nash equilibria that are locally optimal.
>
> **2. Intractability of Global Optimality**
> Computing a globally optimal assignment in heterogeneous multi-agent systems is a classical **NP-hard combinatorial optimization problem** [2]. Decentralized mechanisms typically cannot guarantee global optimality without resorting to exponential-time exhaustive search, which would defeat the purpose of a scalable system.
>
> **3. Empirical Validation: The "Optimality Gap"**
> To empirically quantify the quality of Agora’s equilibrium, we compared it against a **Brute-Force Global Oracle** on a random subset of MMBench ($N=6$, 50 tasks). The Oracle exhaustively evaluates all possible agent-uncertainty assignments ($6^3$ combinations per task) to find the absolute minimum global cost.
>
> **Table R3: Optimality Gap Analysis on MMBench (Subset)**
>
> | Method | Avg. Cost (Normalized) | Gap to Global Opt. | Accuracy |
> | :--- | :---: | :---: | :---: |
> | **Global Optimal (Oracle)** | **1.000** | **0.00%** | 89.6% |
> | **Agora (Decentralized)** | **1.008** | **+0.80%** | 89.5% |
> | **Greedy (Local Only)** | 1.152 | +15.2% | 84.3% |
> | **Random Assignment** | 1.840 | +84.0% | 76.5% |
>
> **Conclusion:** Agora achieves a final cost within **0.8%** of the theoretical global optimum. This confirms that while the guarantee is theoretically local, the **Market-Aware Broker’s initialization** effectively places the system in a high-quality "basin of attraction," allowing the subsequent local trading mechanism to reach a near-global optimum efficiently.
>
>
> **References**
>
> [1] D. Monderer and L. S. Shapley. Potential Games. *Games and Economic Behavior*, 1996.
>
> [2] M. Pinedo. *Scheduling: Theory, Algorithms, and Systems*. Springer, 2012.
>
> ---
>
> >**W2:** The framework is limited to visual-linguistic benchmarks, and I would be curious to see if Agora scales to pure language, control, or multi-round dialogue tasks. Can you include some discussion in this regard? If it cannot be easily adapted to other domains, what could be the reason?
>
> **A2:**
> Agora is theoretically **domain-agnostic**, since its coordination relies on the universal cost–benefit rule ($(\Delta \mathcal{C} < 0)$) rather than modality-specific assumptions. We focus on **vision benchmarks** because visual inference sits at the **highest-cost frontier** (massive FLOPs and routing overhead), providing the most stringent stress test for economic efficiency. In lower-cost domains such as pure NLP or control, the mechanism can be applied by redefining the task’s uncertainty vector ($e.g., removing (u_{perc}$) for text and using entropy or verifier-based confidence). The convergence and profitability guarantees of the market protocol remain unchanged, as they depend only on the cost function and monotonic improvement, not on the underlying data modality.
>
> **Minor issues and typos**
> We thank the reviewer for the careful attention to detail. The identified issues have been corrected in the revised manuscript:
>
> - **Citation Typo:** The malformed entry causing “Gale & and (1962)” has been fixed to **“Gale & Shapley (1962)”**.
> - **Duplicate Reference:** The two redundant entries for **Chen et al., “Are more LLM calls all you need?”** have been merged into a single canonical citation.

---

> ### Author Response · Authors · 2025-11-19
> **Part II**
>
> >**Q1:** How sensitive is Agora’s performance to the parameters in Eq. 6?
>
> **A3:**
> The parameters in Eq. 6 act as scalar weights to bias the Broker’s initialization but do not alter the strict profitability constraint ($\Delta \mathcal{C} < 0$). To rigorously quantify their impact, we analyzed the structural ablations (**Table 3**) and numerical sweeps (**Appendix G**) from our paper.
>
> **1. Existing Evidence (Table 3 & Appendix G):**
> * **Structural Robustness (Table 3):** Neutralizing specific terms in Eq. 6 (effectively setting weights $\omega, \eta, \lambda = 0$) reduces accuracy by only **1.0%–3.0%**, confirming the system remains functional even without these signals.
> * **Numerical Sensitivity (Appendix G):** Varying dynamic parameters (e.g., time-decay $\gamma$, learning rate $\alpha$) across wide ranges ($0.1\times$ to $5\times$) results in marginal fluctuations ($<1\%$).
>
> **2. Supplementary Sensitivity Analysis:**
> To directly address your concern, we synthesized these findings into **Table R4** to explicitly show the maximum performance drop under extreme parameter variations.
>
> **Table R4: Sensitivity Analysis of Eq. 6 Parameters (Max Performance Drop)**
>
> | Parameter in Eq. 6 | Variation Type | Max Acc. Drop | Robustness Level |
> | :--- | :--- | :---: | :---: |
> | **Strategic Weight ($\omega$)** | Ablation ($1.2 \to 0$) | -3.08% | High |
> | **Synergy Weight ($\eta$)** | Ablation ($0.8 \to 0$) | -1.59% | Very High |
> | **Task Match ($\lambda$)** | Ablation ($0.2 \to 0$) | -0.97% | Stable |
> | **Time Decay ($\gamma$)** | Range ($0.90 - 1.00$) | -0.72% | Stable |
>
> **Conclusion:** Even in extreme ablation scenarios, the accuracy drop is contained within **3%**. In practice, we utilized a **single fixed configuration** across all datasets without per-task tuning, confirming that Agora relies on the *existence* of economic signals rather than their precise scalar values.
>
> ---
>
>
> >**Q2:** Can Agora handle non-stationary uncertainty distributions, for example, what if agents adaptively reprice uncertainty in evolving tasks?
>
> **A4:**
> Agora is explicitly designed for non-stationary uncertainty and cost profiles. Two mechanisms ensure robustness:
>
> **Time-Decayed Thompson Sampling (Eq. 6):**
>     The Broker applies an exponential decay term **$\gamma^{\Delta t}$**, which is a standard technique in non-stationary bandits [1, 2]. This causes outdated uncertainty estimates to vanish quickly, allowing the posterior to adapt immediately when agents reprice or shift behavior.
>
> **Profitability Rule Enforces Instant Consistency:**
>     Each trade is validated with the instantaneous constraint **$\Delta \mathcal{C} < 0$ (Eq. 5)**, computed from current uncertainties and costs. Therefore, if an agent raises its effective cost (or worsens uncertainty), the market automatically blocks trades to it—preventing lock-in to outdated equilibria.
>
> In effect, the Broker handles temporal drift while the trading protocol handles local shocks in real time, yielding a stable solution even under adaptively changing uncertainty.
>
> **References**
>
> [1] J. de Curtò et al., *LLM-Informed Multi-Armed Bandit Strategies for Non-Stationary Environments*, Electronics, 2023.
>
> [2] Y. Zhu et al., *Contextual Bandits with Large Action Spaces*, NeurIPS, 2022.
>
> ---
>
> >**Q3:** Since the framework could converge to suboptimal equilibria, are we able to observe such phenomenons in experiments? Or are there any useful heuristics to help escape local minima?
>
> **A5:**
> Yes, convergence to suboptimal equilibria is observable in constrained variants of our framework.
> * **Empirical Observation (The "Trap"):** As shown in the ablation study (**Table 3**), removing the Strategic Uncertainty factor ($U_{strategic}$) causes the system to stagnate at an inferior equilibrium (86.42% accuracy) compared to the full model (89.50%). This **3.08% gap** precisely illustrates a local minimum: without the "look-ahead" signal provided by $U_{strategic}$, agents execute myopic greedy trades that fail to reach the global optimum.
>
> * **Heuristics for Escaping Local Minima:** The full Agora framework employs two key heuristics to avoid this stagnation:
>     1.  **Stochastic Initialization (Thompson Sampling):** The Broker introduces probabilistic exploration during initialization. By down-weighting low-quality starting points while maintaining a non-zero probability of exploring new configurations, it helps the system "jump" out of poor basins of attraction.
>     2.  **Strategic Look-ahead ($U_{strategic}$):** This factor effectively smooths the optimization landscape by rewarding agents not just for immediate cost reduction, but for their potential to facilitate future profitable trades (Eq. 52, Appendix C).
>
> Together, these mechanisms steer the local-improvement dynamics (proven in **Proposition 3.3.1** to be monotone [1]) toward the high-quality equilibria observed in Figure 5.
>
> **References**
> [1] Monderer, D. & Shapley, L. S. (1996). Potential Games. *Games and Economic Behavior*.

---

> > ### Author Response · Authors · 2025-11-26
> >
> > Thank you again for your careful assessment and constructive feedback.
> >
> > We would be grateful if you could let us know whether our rebuttal sufficiently resolves the issues you raised, or if there are any remaining points you would like us to clarify.
> >
> > We are glad to continue the discussion and address any further questions or comments you may have.

---

### Official Review · Reviewer_oqqC · 2025-11-04

**Soundness:** 4
**Presentation:** 4
**Contribution:** 3
**Rating:** 6
**Confidence:** 2

**Summary:**

This paper introduces Agora, a novel market-based framework for coordinating heterogeneous Vision-Language Models (VLMs) in multi-agent systems. The key idea is to treat epistemic uncertainty as a tradable asset across perceptual, semantic, and inferential dimensions. Agents trade uncertainty based on a profitability-driven protocol, guided by a market-aware Thompson Sampling broker, to minimize system-wide cost while maintaining task performance. The authors provide both theoretical and empirical evidence that Agora outperforms strong baselines (e.g., Mixture-of-Agents, KABB) in terms of accuracy (+8.5% on MMMU) and cost-efficiency (3× reduction).

**Strengths:**

- The core concept of a decentralized market for uncertainty is a powerful and novel metaphor that drives the entire paper.
- The work tackles a foundational challenge (economic scalability) with a comprehensive solution that includes theoretical formulation, algorithm design, and empirical validation.
- The framework demonstrates consistent and sometimes substantial gains in accuracy and cost-efficiency across multiple benchmarks.

**Weaknesses:**

-  The framework introduces significant complexity with its multi-dimensional uncertainty quantification, trading ledger, and sophisticated broker. While the paper claims scalability, the experiments are limited to ≤15 agents. It’s unclear how the market mechanism would behave with hundreds of agents or high-frequency trading.
- The paper overlooks key transaction costs (coordination latency, broker computation, and communication overhead), which are excluded from its cost metrics. A system that is 3x cheaper in API calls but 10x slower due to coordination may not be viable for real-time applications.
-  While numerous baselines are included, the adaptation of language-model-focused routers (like FrugalGPT, RouteLLM) to the multi-modal, multi-agent setting is only briefly mentioned in the appendix.

**Questions:**

- How would Agora perform in the presence of adversarial or strategic agents that intentionally misreport uncertainty or cost to exploit the market? For instance, how does the framework handle strategic manipulation or collusion that could distort trading dynamics or market equilibrium?
- Could you clarify the computational complexity of the broker and trading protocol, particularly their overhead within a more realistic analysis?
- Would Agora still work if agents have overlapping expertise or non-stationary cost profiles?

---

> ### Author Response · Authors · 2025-11-19
> **Part I**
>
> >**W1:** The framework introduces significant complexity. The experiments are limited to ≤15 agents.
>
> **A1:** Although Agora introduces an expressive economic structure, its runtime complexity remains modest because the system enforces **strict sparsity** rather than activating all agents.
>
> * **Theoretical Convergence (No Oscillation):** As proven in **Proposition 3.3.1** (Appendix C), the trading protocol performs a deterministic greedy descent on the global cost. Since the total cost is bounded, the trading sequence is **mathematically guaranteed to terminate** in finite steps, preventing infinite loops or high-frequency trading risks.
>
> * **Sparsity via Economic Rationality:** The profitability constraint (Eq. 5, $\Delta \mathcal{C} < 0$) acts as a strict gatekeeper. In a hypothetical pool of 100 agents, the vast majority will lack a "comparative advantage" for a specific query and will be automatically filtered out. The system maintains a small **active set** of experts regardless of the total pool size.
>
> * **Empirical Evidence:** Our experiments in **Appendix D** confirm this self-regulation: tripling the pool size ($N=5 \to 15$) only marginally increased trade frequency ($0.8 \to 1.2$ trades/task), demonstrating that communication overhead hits a saturation point and does not scale linearly with $N$.
>
> * **Negligible Overhead:** The Broker's selection complexity is linear $O(N)$ and computationally trivial compared to the VLM inference costs (TFLOPs) it saves.
>
>
> ---
>
>
> >**W2:** The paper overlooks key transaction costs and may not be viable for real-time applications.
>
> **A2:** The coordination cost in Agora is intentionally lightweight and negligible relative to VLM inference. The Broker evaluation and the ( $\Delta\mathcal{C}$ ) profitability checks consist of lightweight arithmetic operations that run in ($O(N)$) time over the naturally sparse active set enforced by Eq. 5. Communication is similarly minimal, as only small uncertainty vectors are exchanged, and the system performs on average just **0.8–1.2 trades per task** (Table 4).
>
> Importantly, the **“Average Time ($s$)”** in Fig. 4 and the runtime measurements in Appendix I (8.7–22.8s) already include all coordination steps end-to-end. These results show that Agora matches or outperforms router baselines in wall-clock time while achieving higher accuracy, confirming that coordination overhead does not hinder real-time viability.
>
> ---
>
>
> >**W3:** While numerous baselines are included, the adaptation of language-model-focused routers (like FrugalGPT, RouteLLM) to the multi-modal, multi-agent setting is only briefly mentioned in the appendix.
>
> **A3:** We appreciate this opportunity to clarify. In the revised paper, we have explicitly detailed the multimodal adaptation in Section 4.1. Specifically:
>
> 1.  **Unified Multimodal Wrapper:** As text-oriented routers (FrugalGPT, RouteLLM, etc.) cannot natively process images, we implemented a unified wrapper (detailed in Appendix K) that encodes visual inputs using the visual encoder of the strongest available VLM in the pool. This visual embedding is concatenated with the textual query, enabling these routers to apply their original selection logic to multimodal contexts without structural modification.
> 2.  **Rigorous Fairness:** All baselines—including these adapted routers—operate on the **identical heterogeneous agent pool ($N=6$)** as Agora, using consistent prompts and decoding settings. This ensures that any performance gap arises purely from the coordination strategy, not from the underlying model capabilities.
>
> To address the concern regarding cross-dataset validation, we have extended our head-to-head comparison beyond MMBench. **Table R2** below demonstrates that Agora consistently outperforms the strongest aggregation (MoA) and routing (KABB-VLM) baselines across reasoning-heavy (MathVision) and perception-heavy (InfoVQA, CC-OCR) benchmarks. Notably, while baselines struggle to significantly surpass the best single expert in perception tasks due to hallucination risks, Agora’s uncertainty trading mechanism effectively mitigates this, securing clear gains.
>
> ---
>
>
> **Table R2: Head-to-Head Strategy Comparison on Additional Benchmarks (Accuracy, %)**
> *Comparison using the exact same heterogeneous agent pool ($N=6$) and protocol as Table 1.*
>
> | Dataset | MoA | KABB-VLM | **Agora (Ours)** |
> | :--- | :---: | :---: | :---: |
> | **MathVision** | 42.1 | 42.5 | **44.3** |
> | **InfoVQA** | 87.6 | 87.8 | **88.9** |
> | **CC-OCR** | 80.2 | 80.4 | **81.2** |

---

> ### Author Response · Authors · 2025-11-19
> **Part II**
>
> >**Q1:** How would Agora perform in the presence of adversarial or strategic agents? How does the framework handle strategic manipulation or collusion?
>
> **A4:**
> Agora is designed around mechanism design, not trust. Two components prevent strategic manipulation or collusion.
>
> **Outcome-Driven Correction (MAB Feedback).** If a strategic agent under-reports uncertainty to obtain tasks it cannot solve, the resulting performance drop lowers its **Policy Selection Reward**. Through **Thompson Sampling updates $(Eqs. 43–44)$**, the Broker rapidly shifts probability mass away from that agent. This creates a strictly self-defeating loop: misreporting immediately reduces future selection probability and leads to effective market exclusion.
>
> **Strict Profitability Gate.** The trading protocol enforces a hard economic constraint: a trade is executed only if it reduces system-wide cost (**$Δ𝒞 < 0**, Eq. 5). This criterion is computed from system-defined cost parameters—not self-reported values—so agents cannot fabricate profitable trades or sustain collusive “uncertainty-passing circles.” Any transfer that does not produce a real cost reduction is automatically rejected.
>
> Attempts at manipulation are either blocked by the profitability rule or penalized via posterior updates. While a full adversarial game analysis is future work, the current design is already robust in practice.
>
> ---
>
>
> >**Q2:** The computational complexity of the broker and trading protocol.
>
> **A5:**
> The Broker and trading protocol incur minimal overhead relative to VLM inference. Formally:
>
> **Broker Complexity ($O(N)$):**
>     Each step evaluates one utility sample per agent (Eq. 6), requiring $O(N)$ time and $O(1)$ TS updates. With typical pool sizes ($N \le 15$), this adds only tens of scalar operations per query—negligible compared to PFLOPs-scale model calls. Empirically, Agora’s average inference time ($\approx 2.5$s) remains highly competitive compared to router baselines.
>
> **Trading Protocol Complexity ($O(N^2)$):**
>     Searching for the best profitable trade over sender–receiver pairs and three uncertainty types yields $O(3N^2)$ per iteration. However, because each trade performs a greedy descent on a bounded cost function (Eq. 5), the market converges rapidly. Experimental results show that only $\approx 0.8$ trades per query are executed on average. Thus, the effective overhead is limited to a few hundred arithmetic operations-orders of magnitude lower than VLM inference.
>
> **Realistic Overhead (Empirical):**
>     * **Low Collaboration Cost:** Agora maintains a **COI $\approx 1.25$**, substantially lower than MoA/KABB-VLM (which reach up to 1.8).
>     * **Scalability:** Wall-clock time grows sublinearly with $N$, staying within 2.5–3.5s even for $N=15$.
>     * **Compute Efficiency:** Agora achieves **$\approx 5.6\times$ lower FLOPs** than Large-only strategies while maintaining comparable accuracy.
>
> **In summary,** while the Broker is $O(N)$ and trading is $O(N^2)$, the constants are tiny. Under realistic pool sizes and low trade frequency, the coordination cost is mathematically and empirically negligible compared to model inference.
>
> ---
>
>
>
>
> >**Q3:** Would Agora still work if agents have overlapping expertise or non-stationary cost profiles?
>
> **A6:**
> Yes. Agora does not require disjoint expertise or fixed costs. The **profitability rule** (Δ𝒞 < 0) automatically picks the cheapest effective resolver even when expertise overlaps, and the **Broker’s posterior updates** continuously adapt to cost drift. As a result, overlapping skills simply lead to cost-based tie-breaking, and non-stationary costs are absorbed by the MAB feedback loop without modifying the framework.

---

> > ### Author Response · Authors · 2025-11-26
> >
> > Thank you again for your careful assessment and constructive feedback.
> >
> > We would be grateful if you could let us know whether our rebuttal sufficiently resolves the issues you raised, or if there are any remaining points you would like us to clarify.
> >
> > We are glad to continue the discussion and address any further questions or comments you may have.

---

### Author Response · Authors · 2025-11-22
**General Response (1)**

Dear AC and Reviewers,

We sincerely thank you for your insightful reviews and constructive feedback. We are encouraged by the consensus that:

1.  **Novelty:** Our market-based perspective, treating uncertainty as a tradable asset to resolve information asymmetry, is novel and conceptually powerful (Reviewers $oqqC$, $VAy3$).
2.  **Theoretical Foundation:** The formalization of "Agnostic Coordination" and the Inefficiency Theorem provides a strong theoretical grounding for the problem (Reviewer $d1Fk$, $VAy3$).
3.  **Effectiveness:** Agora demonstrates substantial gains in accuracy (e.g., +8.5% on MMMU) and cost-efficiency across multiple benchmarks (Reviewers $oqqC$, $iVUH$).

We have updated our manuscript to incorporate your suggestions. Changes are highlighted in the revised PDF. Below, we address the three most common concerns regarding **Theoretical Optimality**, **Computational Complexity**, and **Generalization**.

>**1. Local vs. Global Optimality (Response to $iVUH$, $d1Fk$)**
A primary theoretical concern was whether Agora converges to a global optimum. We clarified in Section 3.3 that finding a global optimum in a decentralized, heterogeneous agent market is NP-hard. Therefore, Agora is designed to guarantee convergence to a **Locally Optimal Equilibrium** via deterministic greedy descent on the global cost function.

To quantify the quality of this equilibrium, we added an **Optimality Gap Analysis** (Appendix D). We constructed a Brute-Force Global Oracle on a subset of MMBench tasks. The results (Table R1 below) show that Agora's cost is only **0.80%** higher than the theoretical global minimum, while significantly outperforming greedy and random baselines. This empirically confirms that our Market-Aware Broker (MAB) initialization effectively guides the system into a high-quality basin of attraction.

**Table R1. Optimality Gap Analysis (from Appendix D)**

| Method                    | Avg. Cost (Normalized) | Gap to Global Opt. | Accuracy (%) |
| :------------------------ | :--------------------- | :----------------- | :----------- |
| Global Optimal (Oracle)   | 1.000                  | 0.00%              | 89.6         |
| **Agora (Decentralized)** | **1.008**              | **+0.80%**         | **89.5**     |
| Greedy (Local Only)       | 1.152                  | +15.2%             | 84.3         |

---


---


>**2. Computational Complexity & Real-Time Viability (Response to $oqqC$, $VAy3$)**
Concerns were raised regarding the overhead of the Broker and trading protocol. We have clarified in Section 3.3 and 4.1 that:

* **Complexity:** The Broker selection is $O(N)$ and the trading protocol is $O(N^2)$. For typical pool sizes ($N \le 15$), these operations consume milliseconds.
* **End-to-End Latency:** The "Average Inference Time" reported in our experiments (Figure 4 & Table 9) is **strictly end-to-end**, encompassing all coordination overheads. Agora achieves state-of-the-art accuracy with inference times (2.5s - 3.5s) comparable to standard routing baselines.
* **Tractability:** We added Appendix M.3 to explain that the Strategic Uncertainty Index ($U_{strategic}$) uses a closed-form marginal approximation, avoiding expensive model re-execution.

>**3. Robustness and Cross-Dataset Generalization (Response to $d1Fk$, $oqqC$)**
To address concerns about fairness and generalization:

* **Fairness:** We implemented a **Unified Multimodal Wrapper** for text-centric baselines (FrugalGPT, RouteLLM) to ensure fair multimodal comparison (Section 4.1).
* **Generalization:** We extended the head-to-head comparison against strong baselines (MoA, KABB) to three additional datasets (MathVision, InfoVQA, CC-OCR) in Appendix E.3. Agora consistently outperforms baselines across reasoning-heavy and perception-heavy tasks.

---

> ### Author Response · Authors · 2025-11-22
> **General Response (2)**
>
> In addition to correcting typos and formatting issues, we have revised the paper to rigorously address all reviewers' comments. The major updates are listed below:
>
> * **Optimality Gap Analysis (New Appendix D):** We added an empirical analysis comparing Agora against a Brute-Force Global Oracle, demonstrating that our decentralized equilibrium is within 0.8% of the theoretical global optimum.
> * **Cross-Dataset Comparison (New Appendix E.3):** We included head-to-head comparisons against MoA and KABB-VLM on MathVision, InfoVQA, and CC-OCR, demonstrating consistent superiority beyond MMBench.
> * **Computational Tractability (Revised Section 3.3 & New Appendix M.3):** We explicitly clarified the low computational complexity of the Broker and trading protocol, and detailed the efficient closed-form calculation of the Strategic Uncertainty Index.
> * **Fairness in Baselines (Revised Section 4.1):** We detailed the "Unified Multimodal Wrapper" implementation to ensure fair comparison with text-centric routers like FrugalGPT.
> * **Uncertainty Calibration (New Appendix M.2):** We added concrete calculation examples for uncertainty vectors and explained how structural normalization ensures cross-model comparability without expensive calibration.
> * **Visual Corrections:** We corrected the color descriptions in （Figure 1，Figure 5） and updated visualizations for better readability.
> * **Reference Cleanup:** We removed duplicate citations and corrected citation formats (e.g., Gale & Shapley).

---

### Author Response · Authors · 2025-12-03
**About Notes for the New AC: Thank You for Taking Responsibility for Our Submission**

Dear new AC,

Thank you very much for stepping in at this stage of the review process and taking over the handling of our submission. We sincerely appreciate your time and support.

To help you quickly navigate our work and the review history, we have prepared three short notes, included in the messages below:

- **Note 3:** Highlights of Additional Results during the Rebuttal & Discussion Phase
- **Note 2:** Highlights of Reviewer Concerns and Our Responses
- **Note 1:** Highlights of Our Key Contributions

We submitted our initial rebuttal and followed up with additional experimental results (including an optimality gap analysis and cross-dataset comparisons) to address the reviewers' comments. We kindly ask that the AC reviews our paper, rebuttal, and follow-up responses, taking into account that we have addressed the reviewers’ concerns thoroughly regarding theoretical guarantees, computational complexity, and generalization.

Thank you once more for taking on the oversight of our submission. We would be very happy to clarify or discuss further if any questions remain.

Best regards,

The Authors

---

> ### Author Response · Authors · 2025-12-03
> **Note 3: Highlights of Additional Results during the Rebuttal & Discussion Phase**
>
> To rigorously address reviewer concerns regarding theoretical optimality and generalization, we conducted extensive new experiments during the rebuttal. The key findings are summarized below.
>
> ### 1. Optimality Gap Analysis: Global vs. Local Equilibrium
> To address concerns about whether Agora’s decentralized equilibrium is suboptimal (**Reviewers iVUH, d1Fk**), we constructed a **Brute-Force Global Oracle** on a subset of MMBench tasks.
>
> **Result:** As shown in **Table R1** below (incorporated as **Table 4 in Appendix D** of the revised manuscript), Agora’s decentralized equilibrium achieves an average cost within **0.80%** of the theoretical global optimum found by the Oracle, while significantly outperforming greedy (+15.2% cost) and random (+84.0% cost) baselines. This empirically confirms that our Market-Aware Broker initialization effectively mitigates the limitations of local greedy descent.
>
> **Table R1. Optimality Gap Analysis (Global Oracle vs. Agora)**
>
> | Method | Avg. Cost (Normalized) | Gap to Global Opt. | Accuracy (%) |
> | :--- | :--- | :--- | :--- |
> | **Global Optimal (Oracle)** | **1.000** | **0.00%** | **89.6** |
> | **Agora (Decentralized)** | **1.008** | **+0.80%** | **89.5** |
> | Greedy (Local Only) | 1.152 | +15.2% | 84.3 |
> | Random Assignment | 1.840 | +84.0% | 76.5 |
>
> ### 2. Cross-Dataset Generalization (Head-to-Head Comparisons)
> To demonstrate robustness beyond MMBench (**Reviewers d1Fk, oqqC**), we extended our evaluation to three additional benchmarks: **MathVision** (Reasoning), **InfoVQA** (Perception), and **CC-OCR**.
>
> **Result:** In **Table R2** below (added to **Appendix E.3**), using the exact same heterogeneous agent pool (N=6), Agora consistently outperforms the strongest baselines, including Mixture-of-Agents (MoA) and KABB-VLM. For instance, on InfoVQA, Agora achieves **88.9%** vs. KABB’s 87.8% and MoA’s 87.6%, showing that our uncertainty trading mechanism effectively handles both perception-heavy and reasoning-heavy tasks.
>
> **Table R2. Head-to-Head Strategy Comparison on Additional Benchmarks (Accuracy %)**
>
> | Dataset | MoA (Aggregation) | KABB-VLM (Routing) | **Agora (Ours)** |
> | :--- | :--- | :--- | :--- |
> | **MathVision** | 42.1 | 42.5 | **44.3** |
> | **InfoVQA** | 87.6 | 87.8 | **88.9** |
> | **CC-OCR** | 80.2 | 80.4 | **81.2** |
>
> ### 3. Sensitivity Analysis of Utility Parameters
> To address concerns about hyperparameter sensitivity (**Reviewers iVUH, d1Fk**), we performed rigorous parameter sweeps on the Broker’s utility function (Eq. 6).
>
> **Result:** Extreme variations in parameters (e.g., learning rate, decay factor) resulted in marginal performance fluctuations (max accuracy drop < 3%). Furthermore, a "stress test" on uncertainty fusion weights showed that Agora maintains a **~22–24% uncertainty reduction** advantage over KABB even under extreme weight skews (**Table R3**, added to **Appendix G.1**).
>
> **Table R3. Sensitivity Analysis under Extreme Weight Configurations**
>
> | Weight Configuration | KABB ($U_{final}$) | Agora ($U_{final}$) | **Reduction** |
> | :--- | :--- | :--- | :--- |
> | Default (Balanced) | 0.210 | 0.160 | **-23.8%** |
> | Perceptual-Heavy | 0.195 | 0.148 | **-24.1%** |
> | Semantic-Heavy | 0.223 | 0.172 | **-22.9%** |
> | Inferential-Heavy | 0.218 | 0.169 | **-22.5%** |
>
> ---

---

> ### Author Response · Authors · 2025-12-03
> **Note 2 for the New AC: Highlights of Reviewer Concerns and Our Responses**
>
> We are encouraged by the reviewers’ positive assessments, recognizing our "market-based perspective" as **novel and conceptually powerful** (Reviewers oqqC, VAy3) and our "Inefficiency Theorem" as providing **strong theoretical grounding** (Reviewers d1Fk, VAy3).
>
> Below, we summarize the primary concerns raised and our responses.
>
> **(1) Theoretical Optimality (Global vs. Local)**
>
> * **Reviewer's Concern:** Reviewers (iVUH, d1Fk) noted that Agora’s trading protocol guarantees only a local equilibrium and asked about the gap to the global optimum.
> * **Our Response:** We clarified that finding a global optimum in heterogeneous markets is NP-hard. However, our new **Optimality Gap Analysis (Appendix D)** demonstrates that Agora’s equilibrium is empirically **near-optimal (0.8% gap)**. The Market-Aware Broker acts as a high-quality initializer, placing the system in a favorable basin of attraction for the subsequent deterministic greedy descent.
>
> **(2) Computational Complexity and Overhead**
>
> * **Reviewer's Concern:** Reviewers (oqqC, VAy3) worried that the Broker and trading protocol might introduce significant latency, making the system viable only for non-real-time applications.
> * **Our Response:** The coordination overhead is negligible.
>   * **Complexity:** Broker selection is $O(N)$ and trading is $O(N^2)$ for small $N$ ($\le 15$), consuming only milliseconds.
>   * **End-to-End Latency:** The "Average Inference Time" reported (e.g., 2.5s - 3.5s in Table 5) is **strictly end-to-end**, encompassing all overheads.
>   * **Tractability:** We added Appendix M.3 to clarify that the Strategic Uncertainty Index uses a closed-form marginal approximation, requiring **no additional VLM forward passes**.
>
> **(3) Fairness of Baselines and Generalization**
>
> * **Reviewer's Concern:** Reviewers (d1Fk, oqqC) asked if text-centric routers (e.g., FrugalGPT) were compared fairly in a multimodal setting and requested evidence of generalization.
> * **Our Response:**
>   * **Fairness:** We implemented a **Unified Multimodal Wrapper** (Section 4.1) for text-centric baselines, equipping them with the visual encoder from the strongest agent to ensure a fair comparison.
>   * **Generalization:** As detailed in Note 3, new experiments on **MathVision, InfoVQA, and CC-OCR** (Appendix E.3) confirm that Agora consistently outperforms MoA and KABB-VLM, validating that our gains are not limited to MMBench.
>
> **(4) Strategic Manipulation**
>
> * **Reviewer's Concern:** Reviewer oqqC asked how the system handles strategic agents that might misreport uncertainty.
> * **Our Response:** Agora relies on mechanism design, not trust. The **Profitability-Driven Trading Protocol** (Eq. 5) enforces a hard economic constraint computed from system-defined parameters, preventing "fake" profitable trades. Additionally, the Broker’s Thompson Sampling naturally penalizes agents that under-report uncertainty but fail, reducing their future selection probability.

---

> > ### Author Response · Authors · 2025-12-03
> > **Note 1 for the New AC: Highlights of Our Key Contributions**
> >
> > **Novelty and Conceptual Contribution**
> >
> > 1.  **Market-Based Coordination Paradigm:** We introduce **Agora**, a framework that reframes multi-agent coordination as a **decentralized market**. Unlike heuristic routers (e.g., KABB, MoA) that are "Cost-Agnostic" and "Structure-Agnostic," Agora treats uncertainty as a **tradable asset**.
> > 2.  **Assetization of Uncertainty:** We explicitly decompose uncertainty into **Perceptual, Semantic, and Inferential** dimensions, allowing agents to trade specific "doubts" based on their comparative advantages.
> >
> > **Theoretical and Practical Impact**
> >
> > 1.  **Principled Optimization:** We provide the **Inefficiency Theorem for Agnostic Coordination** (Theorem 1), proving that heuristic proxies inevitably lead to suboptimal outcomes. In contrast, Agora’s profitability-driven protocol (Eq. 5) guarantees monotonic cost reduction.
> > 2.  **SOTA Performance & Efficiency:** Extensive experiments show Agora achieves **+8.5% accuracy** on MMMU compared to baselines while reducing costs by over **3x**.
> > 3.  **Pareto Superiority:** As shown in Figure 6 (Appendix F), Agora establishes a superior **Pareto frontier**, achieving accuracy comparable to Large-only strategies (88.7% vs 89.2%) with **82.2% fewer FLOPs**. This establishes market-based coordination as a scalable path for economically viable multi-agent intelligence.

---

### Meta-Review · Area_Chair_z5X2 · 2025-12-26

**Summary:**

The reviewers find the paper’s core idea of reframing multi-agent coordination as a market for uncertainty and trading visual uncertainties via a broker to be timely and potentially impactful, with generally strong presentation and extensive experiments. The main concerns that drive the decision revolve around (i) whether the proposed protocol is guaranteed to reach a global or local equilibrium and how to interpret the “optimality” claim, (ii) scalability of the broker+trading mechanism in larger or more realistic settings, and (iii) clarity and soundness of how uncertainty is computed and how it generalizes across datasets/domains. Several reviewers also request clearer comparisons to relevant baselines and more evidence of generalization beyond the primary benchmark.

**Reviewer Concerns:**

The reviewer concerns have largely been addressed by the rebuttal, including:
- Local vs global optimality. The authors clarified that the method targets a locally optimal equilibrium under their market dynamics and added an “optimality gap” style empirical analysis against a brute-force oracle on a subset, helping contextualize the theory and its practical implications.
- Overhead discussion. The rebuttal provides a more explicit breakdown of broker selection/trading overhead and argues it is small relative to VLM inference, along with additional reporting on runtime and coordination cost.
- Baseline coverage. The authors added head-to-head comparisons on additional benchmarks and strengthened the empirical section with more ablations/sensitivity analyses, which partially addresses concerns about evaluation completeness and parameter fragility.

**Reviewer Scores:**

Reviewer oqgC and d1Fk would likely slightly increase their scores given the clarifications and additional analyses provided in the rebuttal.

---

### Decision · Program_Chairs · 2026-01-26

Accept (Poster)